# Cellular development and evolution of the mammalian cerebellum

Mari Sepp[1,15 ✉], Kevin Leiss[1,15 ✉], Florent Murat[1,2], Konstantin Okonechnikov[3,4], Piyush Joshi[3,4,5], Evgeny Leushkin[1], Lisa Spänig[3,4,5], Noe Mbengue[1], Céline Schneider[1], Julia Schmidt[1], Nils Trost[1], Maria Schauer[6], Philipp Khaitovich[7], Steven Lisgo[8], Miklós Palkovits[9], Peter Giere[6], Lena M. Kutscher[3,5], Simon Anders[1,10], Margarida Cardoso-Moreira[11], Ioannis Sarropoulos[1,14,16 ✉], Stefan M. Pfister[3,4,12,13,16 ✉] & Henrik Kaessmann[1,16 ✉]

The expansion of the neocortex, a hallmark of mammalian evolution[1,2], was accompanied by an increase in cerebellar neuron numbers[3]. However, little is known about the evolution of the cellular programmes underlying the development of the cerebellum in mammals. In this study we generated single-nucleus RNA-sequencing data for around 400,000 cells to trace the development of the cerebellum from early neurogenesis to adulthood in human, mouse and the marsupial opossum. We established a consensus classification of the cellular diversity in the developing mammalian cerebellum and validated it by spatial mapping in the fetal human cerebellum. Our cross-species analyses revealed largely conserved developmental dynamics of cell-type generation, except for Purkinje cells, for which we observed an expansion of early-born subtypes in the human lineage. Global transcriptome profiles, conserved cell-state markers and gene-expression trajectories across neuronal differentiation show that cerebellar cell-type-defining programmes have been overall preserved for at least 160 million years. However, we also identified many orthologous genes that gained or lost expression in cerebellar neural cell types in one of the species or evolved new expression trajectories during neuronal differentiation, indicating widespread gene repurposing at the cell-type level. In sum, our study unveils shared and lineage-specific gene-expression programmes governing the development of cerebellar cells and expands our understanding of mammalian brain evolution.

Establishing causal relationships between the molecular and phenotypic evolution of the nervous systems of humans and other mammals is a primary goal in biology. The expansion of the neocortex, considered to be one of the hallmarks of mammalian evolution[1,2], was accompanied by an increase in the number of cerebellar neurons[3]. The cerebellum varies substantially in size and shape across vertebrates[4]. In mammals, it contains more than half of the neurons of the entire brain[3] and is involved in cognitive, affective and linguistic processing, in addition to its well-established role in sensory–motor control[5]. The cellular architecture of the adult cerebellum has long been viewed as being relatively simple, with its characteristic Purkinje and granule cells organized into cortical layers and the deep nuclei neurons embedded inside the white matter, but it is increasingly recognized to exhibit rather complex regional specializations[6–8]. Our understanding of cerebellum development stems mostly from studies in rodents[6], although differences in the cellular composition of the human cerebellum have been recognized[8,9]. Recent single-cell transcriptome studies of the developing mouse[10–12] and human[13] cerebellum have provided new insights into gene-expression programmes in cerebellar cells, but an evolutionary analysis of the molecular and cellular diversity of the mammalian cerebellum across development is missing. In this study, we used single-nucleus RNA-sequencing (snRNA-seq) to examine cerebellum development from early neurogenesis to adulthood in three therian species: two eutherians (human and mouse) and a marsupial (opossum, *Monodelphis domestica*). Our analyses of these data, which provide an extensive resource (https://apps.kaessmannlab.org/sc-cerebellum-transcriptome), unveiled ancestral as well as species-specific cellular and molecular features of cerebellum development spanning around 160 million years of mammalian evolution.

[1]Center for Molecular Biology of Heidelberg University (ZMBH), DKFZ-ZMBH Alliance, Heidelberg, Germany. [2]INRAE, LPGP, Rennes, France. [3]Hopp-Children's Cancer Center Heidelberg (KiTZ), Heidelberg, Germany. [4]Division of Pediatric Neurooncology, German Cancer Research Center (DKFZ) and German Cancer Consortium (DKTK), Heidelberg, Germany. [5]Developmental Origins of Pediatric Cancer Junior Group, German Cancer Research Center (DKFZ) and German Cancer Consortium (DKTK), Heidelberg, Germany. [6]Museum für Naturkunde Berlin, Leibniz Institute for Evolution and Biodiversity Science, Berlin, Germany. [7]NHC Key Laboratory of Diagnosis and Treatment on Brain Functional Diseases, The First Affiliated Hospital of Chongqing Medical University, Chongqing, China. [8]Biosciences Institute, Newcastle University, Newcastle, UK. [9]Human Brain Tissue Bank, Semmelweis University, Budapest, Hungary. [10]BioQuant, Heidelberg University, Heidelberg, Germany. [11]Evolutionary Developmental Biology Laboratory, Francis Crick Institute, London, UK. [12]Department of Pediatric Hematology and Oncology, Heidelberg University Hospital, Heidelberg, Germany. [13]National Center for Tumor Diseases (NCT), Heidelberg, Germany. [14]Present address: Wellcome Sanger Institute, Cambridge, UK. [15]These authors contributed equally: Mari Sepp, Kevin Leiss. [16]These authors jointly supervised this work: Ioannis Sarropoulos, Stefan M. Pfister & Henrik Kaessmann. ✉e-mail: m.sepp@zmbh.uni-heidelberg.de; k.leiss@zmbh.uni-heidelberg.de; i.sarropoulos@zmbh.uni-heidelberg.de; s.pfister@kitz-heidelberg.de; h.kaessmann@zmbh.uni-heidelberg.de

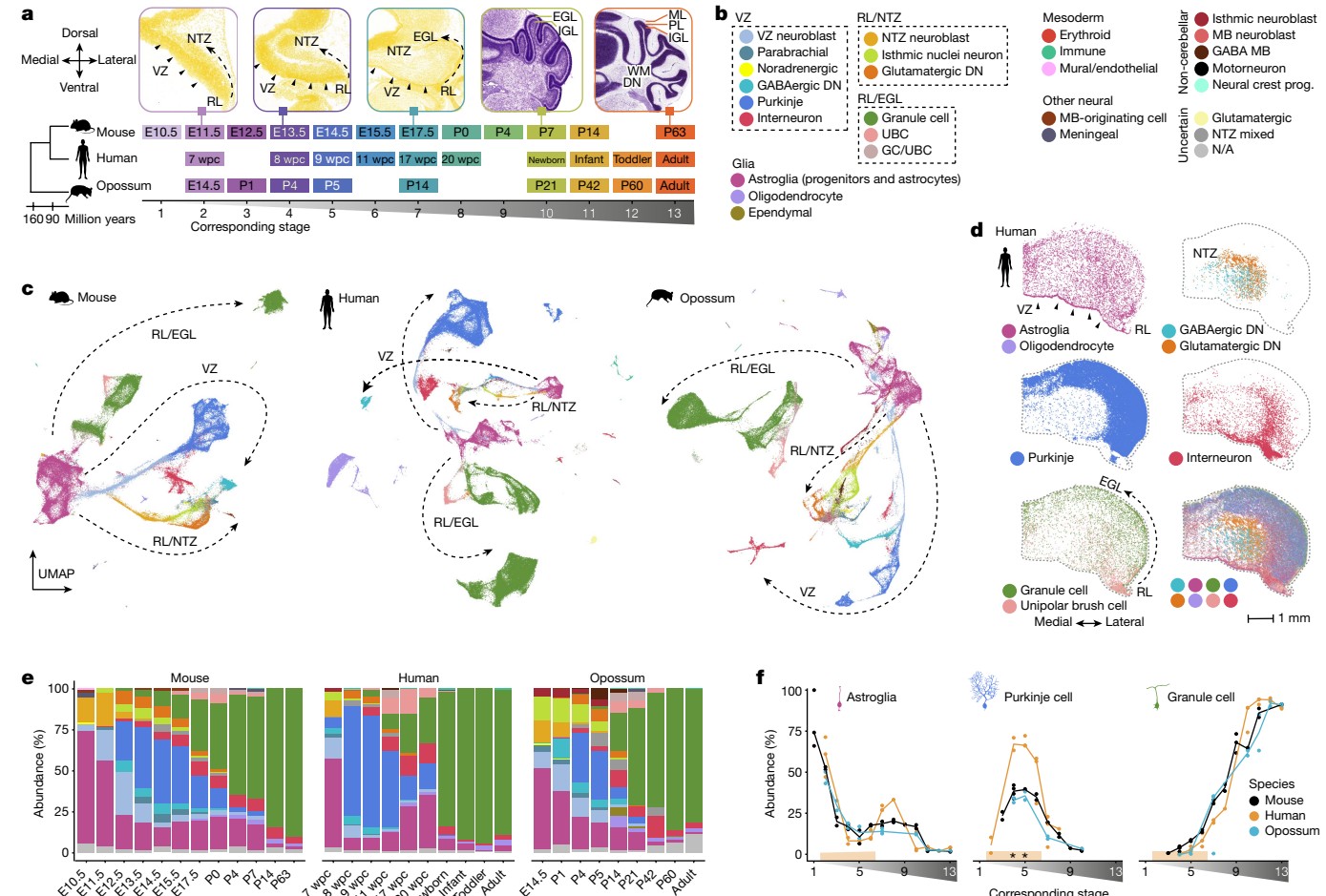

**Fig. 1 | Atlases of cerebellum development across mammals. a**, Bottom, stages sampled in mouse, human and opossum, aligned across species. Top, coronal sections of the mouse cerebellum[15] stained with HP Yellow or Nissl. DN, deep nuclei; EGL, external granule cell layer; IGL, internal granule cell layer; ML, molecular layer; NTZ, nuclear transitory zone; PL, Purkinje cell layer; RL, rhombic lip; WM, white matter. **b**, The detected cell types, grouped by their developmental origin. MB, midbrain; N/A, not available. **c**, Uniform manifold approximation and projection (UMAP) of 115,282 mouse, 180,956 human and 99,498 opossum cells coloured by cell type. Arrows indicate the broad neuronal lineages. **d**, Mapping of the main cerebellar cell types in the 12 wpc human cerebellum by alignment of the multiplexed smFISH data with 11 wpc snRNA-seq data. **e**,**f**, Relative cell-type abundances across developmental stages in the whole datasets (**e**) or among the cerebellar cells (**f**). For adult human, only data from cerebellar lobes is included. In **c**–**e**, colours are as in **b**. In **f**, stages are aligned as in **a**, the line denotes the median of biological replicates, orange shading shows stages with representative sampling in human, and asterisks mark differences in relative abundances compared with mouse and opossum. Panel **a** is from the Allen Mouse Brain Atlas[15].

## Map of cerebellum development in mammals

The cerebellum has a protracted course of development, extending from early embryogenesis well into postnatal life[6]. To characterize mammalian cerebellum development, we produced snRNA-seq data for cerebella from 9–12 developmental stages in mouse, human and opossum (Fig. 1a and Extended Data Fig. 1a,b). We acquired high-quality transcriptional profiles of 395,736 cells sequenced in 87 libraries, and used linked inference of genomic experimental relationships[14] (LIGER) to integrate datasets from all stages for each species (Extended Data Fig. 1 and Supplementary Table 1). Because cerebellum development is best understood in mouse, we used known cell-type markers[6,12] and public in situ hybridization data[15,16] to build a hierarchical annotation of the mouse dataset. We then transferred this to the human and opossum datasets by pairwise integration of the datasets within the orthologous gene-expression space, followed by manual curation to account for biological and technical variance between the datasets (Extended Data Fig. 2a,b and Supplementary Tables 2–5). Consistent with the ongoing efforts in establishing cell ontologies[17], we grouped the cells into broad lineages based on their developmental origin, into cell types (25 across

the three species), into cell differentiation states (43; hereafter referred to as cell states), and for 12 cell states that displayed remaining variability, we further split the cells into subtypes (36–38 in each species; Fig. 1b,c and Extended Data Fig. 2c). As a validation of the annotations, we mapped spatial expression patterns of 74 marker genes using multiplexed single-molecule fluorescence in situ hybridization (smFISH) in the 12 week post-conception (wpc) human cerebellum, and located the cell types by aligning the spatial data with our snRNA-seq data (Fig. 1d, Extended Data Figs. 3–6 and Supplementary Table 6).

To establish correspondences between the developmental stages sampled in mouse, human and opossum, we performed stagewise cross-species comparisons of (1) synthetic bulk transcriptomes using Spearman's correlations of orthologous variable gene expression; (2) pseudoages[18] based on the median age of neighbouring mouse cells in the cross-species integrated manifold; and (3) cellular composition by measuring similarities at the level of cell states (Extended Data Fig. 2d–g). Combining these approaches, we inferred, for instance, that among the sampled stages, the cerebellum of the newborn human most closely resembles that of a one-week-old mouse and a three-week-old opossum (Fig. 1a). The estimated stage correspondences are supported

by the morphological characteristics of the developing cerebellum in the three species, and agree with the correspondences previously established by jointly considering multiple somatic organs[19] (Extended Data Fig. 2h–k).

On the basis of the expression patterns of orthologous genes that are differentially expressed within each species, we created a consensus classification of the cellular diversity in the developing mammalian cerebellum (Extended Data Figs. 2c and 3–6). UMAP embeddings of the three datasets show a radiation of lineage-committed cells stemming from a population of proliferating neural progenitors, with cells ordered by age along the trajectories (Fig. 1c and Extended Data Figs. 1f and 7a). The major neuronal trajectories (Fig. 1b,c) reflect the known cerebellar germinal zones[6]: the ventricular zone (VZ), which produces cerebellar γ-aminobutyric acid-producing (GABAergic) neurons; the early rhombic lip, which gives rise to glutamatergic neurons assembling at the nuclear transitory zone (RL/NTZ); and the late rhombic lip, which is associated with a secondary germinal zone in the external granule cell layer (RL/EGL). The detected VZ cell populations include parabrachial neurons (marked by *LMX1B* and *LMX1A* expression) and a small group of noradrenergic neurons (*LMX1B* and *PHOX2B*), both of which migrate to the brainstem during development[20], as well as all cerebellar GABAergic neuron types; that is, GABAergic deep nuclei neurons (*SOX14*), Purkinje cells (*SKOR2*) and interneurons (*PAX2*) (Fig. 1b–d and Extended Data Fig. 3a–e). Among the RL/NTZ cells (*SLC17A6*) we discerned extra-cerebellar isthmic nuclei neurons (*PAX5* and *SCG2*) that locate to the anterior NTZ during development, and glutamatergic deep nuclei neurons (*NEUROD6*) (Extended Data Fig. 4a–f,l). In the RL/EGL trajectory we observed granule cells and unipolar brush cells (UBCs) transitioning from progenitors (*ATOH1*) and differentiating cells (*PAX6*) towards defined granule cell (*GABRA6*) and UBC (*LMX1A* and *EOMES*) states (Extended Data Fig. 5a–e). Along all major neuronal trajectories, cells from different cell-type lineages often clustered together at the earliest differentiation states and are designated as VZ neuroblasts, NTZ neuroblasts and granule cell/UBC (GC/UBC) (Fig. 1b,c and Extended Data Figs. 3–5). Among these, the GC/UBC progenitor population reflects a true cell state, as we detected proliferating (*TOP2A*) cells co-expressing granule cell and UBC lineage markers (*ATOH1*, *OTX2*, *LMX1A* and *EOMES*) in the 12 wpc human cerebellum. These cells mostly mapped to the external rhombic lip and proximal EGL, although co-expression of *EOMES, ATOH1* and *TOP2A* was seen even in the distal EGL cells (Extended Data Fig. 5e). By contrast, further dissection of the VZ neuroblasts, often based on developmental stage (Supplementary Information), revealed differential expression of known markers of the VZ-derived cell types (for example, parabrachial and noradrenergic neuron marker *LMX1B*[20] in the early neuroblasts, interneuron marker *PAX2*[6] in the late neuroblasts; Extended Data Fig. 3g,h), consistent with these cells already being lineage-committed, despite common differentiation programmes. This is in line with the pan-neuronal transcriptional state previously observed in early neuroblasts across the whole developing mouse brain[18].

In all three datasets, neural progenitors with temporally progressing transcriptional states, glioblasts and astrocytes (*SLC1A3* and *AQP4*) form the most abundant glial lineage (Fig. 1b–e and Extended Data Fig. 6a–f)—hereafter collectively referred to as astroglia. In the oligodendrocyte lineage, we discerned proliferating oligodendrocyte progenitor cells (OPCs) (*PDGFRA*), committed oligodendrocyte precursors (*TNR*) and postmitotic oligodendrocytes (*MAG*) (Extended Data Fig. 6a–d,f). Additionally, in human and opossum, we detected an intermediate cell population between glioblasts and OPCs, probably representing a pre-OPC state[18] (*EGFR*) (Extended Data Fig. 6a–d,f). We distinguished ependymal cells (*SPAG17*) in the mouse and opossum but not in the human dataset, and in opossum, we further identified ependymal progenitors that share transcriptional traits with glioblasts but express ciliogenesis-related *SPAG17* (Extended Data Fig. 6a–d,f). We also detected neural crest- and mesoderm-derived cell types (meningeal,

immune (mostly microglia), vascular (mural and endothelial) and erythroid) and small groups of neural cells from neighbouring brain regions, resulting from the migration of a midbrain-originating cell population (*LEF1*) to the cerebellar primordia[12] or sample contamination (isthmic and midbrain neuroblasts, GABAergic midbrain cells and motor neurons; Fig. 1b,c and Extended Data Fig. 6j).

A comparison of cell-type abundances across development revealed highly dynamic patterns that are similar in the three species (Fig. 1e and Extended Data Fig. 7b) and consistent with the current understanding of cerebellum development[6]. Astroglia (progenitors) are most abundant at the earliest stages, Purkinje cell relative abundances peak at the transition from embryonic to fetal development (embryonic day (E)13.5–E15.5 in mouse), and granule cells dominate at late stages, outnumbering all other cell types already in postnatal day (P)4 mouse, newborn human and P21 opossum (Fig. 1e,f). We note that our sampling of human cerebellum fragments for stages from 17 wpc onwards might not precisely reflect cell-type proportions in the entire cerebellum (Extended Data Fig. 7c and Supplementary Table 1). Thus, we focused on early stages that are less influenced by sampling differences, and applied Bayesian modelling to compare the relative cell-type abundances across matched developmental stages between species (Fig. 1f and Extended Data Fig. 7d). The most striking difference is an approximately twofold higher Purkinje cell proportion in human compared with mouse and opossum at two stages when their relative abundances peak during development (8–9 wpc in human; Fig. 1f). The difference remains statistically significant even when additionally considering the VZ neuroblasts (Extended Data Fig. 7d,e). A meta-analysis of 19 mouse (E13.5–E15.5) and 20 human (8–11 wpc) cerebellum samples from this and other studies[10,11,13,21,22] confirmed the significantly higher Purkinje cell abundances in human (Extended Data Fig. 7f). This change in Purkinje cell dynamics in the human lineage could be related to differences in developmental durations between species and/or the unique presence of basal progenitors in the human cerebellum[9] that may serve as an additional pool of Purkinje cell progenitors. Together, our snRNA-seq atlases provide a comprehensive view of cerebellar cell types in mammals, revealing the largely conserved developmental sequence of their generation but also notable differences in human Purkinje cell dynamics.

## Spatiotemporal cell-type diversification

Although traditionally viewed as a brain region with a simple cellular architecture, the adult cerebellum is increasingly recognized to exhibit regional specialization of cell types and a complex pattern of functional compartments organized around the parasagittal ALDOC-positive and -negative Purkinje cell domains[6–8]. To characterize the molecular diversification of cerebellar cells during development, we examined the within-cell-type heterogeneity in our snRNA-seq atlases. We divided mouse Purkinje cells into four developmental subtypes (Fig. 2a). Combinatorial expression of the transcription factor genes *Ebf1* and *Ebf2* differentiates the subtypes along the spatial and temporal axes: Purkinje subtypes that locate medially in the developing cerebellum (named by their marker genes as RORB and CDH9 types) display higher *Ebf1* levels than the lateral subtypes (FOXP1 and ETV1), whereas *Ebf2* is upregulated in the late-born subtypes (CDH9 and ETV1) compared with the early-born subtypes (RORB and FOXP1) (Fig. 2a–c and Extended Data Fig. 3i–l). The genes with variable expression across these developmental subtypes are enriched for the cadherin family of adhesion molecules (homophilic cell adhesion, $P < 10^{-15}$), supporting their proposed role in providing a molecular code for the formation of Purkinje cell domains[23]. To link the developmental Purkinje subtypes to the recently described adult subtypes[7], we calculated correlation coefficients between their expression profiles using genes differentially expressed in both groups. We detected similarities between early-born subtypes (RORB and FOXP1) and adult *Aldoc*-positive subtypes that

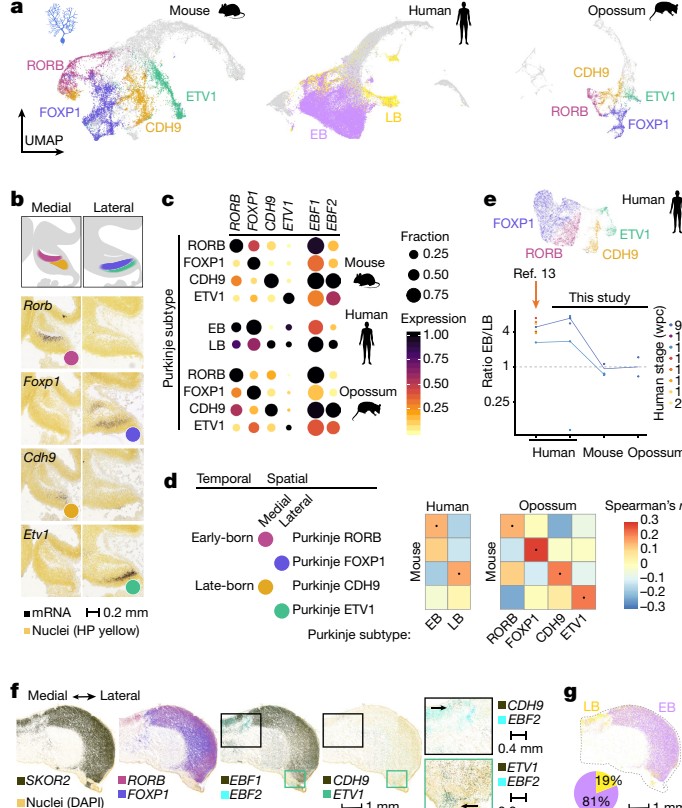

**Fig. 2 | Spatiotemporally defined Purkinje cell subtypes. a**, UMAPs of 23,255 mouse, 49,399 human and 8,973 opossum cells assigned to the Purkinje cell lineage. Colours highlight the developmental subtypes. EB, early born; LB, late born. **b**, E13.5 mouse cerebellum RNA in situ hybridization data[15] for Purkinje subtype markers. Medial and lateral sagittal sections are shown. **c**, Expression of key markers in mouse, human and opossum Purkinje subtypes. Dot size indicates the fraction of cells expressing a gene, and colour shows the mean expression level scaled by species and gene. **d**, Cross-species Spearman's correlation coefficients between orthologous variable gene (*n* = 107) expression profiles from Purkinje subtypes. Dots denote the highest correlation for each column. **e**, Top, UMAP of 14,246 human 9–20 wpc Purkinje cells from a published dataset[13] with cells coloured by subtype. Bottom, ratio of early-born to late-born Purkinje cell numbers in fetal samples. Biological replicates are shown with dots, and lines indicate stage median. Corresponding stages (Fig. 1a) are shown for mouse and opossum. **f**, Detection of Purkinje cell markers in the 12 wpc human cerebellum by smFISH. Close-up views are indicated with rectangles; arrows point to double-positive regions. **g**, Mapping of the early- and late-born Purkinje cells in the 12 wpc human cerebellum by alignment of the smFISH data with 11 wpc snRNA-seq data. The pie chart indicates the percentage of cells by group.

are enriched in cerebellum hemispheres; late-born medial subtype (CDH9) and *Aldoc*-positive subtypes enriched in posterior vermis; and late-born lateral subtype (ETV1) and *Aldoc*-negative subtypes (Extended Data Fig. 3m). Together, these results suggest that Purkinje cells with distinct settling patterns are specified not only by their 'birthdate'[6,24] but also by 'birthplace'.

Based on key marker genes and the correlation of orthologous variable gene expression, we identified the same four developmental Purkinje subtypes in opossum, whereas in human we reliably distinguished two subgroups (*EBF1/2*-low and -high); however, patterned expression of subtype markers indicated additional diversity (Fig. 2a,c,d and Extended Data Fig. 3l). To investigate this further, we reanalysed an independent snRNA-seq dataset of human fetal (9–20 wpc) cerebellum[13] and explored the expression of subtype markers in our 12 wpc spatial dataset. These analyses confirmed the presence of all four Purkinje

subtypes in the human fetal cerebellum (Fig. 2e,f and Extended Data Fig. 3l,n). In the 12 wpc Purkinje cell compartment (*SKOR2*), early-born Purkinje cell markers *FOXP1* and *RORB* exhibited widespread expression, whereas the late-born Purkinje cell marker *EBF2* was detected in restricted spatial domains, where *CDH9*-positive cells were located medially and *ETV1*-positive cells laterally (Fig. 2f). Comparison of Purkinje subtype prevalence across the three species revealed increased numbers of early-born Purkinje cells in human fetal samples (Fig. 2e,g). In sum, although Purkinje cell patterning is conserved in mammals, the subtype ratios shifted in the lineage leading to humans, probably facilitated by augmented generation of early-born Purkinje cells.

We further defined 16 subtypes among the other neuronal cell types, 13 of which were detected in all 3 species (Extended Data Figs. 3–5). We distinguished 5 homologous subtypes of GABAergic interneurons: an early-born type (*ZFHX4*) that in the 12 wpc human cerebellum is detected in the forming deep nuclei, and 4 types that we matched to the transcriptionally-defined adult subtypes with layer-specific localizations in the mouse cerebellar cortex[7] (Extended Data Fig. 3o–t). An additional unknown cell group (*MEIS2*) is present in the opossum. Among the RL/NTZ cells, we distinguished 2 subtypes of glutamatergic deep nuclei neurons located ventrally (*LMO3*) and posteriorly (*LMX1A*) in E13.5 mouse NTZ, as also reported previously[12,25], and three subsets of isthmic nuclei neurons expressing markers related to somatostatin (*SST*), dopaminergic (*NR4A2*) or cholinergic (*SLC5A7*) identities (Extended Data Fig. 4g–k). The latter subtype was not detected in the human snRNA-seq dataset, yet we observed cells co-expressing *SLC5A7* and *SLC17A6* in the 12 wpc cerebellum by smFISH (Extended Data Fig. 4m). Consistent with prior work on the adult mouse cerebellum[7], developing UBCs and granule cells display continuous variation in all three species (Extended Data Fig. 5f–i). Differentiating granule cells clustered into early and late populations, and in mouse and opossum we additionally detected a distinct *OTX2*-expressing subset (Extended Data Fig. 5f,i). The latter was not distinguished in the human snRNA-seq dataset due to sampling biases, given that we detected *OTX2*-expressing granule cells by spatial mapping in the domain proximal to the rhombic lip at 12 wpc (Extended Data Fig. 5j). Comparing the three groups to the granule cell subtypes defined in the adult mouse cerebellum[7], we observed correspondences with the adult subtypes that are spatially invariant (early), enriched in the posterior hemisphere (late) or nodulus (OTX2) (Extended Data Fig. 5k), supporting the notion that the topographic granule cell heterogeneity is at least partially driven by the temporal ordering of granule cell differentiation[26]. We classified UBCs into two subsets: one strongly expresses the canonical pan-UBC marker *EOMES* and is co-labelled by markers of known UBC subtypes[6,7] (*TRPC3*, *GRM1* and *CALB2*), whereas the other is a so far uncharacterized *EOMES*-low subset that expresses *HCRTR2* (Extended Data Fig. 5f,g,i). We confirmed the presence of UBCs expressing *TRPC3*, *HCRTR2* or both in the human 12 wpc cerebellum by smFISH, and observed the brush-like phenotype of the HCRTR2-positive cells in the mouse P7 cerebellum by immunohistochemistry (Extended Data Fig. 5l–n). Thus, the *HCRTR2*-expressing subset represents a previously unappreciated, mammalian-conserved UBC subtype.

The neuronal diversity in the cerebellum aligns with heterogeneity among progenitors. In the three species, embryonic neurogenic progenitors display a gradient of molecular variation along the neuroepithelium, include a group of potentially apoptotic cells (*NCKAP5* low, *BCL2L11* high), and have higher expression of cell cycle-related genes compared with the later-emerging bipotent (that is, producing both interneurons and parenchymal astrocytes[6,27]) and gliogenic progenitors (producing parenchymal and Bergmann astrocytes[28]) (Extended Data Fig. 6a–g). Spatial mapping of progenitors (marked by *SOX2*, *NOTCH1*, *PAX3* and *TOP2A*) in the human 12 wpc cerebellum revealed their presence not only in the VZ and RL, but also scattered in the prospective white matter (PWM) and cortical transitory zone, consistent with the marker gene-expression patterns in the E15.5 mouse

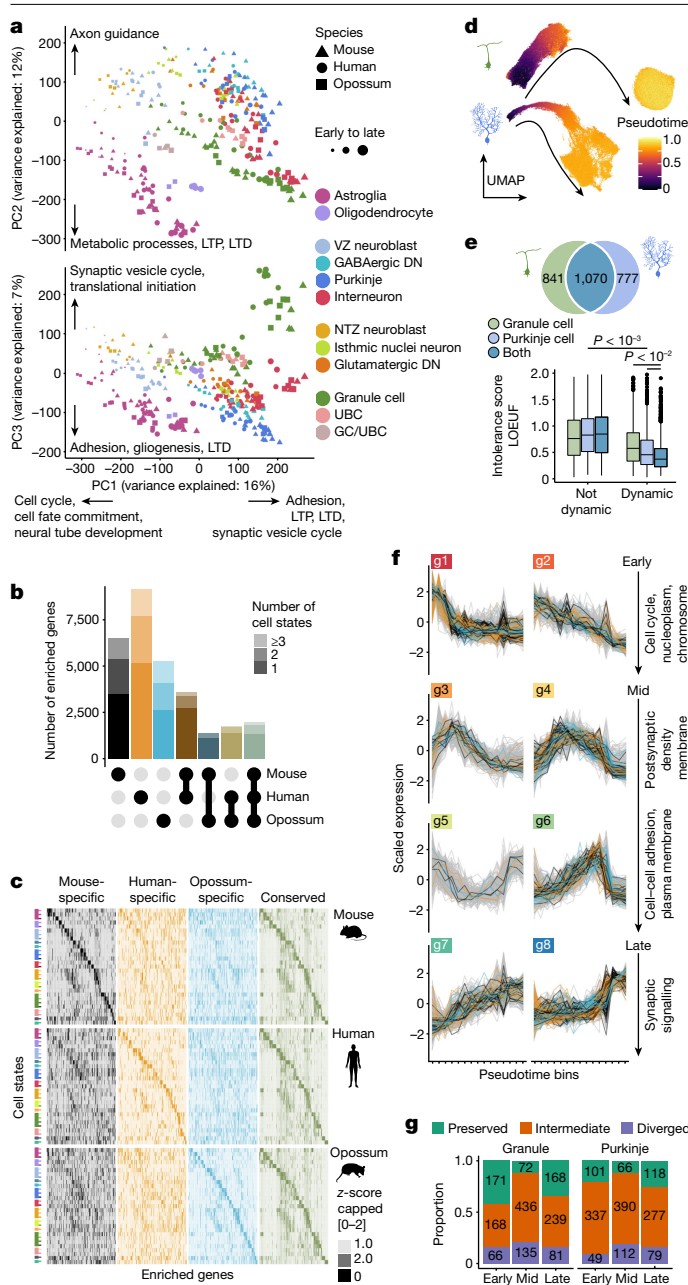

**Fig. 3 | Cell-type-defining transcriptional programmes. a**, Principal components analysis based on 10,276 orthologous genes expressed in all species. Data points represent cell-type pseudobulks for each replicate. Examples of enriched gene ontology terms and pathways for the genes loaded to principal components 1–3 are indicated. LTD, long term depression; LTP, long term potentiation; PC, principal component **b,c**, Numbers (**b**) and expression patterns (**c**) of species-specific and conserved markers. Twenty genes per state are shown in **c. d**, UMAPs of cells from granule cell and Purkinje cell lineages aligned across species and coloured by diffusion pseudotime values. **e**, Intolerance to functional mutations in human population (low values indicate intolerance) for genes dynamic or non-dynamic across differentiation of granule cells, Purkinje cells, or both neuron types in all species. Numbers of dynamic genes per category are indicated at the top. Boxes display interquartile range, whiskers extend to values within 1.5× interquartile range, and the line marks the median. Adjusted $P$ values were calculated via two-sided permutation tests of pairwise comparisons between all categories. LOEUF, loss-of-function observed/expected upper bound fraction. **f**, Clusters (g1–g8) of gene-expression trajectories during granule cell differentiation, ordered from early to late differentiation based on the mean centre-of-mass values of the confident cluster members' trajectories. Strongly preserved trajectories of the orthologues are highlighted. Examples of enriched gene ontology terms for the genes with preserved trajectories are shown for each row (excluding g5). **g**, Proportion of human genes with strongly preserved, intermediate and diverged trajectories in early, mid (excluding g5) and late clusters.

datasets. We aggregated expression values into cell-type pseudobulks for each sample and performed principal components analysis using orthologous genes that are expressed in all species. The two first principal components order samples by age and split glial and neuronal cells, and the third principal component further separates the neuronal types; in a separate analysis only of neurons the first two principal components arrange samples by age and cell type (Fig. 3a, Extended Data Fig. 8a–c and Supplementary Table 7). These patterns indicate that gene-expression variance in the developing cerebella is to a large extent explained by developmental and cell-type signals that are shared across the species. Thus, we sought to identify the core gene-expression programmes that underlie the identity of cerebellar cell types, similar to previous comparative cross-species approaches[29–31]. We called enriched genes (markers) for each cell state (Supplementary Information) and determined their overlap across species. On average, 58% of the markers in each species are cell-state-specific, 26% are enriched in two cell states and 15% are enriched in three or more states (Extended Data Fig. 8d). Similarly to observations for the adult motor cortex[29], many of the markers displayed cell-state enrichment in only one species (Fig. 3b,c and Extended Data Fig. 8e). Nevertheless, each cell-state category exhibited a set of conserved markers (Supplementary Table 8) that are likely to represent genes that drive cerebellar cell-type identities, given that their expression specificity has been retained for at least 160 million years of evolution. Consistently, conserved markers are associated with pertinent gene ontology terms, including 'neural tube development' for progenitors and 'ensheathment of neurons' for oligodendrocytes (Supplementary Table 9). In terms of molecular functions, the conserved markers are enriched for extracellular matrix and adhesion proteins, transmembrane transporters, ligands and receptors, transcription factors and proteins involved in plasma membrane and vesicle dynamics (Extended Data Fig. 8f,g and Supplementary Table 9). Sharing of the conserved markers typically involves closely related cell states (Extended Data Fig. 8h). At states of differentiation, when cell-type or subtype specification is ongoing, there is an enrichment of transcription factor genes among the conserved markers (Extended Data Fig. 8g), in line with the central role of transcription factors in inducing cell-type-specific downstream effector genes[32]. The conserved markers across all states include 185 transcription factors (Supplementary Table 8) and many of these are known to function in specific cerebellar cell types (for example, *ESRRB* and *FOXP2* in Purkinje cells,

cerebellum (Fig. 1d and Extended Data Fig. 6h,i). Markers of bipotent (*GLIS3*) and gliogenic (*TNC*) progenitors showed reverse gradients, with *TNC* expressed highly in ventricular cells within and near the RL, and in the cortical transitory zone, whereas *GLIS3* was detected in the more distal VZ and the PWM (Extended Data Fig. 6i). In line with the presence of two late progenitor types and our previous observations in the mouse[25], we identified two glioblast populations in all three species, PWM glioblasts and astroblasts (Extended Data Fig. 6a–d,f). Collectively, these results suggest developmental specification of the regional heterogeneity among the cerebellar cell types and highlight the overall conservation of the cellular architecture, including neural subtypes, of the developing cerebellum across mammals.

## Cell-type-defining programmes

Having established cross-species correspondences between developmental stages, as well as cell types and states, we next sought to characterize global gene-expression patterns in the three cerebellum

*PAX2* in interneurons, and *ETV1* in granule cells[33]). However, this list also includes potential novel regulators such as interneuron-enriched *PRDM8* and *BHLHE22*, known to form a repressor complex involved in pallial circuit formation[34], and *SATB2*, which is enriched in differentiating granule cells and primarily recognized as a determinant of neocortical upper-layer neurons[35] (Extended Data Fig. 8i). Among all mouse and human transcription factor markers, conservation of expression specificity is associated with higher expression levels of their predicted target genes in the respective cell states, as revealed by SCENIC[36] modelling (Extended Data Fig. 8i,j and Supplementary Table 10). Thus, the identified conserved transcription factor code provides a shortlist of candidates for elucidating the mechanisms of cerebellar cell-type specification.

The above analyses are based on discrete cell categories but developmental processes are inherently continuous. Thus, we set out to delineate the conserved gene-expression cascades across differentiation of the principal cerebellar neuron types: Purkinje and granule cells. We integrated cells from the two neuronal lineages across all species and calculated diffusion pseudotime[37] (Fig. 3d and Extended Data Fig. 9a). Corresponding cell states across species display comparable pseudotime values and the distribution of the values across stages is in accordance with the different generation modes of the two neuron types (Extended Data Fig. 9b,c)—transient for Purkinje cells and protracted for granule cells[6,38]—corroborating the alignment of cells across species and stages. Next, we identified orthologous genes with dynamic expression during neuronal differentiation in all three species (Supplementary Information). The two neuron types share 56–58% of the dynamic genes, suggesting considerable overlap in their differentiation programmes (Fig. 3e). The dynamic genes show low tolerance to heterozygous inactivation in human population[39], with those dynamic in both neuron types under the strongest functional constraint (Fig. 3e). This is in line with studies linking phenotypic severity to expression pleiotropy[19,40]. Additionally, dynamic genes are enriched for transcription factors and genes associated with inherited developmental diseases affecting the nervous system[41] (Extended Data Fig. 9d,e). We further focused on neurodevelopmental and neurodegenerative diseases[13] and malignancies[42] that are directly linked to cerebellar functions and cell types. Genes associated with cerebellar malformations, spinocerebellar ataxia and medulloblastoma are enriched among the dynamic genes shared between the two neuron types, whereas high-confidence risk genes of autism spectrum disorders and intellectual disability are additionally enriched among the genes that are dynamic in Purkinje cells only (Extended Data Fig. 9e). These results indicate that many of the cerebellar disease-linked genes are likely to affect more than one neuron type.

Next, we grouped the genes that are dynamic across neuronal differentiation in clusters based on their expression trajectories, and determined centre-of-mass values for the individual trajectories within each cluster to ensure comparable distributions across species (Fig. 3f and Extended Data Fig. 9f,g). By comparing the cluster assignments of the orthologues, we assigned the genes into 3 trajectory conservation groups: (1) 23% of genes, on average, were defined as strongly preserved with orthologues confidently assigned (cluster membership > 0.5 and *P* > 0.5) to the same cluster; (2) 17% of genes were defined as diverged, based on the differential cluster assignment (*P* < 0.05) of at least one of the orthologues; (3) the remaining 60% of genes were defined as having intermediate trajectory conservation (Fig. 3g). Consistently, the maximum distances between the orthologues' trajectories increase progressively from the most-preserved to least-preserved gene group (Extended Data Fig. 9h). Genes with strongly preserved trajectories expressed early during differentiation are enriched for functions in the cell nucleus, while late-expressed genes have functions in synaptic signalling (Fig. 3f and Extended Data Fig. 9f). There are 30 and 43 transcription factor genes among the genes with strongly preserved trajectories in granule and Purkinje cells, respectively, including several transcription factors with well-characterized roles in neuronal differentiation in the cerebellum (for example, *PTF1A* and *RORA* for Purkinje cells, and *PAX6* and *ETV1* for granule cells[6]; Supplementary Table 11). We ranked the transcription factors on the basis of the centre-of-mass values, and confirmed the expression patterns of many of the transcription factors using mouse in situ hybridization data[15] (Extended Data Fig. 9i,j). Thus, these analyses reveal a conserved programme of transcription factors, the expression of which follows closely matched patterns during Purkinje or granule cell differentiation in the three species.

## Evolutionary change in gene expression

Changes in gene-expression programmes are considered major drivers of the evolution of species-specific phenotypic features. We therefore aimed to systematically identify genes that display distinct expression patterns in cerebellar cells in one of the three species. First, we traced genes with diverged expression trajectories in Purkinje or granule cells (Fig. 3g). Using opossum as an evolutionary outgroup, we assigned the trajectory changes to the mouse or human lineage (that is, polarized the changes; Fig. 4a). In granule cells, we found a relative excess of trajectory changes in the human lineage ($P < 10^{-6}$, binomial test), whereas in Purkinje cells, we found similar numbers of changes in the human and mouse lineages (Fig. 4b and Supplementary Table 11). In each lineage, only a few (1–4) genes have changed trajectories in both cell types, suggesting that changes in regulatory programmes are largely cell-type-specific. Nevertheless, genes with human-specific changes in either cell type share enrichments for functions related to synaptic membrane and glutamatergic synapse (FDR < 0.05, Supplementary Table 12). Overall, the trajectory changes include shifts in both directions along the differentiation path (towards less or more mature states), and involve all types of trajectories (Fig. 4a and Extended Data Fig. 10a,b). We attempted to obtain a quantitative measure of the amount of change for each gene by assessing the maximum and minimum pairwise distances between the trajectories of orthologues from the three species (Supplementary Information). This approach identified *SNCAIP* (which encodes synuclein-α interacting protein) and *MAML2* (which encodes a transcriptional coactivator in the Notch signalling pathway) as having evolved the strongest changes in expression trajectories during granule cell and Purkinje cell differentiation, respectively, in the human lineage (Fig. 4c,d and Extended Data Fig. 10c,d). Notably, *SNCAIP* is frequently duplicated in group 4 medulloblastoma[43], a childhood brain tumour that has been difficult to model in mouse[44]. Additionally, 12 genes associated with autism spectrum disorder and/or intellectual disability show trajectory differences (Supplementary Table 11), including *MYT1L* and *KANSL1* in granule cells (Fig. 4d) and *SMARCA2*, *DIP2C* and *FOXP1* in Purkinje cells (Extended Data Fig. 10d).

We next sought to identify genes with an even more fundamental expression change; that is, genes displaying presence or absence expression differences between the species in one or more of the eight main cerebellar cell types (Fig. 4e). To mitigate technical biases in cross-species expression level comparisons from snRNA-seq data, we took a conservative approach: we analysed exonic read pseudobulks of cell types and replicates, considered only the orthologous genes with comparable genomic annotation in the three species, assessed relative expression levels within each species, and required at least fivefold differences in absolute expression levels to call a difference between species (Extended Data Fig. 11a–g and Supplementary Information). Out of the 7,062 orthologues included in this analysis, 1,077 (15.3%) displayed presence or absence expression differences in at least one cell type. After polarizing the changes, we found, on average, 62 gains and 19 losses in the human lineage, and 33 gains and 31 losses in the mouse lineage per cell type (Fig. 4e and Supplementary Table 13). The identified differences are consistent with the expression levels of the affected genes in mouse, human and opossum cerebellum

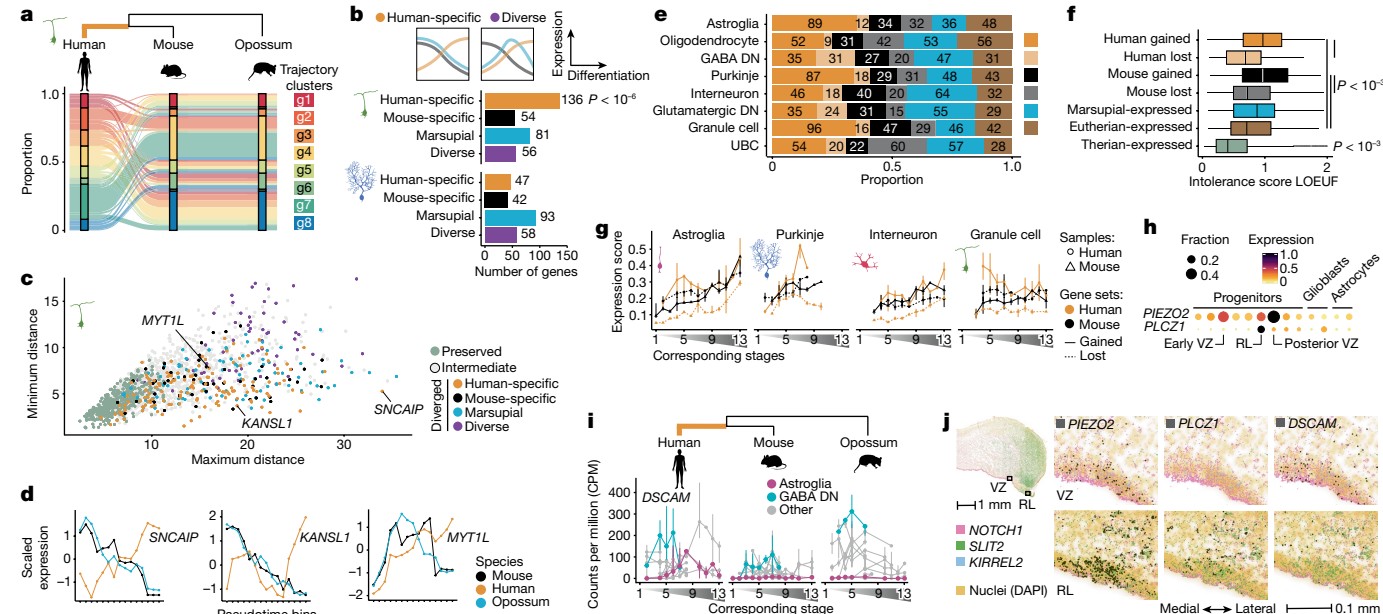

**Fig. 4 | Evolutionary change in gene expression. a**, Changes in gene-expression trajectories during granule cell differentiation assigned to the human lineage. **b**, Bottom, numbers of genes with trajectory changes in granule or Purkinje cells in different phylogenetic branches. Top, scheme illustrating a change in the human lineage and a diverse pattern. **c**, Minimum and maximum pairwise distances between the trajectories of orthologues from the three species. High maximum and low minimum distances indicate the strongest lineage-specific changes. **d**, Examples of genes that evolved a new trajectory during granule cell differentiation in the human lineage. **e**, Presence or absence expression differences assigned to different phylogenetic branches. **f**, Intolerance to functional mutations in human population (LOEUF) for genes grouped based on the presence or absence of expression. Values are summarized across the eight cell types. Boxes display interquartile range, whiskers extend to values within 1.5x interquartile range, and the line marks the median. Numbers of

genes as shown in **e** and Extended Data Fig. 11g. **g**, Expression of genes with gained or lost expression in the mouse or human lineage in selected cell types across development. Genes that were lost in a species were evaluated in the other species. **h,i**, Examples of genes that gained expression in human astroglial cells. In **h**, dot size and colour indicate the fraction of cells expressing a gene and the scaled mean expression level, respectively. **j**, Co-expression of *PIEZO2*, *PLCZ1* and *DSCAM* with *NOTCH1* (progenitors), *KIRREL2* (VZ) or *SLIT2* (rhombic lip) in human 12 wpc cerebellum by smFISH. The expanded regions are indicated by rectangles on the main section (top left). **g,i**, stages are aligned across species as in Fig. 1a; line indicates the median and bars the range across pseudobulks (*n* shown in Extended Data Fig. 11m). **b,f**, Adjusted *P* values were calculated via two-sided binomial (**b**) or permutation tests of pairwise comparisons (**f**).

development, as inferred from bulk RNA-sequencing data[19] (Extended Data Fig. 11h). Compared to the genes expressed in all species, genes that gained expression in the human or mouse lineage are under weaker functional constraint and have higher cell-type specificity, whereas the genes that lost expression show intermediate levels of constraint (Fig. 4f and Extended Data Fig. 11i). Although most presence or absence expression differences were called in a single cell type, expression gains often involve genes that were already expressed in other neural cell types in the cerebellum (Extended Data Fig. 11j–l), suggesting evolutionary repurposing of genes between the cell types. Functional enrichments among the genes with expression differences include sensory perception and myofilament for genes that gained expression in human oligodendrocytes or astroglia, respectively (FDR < 0.05, Supplementary Table 12). Assessment of the expression patterns of genes that gained or lost expression in the mouse or human lineage revealed that the aggregated expression levels of these genes overall increase during development (Fig. 4g and Extended Data Figs. 11m and 12a). Notable exceptions occur in human progenitors (astroglia) and granule cells, which express the genes that gained expression in the human lineage at high levels already at early developmental stages (Fig. 4g). Among the progenitor subtypes, the expression levels of genes gained in human astroglia are the highest in the RL and posterior VZ progenitors (Extended Data Fig. 12b). Fifteen of the 89 genes with gained expression in human astroglia are enriched in the latter progenitor populations (hypergeometric test, *P* < 0.01), including the mechanosensitive ion channel gene *PIEZO2* and the phospholipase gene

*PLCZ1*, which are expressed in human VZ and RL progenitors or only RL progenitors, respectively (Fig. 4h, Supplementary Table 14). We suggest that these gains of expression could have a role in the specification of the unique pool of basal progenitors identified in the developing human cerebellum[9].

We then examined whether genes associated with cerebellum-linked diseases show presence or absence expression differences between human and mouse, the most common model organism used in biomedical studies. In this analysis we additionally considered genes for which polarization using opossum data was not possible (Supplementary Information), and identified 1,392 genes (16.1% of 8,620) with expression differences between the two eutherian species (Extended Data Fig. 12c and Supplementary Table 13). Among these are 26 disease-associated genes. For instance, the autism and Down syndrome-associated gene *DSCAM* gained expression in human astroglia (Fig. 4i), and *FGF2*, which is implicated in pilocytic astrocytoma, is expressed in human but not mouse astroglia and oligodendrocytes (Extended Data Fig. 12d). To substantiate the detected presence or absence expression differences, we spatially mapped 26 of these genes in the 12 wpc human cerebellum, focussing on genes for which absence of expression in mouse is supported by public in situ hybridization data[15,16] (Supplementary Table 6). Visualization of smFISH signals and quantification of the expression levels in cells labelled based on integration with our snRNA-seq data confirmed the co-expression of 22 genes with the respective cell-type markers (Fig. 4j and Extended Data Fig. 12e,f). For instance, *PIEZO2*, *PLCZ1* and *DSCAM* were detected in *NOTCH1*-positive progenitors, and

*CPLX4* was detected in *PAX2*-marked interneurons. We further explored the available human immunohistochemistry data[45] to map the genes that are expressed in a cell-type-specific manner in the adult human but not mouse cerebellum. This confirmed that human mature granule cells express ZP2, a zona pellucida glycoprotein, and granule cell layer interneurons express CPLX4, a complexin that is known to function in synaptic vesicle exocytosis in retina[46] (Extended Data Fig. 12g). Based on adult bulk RNA-sequencing data from nine mammals[47] (including six primates), we inferred that *ZP2* expression in the adult cerebellum was acquired specifically in human in the past approximately 7 million years, after the human–chimpanzee split, in line with previous findings[48], and that the distinct *CPLX4* expression emerged in the lineage leading to the great apes (Extended Data Fig. 12g). Thus, by using orthogonal datasets, we validated a subset of the detected presence or absence expression differences. Together, our comparative molecular analyses revealed many candidate genes, whose expression changes may underlie phenotypic adaptations of the cerebellum during evolution, and disease genes for which functional characterization in a mouse model might not reflect all the disease manifestations in human.

## Discussion

In this study we used a comprehensive comparative approach to characterize the development of the cerebellum from the beginning of neurogenesis to adulthood, and its evolution across mammals. Based on our snRNA-seq atlases of around 400,000 cells from the mouse, human and opossum cerebellum, we established a consensus classification of the cellular diversity in the mammalian cerebellum and identified gene sets that underlie core ancestral transcriptional programmes of cell fate specification in the cerebellum. Although a few rare cell-type or subtype categories were not recovered in all studied species owing to technical limitations, our analyses revealed that the overall cellular architecture of the developing cerebellum is similar across therian mammals, consistent with the previously posited conservation of its developmental programme throughout amniotes[4,49]. Nevertheless, we observed significantly higher relative abundances of early fetal Purkinje cells in human, which may be linked with the expansion of neuronal progenitor pools in the human cerebellum[9]. Given that Purkinje cell signals regulate the transit amplification of granule cell progenitors[6,38], we suggest that higher numbers of Purkinje cells could augment the generation of granule cells and lead to the increase in cerebellar cell numbers required to match the expansion of the neocortex in the human lineage[3]. The increase in human Purkinje cell abundances is biased towards the early-born subtypes, which in the mouse bear similarities to the adult *Aldoc*-positive subtypes that are enriched in the posterior regions of cerebellar hemispheres. Purkinje cells in these regions project to the lateral (dentate) deep nuclei that in the human lineage expanded by selective increase in the numbers of the large-bodied subtype of glutamatergic neurons[8,50]. Thus, it is tempting to speculate that the biased expansion of the Purkinje cells and large-bodied glutamatergic neurons in the lateral nuclei coincided during the course of human evolution. Additionally, adaptations in these areas have been suggested to support cognitive functions in humans[51].

Evolutionary innovation in cellular programmes is expected to be driven by lineage- or species-specific differences in gene expression. Considering the apparent absence of new transcriptomically distinct cell types in the human cerebellum, we propose that the previously observed alterations in the anatomy of progenitor zones[9] may be attributed to gene-expression changes within the mammalian-shared cell types. Consistently, we identified a set of genes that are recruited to the transcriptomes of subpopulations of human progenitor cells in the cerebellar germinal zones, potentially underlying their human-specific characteristics[9]. Furthermore, we found presence or absence expression differences between the species for all neural cell types, and

detected shifts in the expression trajectories during Purkinje and granule cell differentiation. In most cerebellar cell types, the genes that gained or lost expression in the human and mouse lineages are more active at later developmental stages. This pattern is consistent with the progressively increasing molecular divergence of the cerebellum (and other organs) between species during development owing to overall decreasing purifying selection, which enables drift and facilitates adaptations driven by positive selection[19,52]. A limitation of our study is that we did not evaluate lineage-specific genes and isoforms, which additionally contribute to the transcriptome differences between the species. Moreover, further work is required to distinguish between adaptive changes driven by positive selection and changes resulting from genetic drift, and to assess the potential functional relevance of individual expression shifts in the context of interspecies phenotypic differences. Notably, shifts in gene expression can lead to profound phenotypic effects, as shown for *NEUROD1*[53] and *LHX9*[54], which contributed to the emergence of granule cells' transit amplification or the variation in cerebellar deep nuclei numbers in amniotes. Our extensive comparative map of the cellular and molecular diversity in the mammalian cerebellum can be further leveraged to advance a mechanistic understanding of brain development, disease[55] and evolution.

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

## Reporting summary

Further information on research design is available in the Nature Portfolio Reporting Summary linked to this article.

## Data availability

The datasets generated in the current study are available in the heiDATA repository, https://doi.org/10.11588/data/QDOC4E. Processed data can be interactively explored at https://apps.kaessmannlab.org/sc-cerebellum-transcriptome. Mouse and human processed data are also available as a CELLxGENE collection at https://cellxgene.cziscience.com/collections/72d37bc9-76cc-442d-9131-da0e273862db. Previously published cerebellum snRNA-seq datasets are available at https://singlecell.broadinstitute.org/single_cell/study/SCP795 (Kozareva et al.[7]), https://www.covid19cellatlas.org/aldinger20/ (Aldinger et al.[13]), and https://github.com/linnarsson-lab/developing-human-brain (Braun et al.[22]); and gnomAD LOEUF metrics[39] (v2.1.1) at https://gnomad.broadinstitute.org/downloads#v2-constraint.

## Code availability

Custom code is available at https://gitlab.com/kaessmannlab/mammalian-cerebellum.

**Acknowledgements** The authors thank C. Conrad, A. Fallahshahroudi, F. Lamanna, D. Kawauchi, T. Trefzer, T. Yamada-Saito, X. Yuan and members of the Kaessmann group for discussions; M. Langlotz, T. Brüning, K. Mößinger, E. Renner, M. Toronyay-Kasztner, T. Nath Varma, B. Crespo Lopez, A. Billepp, P. Grimm and T. Wedig for assistance; J. L. VandeBerg for providing archived opossum samples; and the Joint MRC/Wellcome (MR/R006237/1) Human Developmental Biology Resource, Maryland Brain Collection at the Maryland Psychiatric Research Center (NIH NeuroBioBank), Chinese Brain Bank Center, and Human Brain Tissue Bank at Semmelweis University for providing human samples. The human histology images were provided by the Joint MRC/Wellcome Trust (MR/R006237/1, MR/X008304/1 and 226202/Z/22/Z) Human Developmental Biology Resource (www.hdbr.org). We acknowledge the access and services provided by the Imaging Centre at the European Molecular Biology Laboratory (EMBL IC), generously supported by the Boehringer Ingelheim Foundation. Purchase of the NextSeq 550 instrument was supported by the Klaus Tschira Foundation. The computational cluster bwForCluster of the Heidelberg University Computational Center is supported by the state of Baden-Württemberg through bwHPC and the German Research Foundation (INST 35/1134-1 FUGG). M.P. was supported by a grant from the Hungarian Brain Research Program (2017-1.2.1-NKP-2017-00002). M.C.-M. was supported by the Francis Crick Institute, which receives its core funding from Cancer Research UK (FC011171), the UK Medical Research Council (FC011171), and the Wellcome Trust (FC011171). This project has received funding from the European Research Council (ERC) under the European Union's Horizon 2020 research and innovation programme (VerteBrain to H.K., grant agreement no. 101019268; BRAIN-MATCH to S.M.P., grant agreement no. 819894), and Seventh Framework Programme (FP7-2007-2013) (OntoTransEvol to H.K., grant agreement no. 615253).

**Author contributions** M. Sepp, K.L., S.M.P. and H.K. conceived and organized the study. M. Sepp, P.G., P.K., S.L. and M.P. collected samples. M. Sepp, K.L., F.M. and I.S. established snRNA-seq methods. M. Sepp performed snRNA-seq experiments with support from N.M., C.S. and J.S. M. Sepp prepared the smFISH slides. K.L. performed snRNA-seq and smFISH data processing. K.L. and M. Sepp analysed the data with contributions from I.S. and E.L., and input from F.M. and N.T. M. Sepp, M. Schauer and P.G. performed histology. L.S. and L.M.K. performed immunohistochemistry. K.L. and N.T. developed the web application. K.O. and P.J. provided critical discussions. L.M.K., S.A. and M.C.-M. provided key scientific advice. I.S. supervised the comparative analyses. S.M.P. and H.K. oversaw the study and provided funding. M. Sepp and K.L. drafted the manuscript, with critical review by I.S., M.C.-M., S.M.P. and H.K. All authors provided feedback on drafts and approved its final version.

**Competing interests** The authors declare no competing interests.

**Additional information**
**Correspondence and requests for materials** should be addressed to Mari Sepp, Kevin Leiss, Ioannis Sarropoulos, Stefan M. Pfister or Henrik Kaessmann.

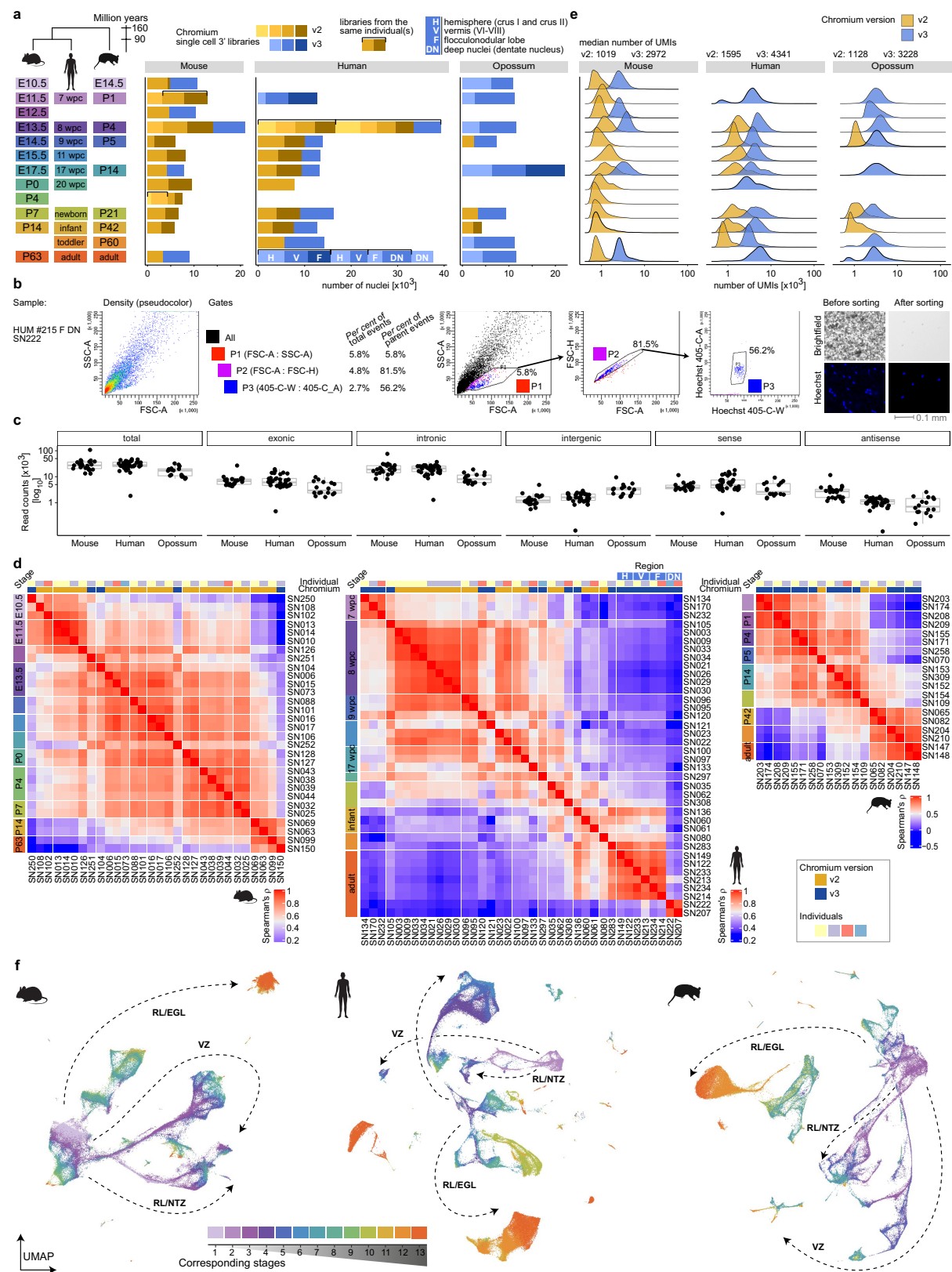

**Extended Data Fig. 1** | See next page for caption.

**Extended Data Fig. 1 | Overview of the datasets. a**, Number of cells profiled by snRNA-seq per stage in mouse, human and opossum. A schematic of the sampled stages is shown on the left. Colours in bar plots indicate individual libraries (Chromium v2 in yellow hues and v3 in blue hues). Samples from different individuals were used for each library, except for the libraries grouped with brackets. The sampled cerebellum region is indicated for human adult libraries. **b**, Example of gating strategy used for fluorescence-activated nuclei sorting. Nuclei were stained with Hoechst. To separate nuclei from the cellular debris the gates were set on FSC/SSC and at the excitation wavelength of 405 nm. Cell counter images before and after sorting are shown at the right. Sorting was applied to some of the adult human samples (Supplementary Table 1). **c**, Mapping statistics for the reads from mouse, human and opossum libraries. Shown are the total, exonic, intronic, and intergenic read counts per library based on the mature mRNA reference. Exonic counts are further split into sense and antisense read counts. Boxes represent the interquartile range, whiskers extend to extreme values within 1.5 times the interquartile range from the box, and line denotes the median. n(mouse) = 30, n(human) = 38, n(opossum) = 19. **d**, Spearman's rho correlation coefficients across libraries in mouse, human and opossum datasets. UMIs counted in mature mRNA mode were aggregated across all cells in each library. Correlations were calculated using the ranks of CPM (counts per million) values of the expressed genes (genes expressed in at least 10% of cells in any pseudobulk; human n = 7,696; mouse n = 4,806; opossum n = 2,765) within each library. Libraries are grouped by developmental stage and Chromium version. The sampled cerebellum region is indicated for human adult libraries, as in **a**. **e**, Distribution of UMI counts in v2 and v3 libraries in mouse, human and opossum datasets. Median UMI counts are shown for each species and Chromium version. **f**, Uniform Manifold Approximation and Projection (UMAP) of 115,282 mouse, 180,956 human and 99,498 opossum cells coloured by their developmental stage. Colours indicate the matched stages as shown in panel **a** and Fig. 1a. The broad neuronal lineages are shown with arrows. EGL, external granule cell layer; NTZ, nuclear transitory zone; RL, rhombic lip; VZ, ventricular zone.

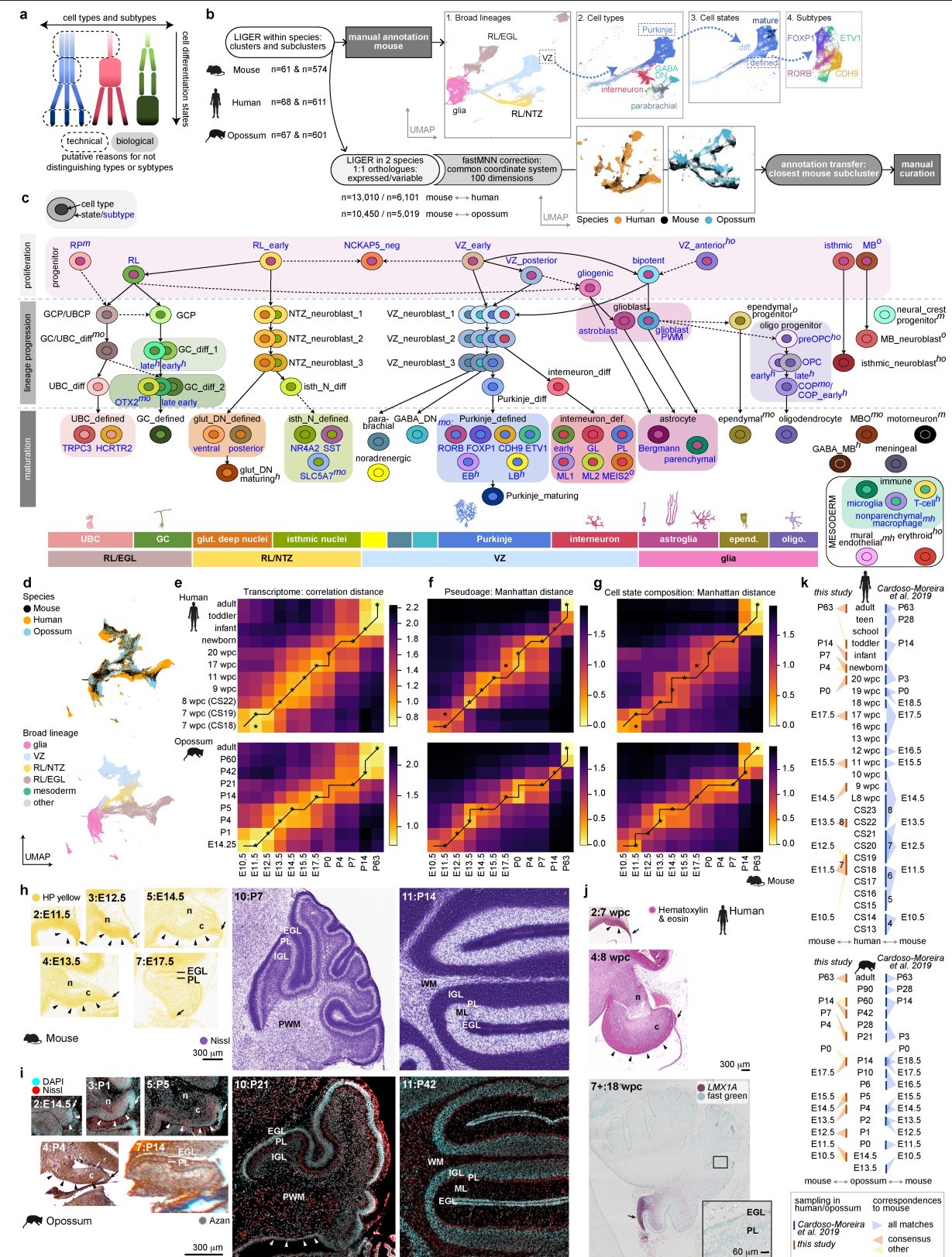

**Extended Data Fig. 2** | See next page for caption.

**Extended Data Fig. 2 | Cell type annotation and stage correspondences.**
**a**, Schematic summary of the annotation strategy. We used the term "type" to group cells committed to a distinct cell fate, and "state" to refer to differentiation status that often form a continuum within each cell type category. Shown are three hypothetical cell types (colours) and their subtypes (rectangles). Note that the cell state categories do not necessarily align across cell types. Both biological and technical reasons could explain why subtypes cannot be distinguished across all states in a given cell type. **b**, Outline of the procedures used for cell type annotation of the mouse, human and opossum datasets. **c**, Overview of the cell type annotation categories. Schematic cells have their nuclei coloured by cell type and cytoplasm coloured by state or subtype. Cell state labels are in black, subtype labels in blue. Cell states at which subtypes were distinguished are highlighted with coloured background. For the categories not detected in all species, superscript text specifies the dataset(s) where a category is present: *h*, human; *m*, mouse; *o*, opossum. Solid arrows depict known lineage relationships, dashed arrows depict additional relationships suggested by our data. Broad cell type lineage groups are shown at the bottom. **d**, Integrated UMAP of mouse, human and opossum cells coloured by species or broad cell type lineage. We used 1:1 orthologous genes detectable in all batches and variable across cells (n = 3,742). **e**–**g**, Pairwise correspondences of developmental stages across species. The line indicates the best alignment between the time series determined by dynamic time warping algorithm using pseudobulk transcriptome correlation distances (**e**), Manhattan distances of pseudoages (**f**), and Manhattan distances of the cellular compositions at the level of cell states (**g**). Mouse was used as the focal species. Asterisks indicate the consensus stage correspondences from the three analyses. In **e**, we only used the genes that were informative in both datasets (intersect of overdispersed genes, human vs. mouse n = 336, opossum vs. mouse n = 369). **h**–**j**, Comparison of the developing cerebellum structures in mouse (**h**), opossum (**i**) and human (**j**). Mouse images are from the Allen Developing Mouse Brain Atlas[15]. The sagittal sections were stained with HP yellow or Nissl. For opossum, sagittal sections were prepared from E14.5-P21 heads and P42 cerebellum. Sections from fresh-frozen or FFPE samples were stained with DAPI and NeuroTrace Nissl, or with Azan, respectively. Human images are from the HDBR Atlas[79–81]. 7 wpc and 8 wpc (CS23) sagittal sections were stained with hematoxylin and eosin; *LMX1*A RNA was probed in the 18 wpc sagittal section, counterstained with fast green. Arrowheads indicate the cerebellar ventricular zone, arrows denote the rhombic lip, n and c label the NTZ and CTZ. The stages are numbered as in Fig. 1a. At E11.5/E14.5 (mouse/opossum) the cerebellar primordium is dominated by the cell-dense neuroepithelium. NTZ and CTZ are first visible at E12.5/P1 and E13.5/P4, respectively. EGL and developing PL are discerned at E17.5/P14. P7/P21 is characterized by a thick EGL, which shrinks but is still present at P14/P42. Similarly in human, at 7 wpc the cerebellar primordium is dominated by the cell-dense neuroepithelium; CTZ is visible at 8 wpc; EGL and PL are discerned at 18 wpc. Newborns are characterized by a thick EGL, which gradually shrinks but is still present in infants after 8 months of postnatal development[103]. **k**, Comparison of the sampling and stage correspondences in this study and in ref. 19. Human samples representing 4–8 wpc may include samples from several Carnegie stages. The correspondences estimated in both studies globally agree. The shifts are explained by differences in sampling, e.g. in this study 8 wpc in human is represented by CS22 and best matches to E13.5 in mouse, whereas in ref. 19 8 wpc stage group includes samples from CS22 to late 8 week and best matches to E14.5 in mouse. COP, committed oligodendrocyte precursor; def., defined; diff, differentiating; CS, Carnegie stage; CTZ (c), cortical transitory zone; DN, deep nuclei; E, embryonic/prenatal day; EB, early-born; EGL, external granule cell layer; GABA, GABAergic; GC, granule cell; GCP, granule cell progenitor; glut, glutamatergic; IGL, internal granule cell layer; isth N, isthmic nuclei neurons; L8, late 8th week; LB, late-born; MB, midbrain; MBO, midbrain-originating cell; ML, molecular layer; NTZ (n), nuclear transitory zone; oligo, oligodendrocyte; OPC, oligodendrocyte progenitor cell; preOPC, precursor of oligodendrocyte progenitor cell; P, postnatal day; PL, Purkinje cell layer; PWM, prospective white matter; RL, rhombic lip; RP, roof plate; UBC, unipolar brush cell; UBCP, unipolar brush cell progenitor; VZ, ventricular zone; WM, white matter; wpc, weeks post conception.

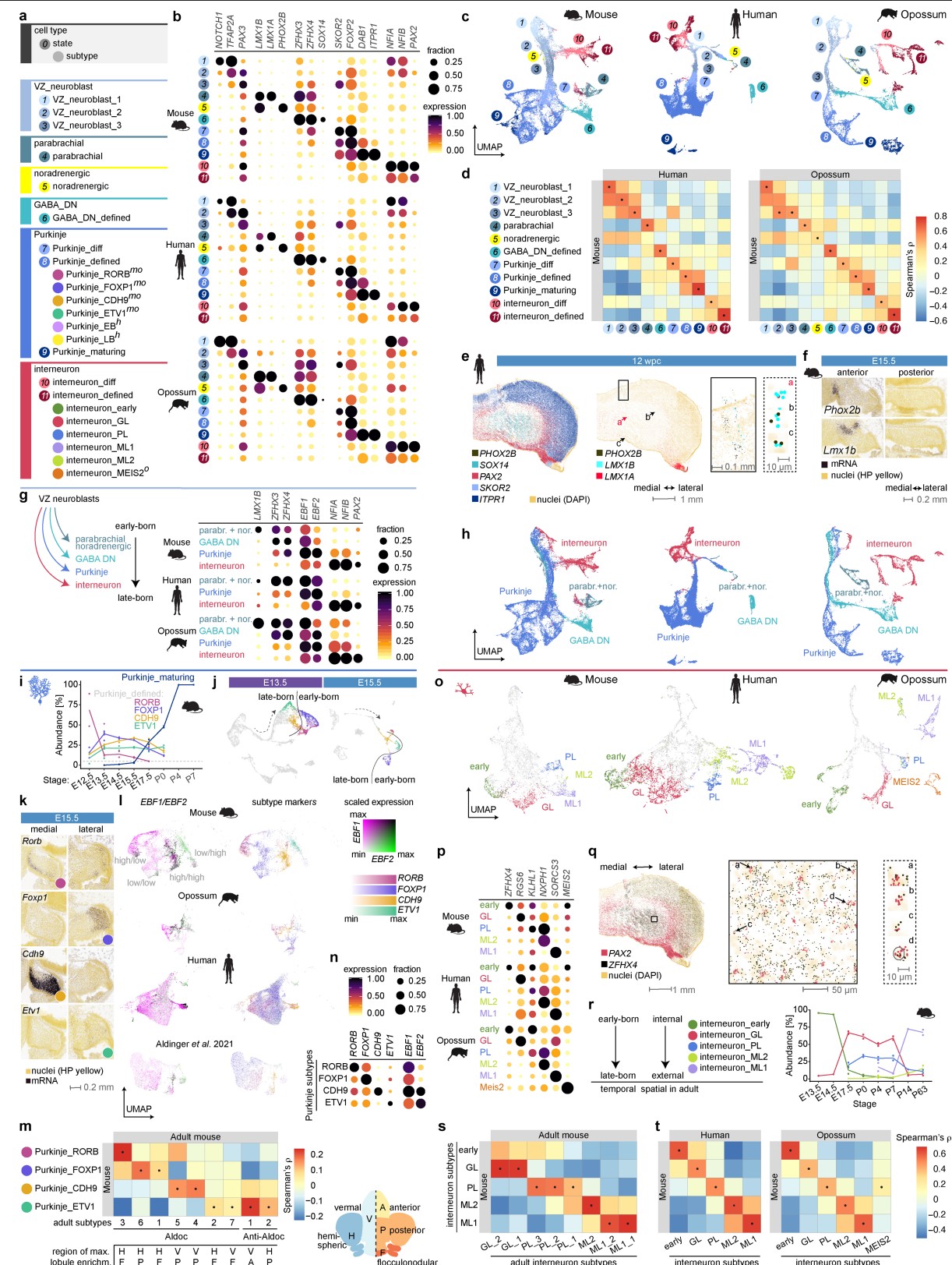

**Extended Data Fig. 3 | Atlas of the VZ cell types. a**, Cell types, states and subtypes of neurons born at the ventricular zone. For the categories not detected in all species, superscript text specifies the dataset(s) where a category is present: *h*, human; *m*, mouse; *o*, opossum. **b**, Expression of key marker genes in the VZ-associated cell states in mouse, human and opossum. **c**, Uniform Manifold Approximation and Projection (UMAP) of 37,391 mouse, 61,585 human and 22,674 opossum VZ-derived cells coloured by their state. Colours and numbers as in **a**. **d**, Spearman's correlation coefficients between orthologous variable gene (n = 208) expression profiles from mouse, human and opossum VZ-associated cell states. **e**, Human 12 wpc cerebellum smFISH data for markers of the VZ cell types. Expression of marker genes of GABAergic deep nuclei neurons (*SOX14*), Purkinje cells (*SKOR2, ITPR1*) and interneurons (*PAX2*) is detected in expected domains (left). Only a few cells outside the rhombic lip and a region with artifactual signals (solid line box) are co-labelled by the markers of the parabrachial (*LMX1A, LMX1B;* red arrow) and noradrenergic (*PHOX2B, LMX1B;* black arrows) cell types in this section (right), which originates from the posterior cerebellum. This is in line with the expression of the parabrachial and noradrenergic cell markers in the anterior cerebellum in mouse, as shown in **f**. **f**, Spatial distribution of parabrachial and noradrenergic cell types in mouse E15.5 cerebellar primordium based on RNA in situ hybridization data[15] for marker genes. Anterior and posterior coronal sections are shown. **g**, Expression of key marker genes in the VZ neuroblasts split into lineages as in **h** in mouse, human and opossum. **h**, UMAPs as in **c** coloured by cell type lineages. VZ neuroblasts were split into lineages giving rise to the different mature cell types based on the information about their developmental stage and marker gene expression. **i**, Relative abundances of cells in the defined and maturing Purkinje cell categories (median of biological replicates) across developmental stages in mouse. The dashed line marks 5%. **j**, UMAPs of all mouse E13.5 and E15.5 cells, Purkinje subtypes are highlighted with colours. The dashed arrow directs from less mature cells (VZ neuroblasts) to more mature cells (defined Purkinje cells). The line separates early- and late-born Purkinje subtypes. **k**, Spatial distribution of Purkinje subtypes in E15.5 mouse cerebellar primordium based on RNA in situ hybridization data[15] for subtype marker genes. Medial and lateral sagittal sections are shown. **l**, UMAPs showing expression of key marker genes in the subtype-assigned Purkinje cells in our mouse, opossum, and human datasets, and in the reanalysed ref. 13 dataset. Scaled expression of *EBF1* and *EBF2* is shown at the left to highlight the combinatorial patterns; scaled expression of subtype markers *RORB, FOXP1, CDH9* and *ETV1* is shown at the right with each cell coloured according to the gene that has the highest scaled expression level. For visualization purposes, the scales were capped at 95th quantile for *RORB, FOXP1, CDH9, EBF1* and *EBF2*, and 99th quantile for *ETV1*. **m**, Spearman's correlation coefficients between shared variable gene (n = 337) expression profiles from mouse Purkinje subtypes from this study and adult subtypes described in ref. 7. For each adult subtype the position of the lobule showing the highest enrichment[7] along the mediolateral and anteroposterior axes is indicated. **n**, Dot plot showing expression of key marker genes in the Purkinje subtypes in the reanalysed ref. 13 dataset. **o**, UMAPs of 6,422 mouse, 7,640 human and 5,815 opossum GABAergic interneurons coloured by their subtype. Subtype colours as in **a**; neuroblasts and differentiating interneurons are in grey. **p**, Expression of key marker genes in the interneuron subtypes in mouse, human and opossum. **q**, Human 12 wpc cerebellum smFISH data for markers of the interneuron "early" subtype. Cells co-expressing *PAX2* and *ZFHX4* are detected in the region of the nuclear transitory zone. **r**, Interneuron subtype relative abundances (median of biological replicates) across developmental stages in mouse. The temporal order of interneuron subtype emergence gives rise to the spatial order in the adult cerebellum. **s**, Spearman's correlation coefficients between shared variable gene (n = 329) expression profiles from mouse interneuron subtypes from this study and adult subtypes described in ref. 7. **t**, Spearman's correlation coefficients between orthologous variable gene (n = 198) expression profiles from mouse, human and opossum interneuron subtypes. In **b**,**g**,**n** and **p**, dot size and colour indicate the fraction of cells expressing each gene and the mean expression level scaled per species and gene, respectively. In **d**,**m**,**s** and **t**, dots indicate the highest correlation for each column. diff, differentiating; EB, early-born; GABA DN, GABAergic deep nuclei neurons; GL, granule cell layer; LB, late-born; ML, molecular layer; parabr.+nor., parabrachial and noradrenergic cells; PL, Purkinje cell layer.

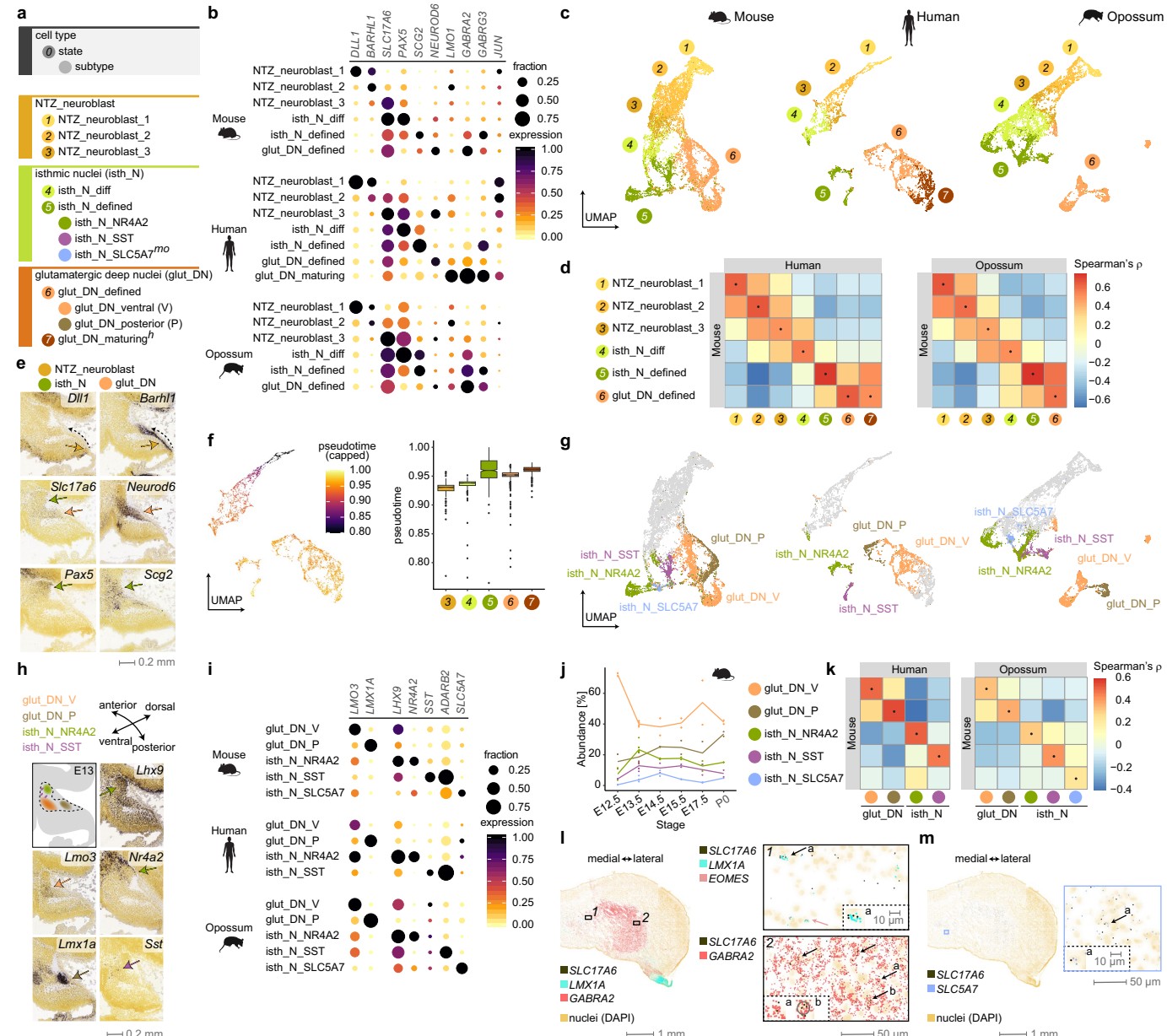

**Extended Data Fig. 4 | Atlas of the RL/NTZ cell types. a**, Cell types, states and subtypes of neurons born at the early rhombic lip and/or located at the nuclear transitory zone during development. For the categories not detected in all species, superscript text specifies the dataset(s) where a category is present: *h*, human; *m*, mouse; *o*, opossum. In human, we distinguished a *LMO1*-marked population of glutamatergic deep nuclei neurons that likely represents a more mature cell state (see b-f and l). **b, i**, Expression of key marker genes in the RL/NTZ cell states (**b**) or subtypes (**i**) in mouse, human and opossum. Dot size and colour indicate the fraction of cells expressing each gene and the mean expression level scaled per species and gene, respectively. **c, g**, Uniform Manifold Approximation and Projection (UMAP) of 10,949 mouse, 6,301 human and 9,965 opossum RL/NTZ cells coloured by their state (**b**) or subtype (**g**). Colours and numbers as in **a**. **d, k**, Spearman's correlation coefficients between orthologous variable gene expression profiles from mouse, human and opossum cell states (**d**; n = 224 genes) or subtypes (**k**; n = 225 genes) in the RL/NTZ broad lineage. Dots indicate the highest correlation for each column. **e, h**, Spatial distribution of RL/NTZ cell states (**e**) or glutamatergic deep nuclei and isthmic nuclei subtypes (**h**) in mouse E13.5 cerebellar primordium based on RNA in situ hybridization data[15] for marker genes. Sagittal sections counterstained with HP Yellow are shown. Coloured arrows indicate the domains expressing markers of the different cell type/state

categories; dotted arrows show the direction of the migration from the rhombic lip to the NTZ. In **h**, a schematic summary is shown in the top left panel. **f**, UMAP of human RL/NTZ cells coloured by their pseudotime values, and distribution of pseudotime values across cell state categories. Colours and numbers as in **a**. Boxes represent the interquartile range, whiskers extend to extreme values within 1.5 times the interquartile range from the box, and line denotes the median. **j**, Subtype relative abundances (median of biological replicates) across developmental stages in mouse. **l**, Human 12 wpc cerebellum smFISH data for markers of the glutamatergic deep nuclei. The locations of the regions expanded at the right are shown with rectangles on the whole section at the left. Black arrows indicate *SLC17A6*-positive glutamatergic deep nuclei neurons. Pink arrow indicates *EOMES*-positive unipolar brush cell. Insets (dashed line) show close-ups of individual cells. *LMX1A*, a marker of *glut_DN_P* is detected in a minority of glutamatergic deep nuclei neurons (1); glutamatergic deep nuclei neurons expressing *GABRA2*, enriched in *glut_DN_maturing* cells, dominate the NTZ at 12 wpc (2). **m**, Detection of cells co-expressing *SLC17A6* and *SLC5A7* in the human 12 wpc cerebellum by smFISH. One multiplexed smFISH experiment was performed. glut DN, glutamatergic deep nuclei neurons; isth N, isthmic nuclei neurons; P, posterior; V, ventral.

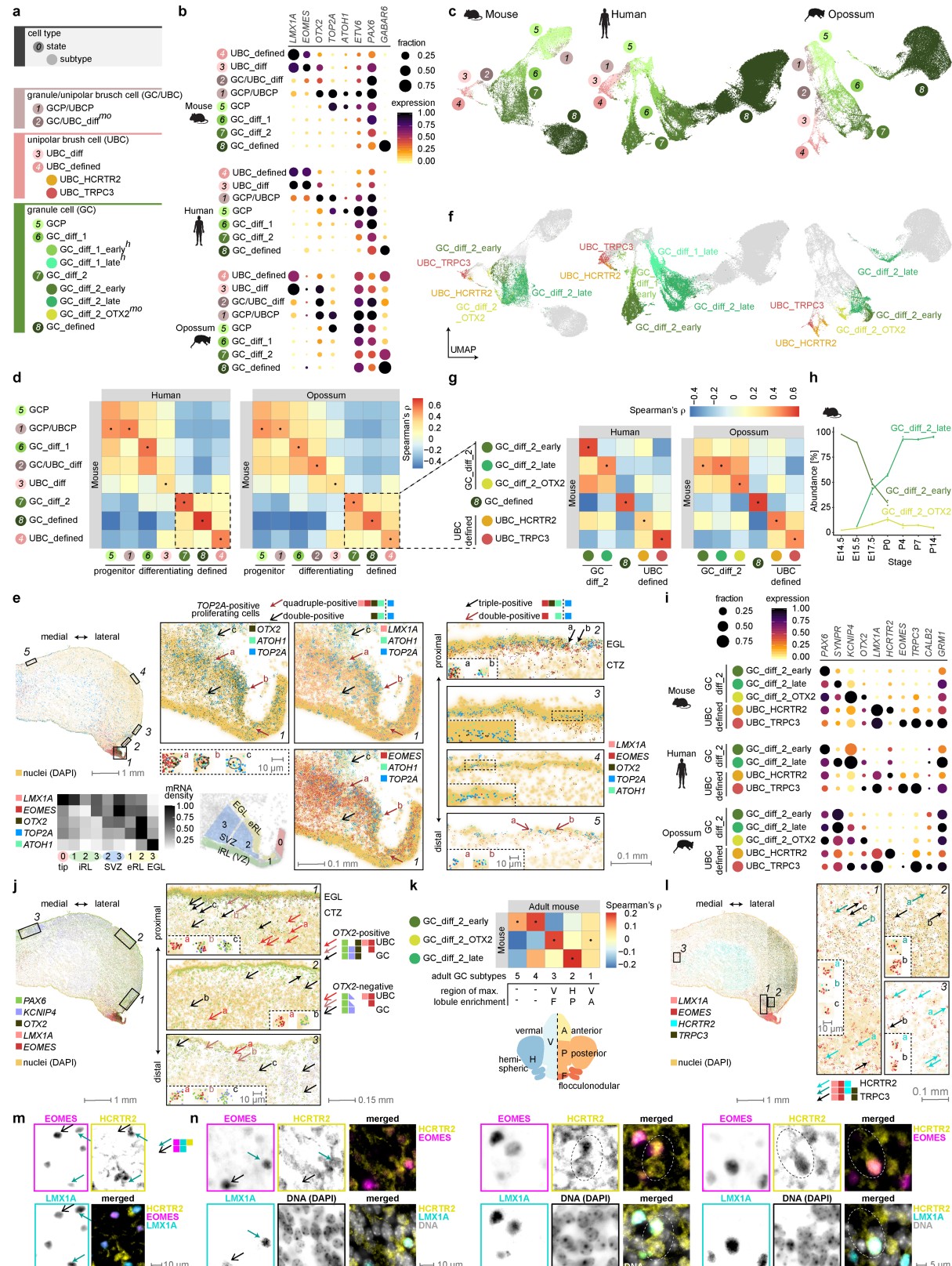

**Extended Data Fig. 5** | See next page for caption.

**Extended Data Fig. 5 | Atlas of the RL/EGL cell types. a**, Cell types, states and subtypes of neurons born at the late rhombic lip associated with the external granule cell layer. For the categories not detected in all species, superscript text specifies the dataset(s) where a category is present: *h*, human; *m*, mouse; *o*, opossum. **b,i**, Expression of key marker genes in the granule and unipolar brush cell states (**b**) and subtype (**i**) in mouse, human and opossum. Dot size and colour indicate the fraction of cells expressing each gene and the mean expression level scaled per species and gene, respectively. **c,f**, Uniform Manifold Approximation and Projection (UMAP) of 32,767 mouse, 73,492 human and 36,585 opossum RL/EGL cells coloured by their state (**c**) or subtype (**f**). Colours and numbers as in **a**. **d,g**, Spearman's correlation coefficients between orthologous variable gene expression profiles from mouse, human and opossum cell states (**e**; n = 110 genes) or subtypes (**f**; n = 101 genes) in the RL/EGL broad lineage. Dots indicate the highest correlation for each column. **e,j,l**, Human 12 wpc cerebellum smFISH data for markers of GC and UBC states (**e**), GC subtypes (**j**), and UBC subtypes (**l**). The locations of the regions expanded at right are shown with rectangles (solid line) on the whole section at left. Arrows indicate cells with specific expression patterns as described in the legends. Insets (dashed line) show close-ups of individual cells. In **e**, the heatmap shows the scaled density of mRNA spots in different rhombic lip compartments, as proposed in mice[104] and in humans[9]. One multiplexed smFISH experiment was performed. **h**, Relative abundances (median of biological replicates) of differentiating granule cell subtypes across developmental stages in mouse. **k**, Spearman's correlation coefficients between shared variable gene (n = 98) expression profiles from mouse differentiating granule cell subtypes from this study and adult subtypes described in ref. 7. For each adult subtype the position of the lobule showing the highest enrichment[7] along the mediolateral and anteroposterior axes is shown. Dots indicate the highest correlation for each column. **m,n**, Detection of HCRTR2 in unipolar brush cells by immunohistochemistry. The HCRTR2, EOMES and LMX1A were detected by indirect immunofluorescence (**m**) or Immuno-SABER (**n**). The HCRTR2 antibodies used for immunohistochemistry were MAB52461 (**m**) and AOR-002 (**n**). Arrows point to HCRTR2-positive and -negative UBCs, as specified in the legend. Dotted circles highlight HCRTR2-positive cells with brush morphology. The fields shown are from the lobule X granule cell layer of P7 mouse. CTZ, cortical transitory zone; diff, differentiating; EGL, external granule cell layer; eRL, external rhombic lip; GC, granule cell; GCP, granule cell progenitor; iRL, internal rhombic lip; SVZ, subventricular zone; UBC, unipolar brush cell; UBCP, unipolar brush cell progenitor; VZ, ventricular zone.

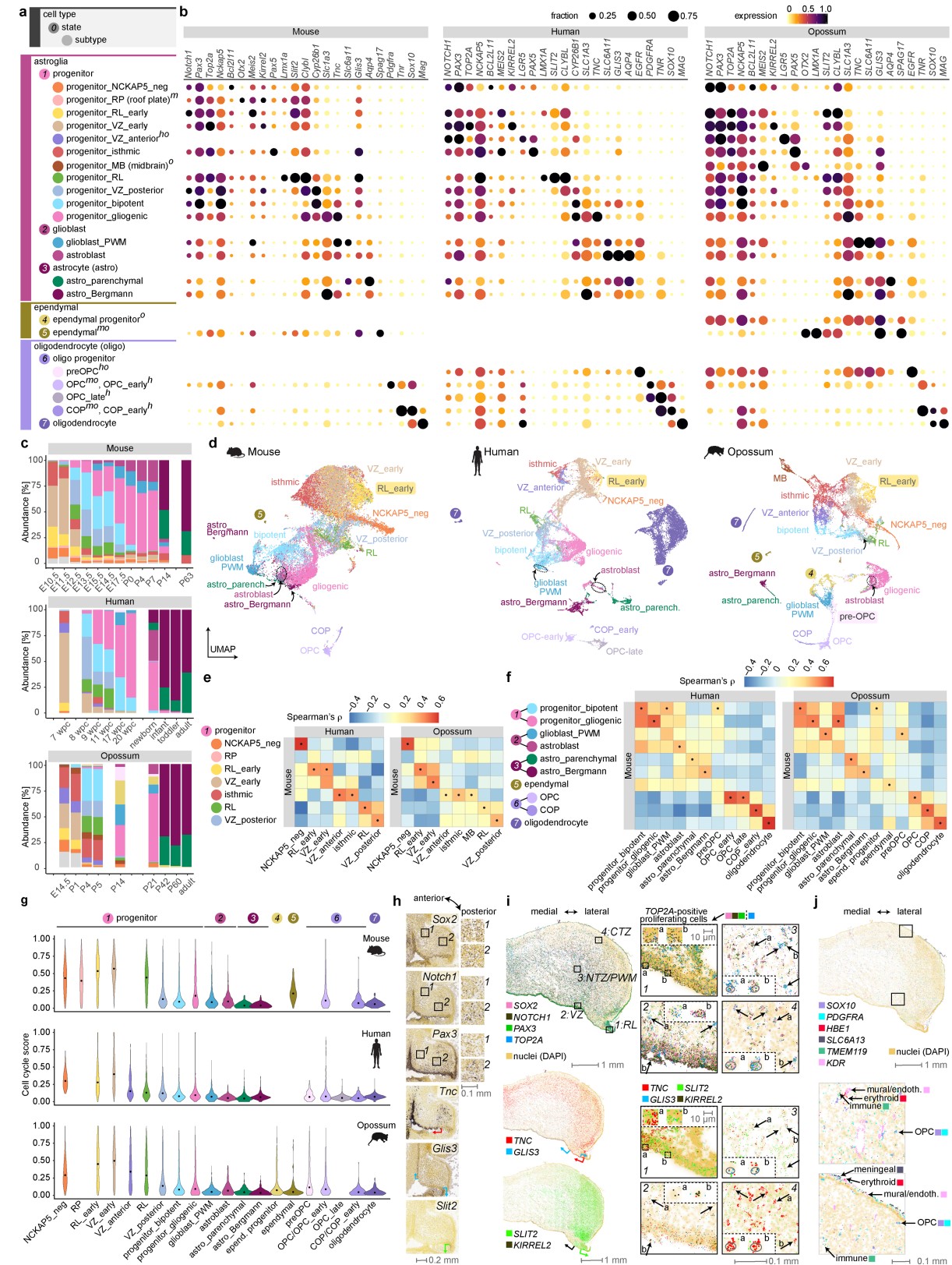

**Extended Data Fig. 6** | See next page for caption.

**Extended Data Fig. 6 | Atlas of the glial cell types. a**, Cell types, states and subtypes of glial cells, including neural progenitor cells. For the categories not detected in all species, superscript text specifies the dataset(s) where a category is present: *h*, human; *m*, mouse; *o*, opossum. **b**, Expression of key marker genes in glial cell states and subtypes in mouse, human and opossum. Dot size and colour indicate the fraction of cells expressing each gene and the mean expression level scaled per species and gene, respectively. **c**, Relative abundances of astroglia subtypes, ependymal progenitors and preOPCs across developmental stages. Colours are as in **a**; astroglial cells not assigned to a subtype are in grey. Stages are aligned as in Fig. 1a. Human adult samples dissected from the deep nuclei region were excluded. **d**, Uniform Manifold Approximation and Projection (UMAP) of 28,486 mouse, 32,897 human and 20,742 opossum glial cells coloured by their subtype or state. Colours and numbers as in **a**. Progenitors not assigned to a subtype are in grey. Mouse roof plate progenitors and human preOPCs are low in numbers and not discernible in this UMAP. Inclusion of human adult samples dissected from the deep nuclei region explains the high numbers of oligodendrocytes in the human UMAP. **e,f**, Spearman's correlation coefficients between orthologous variable gene expression profiles from mouse, human and opossum early progenitors (**e**; n = 92 genes) or late progenitors and other glial cells (**f**; n = 129 genes). Dots indicate the highest correlation for each column. **g**, Distribution of cell cycle score values across glial categories in mouse, human and opossum. Points indicate median score value. **h**, Spatial distribution of astroglia lineage cells in mouse E15.5 cerebellar primordium based on RNA in situ hybridization data[15] for marker genes. Sagittal sections counterstained with HP Yellow are shown.

The regions in nuclear transitory zone (1) and cortical transitory zone (2) shown with rectangles at left are expanded at right, and highlight marker expression outside the VZ and RL. Coloured arrows indicate expression domains along the ventricular zone (including the VZ of the rhombic lip). **i**, Human 12 wpc cerebellum smFISH data for markers of astroglia lineage. The locations of the regions expanded at the right are shown with rectangles (solid line) on the whole section at the left. Coloured arrows indicate expression domains along the ventricular zone (including the VZ of the rhombic lip). Black arrows indicate proliferative progenitor cells. Insets (dashed line) show close-ups of regions or individual cells. The same regions and cells are shown in top and bottom panels. *SLIT2*, a marker of RL progenitors is expressed in the the rhombic lip VZ (1); progenitors expressing *KIRREL2*, a marker of VZ progenitors, are present in the VZ and adjacent subventricular zone (2); progenitors expressing *GLIS3*, enriched in bipotent progenitors, are detected in the PWM/NTZ (3), progenitors in the forming Purkinje cell layer in CTZ express *TNC*, a marker of gliogenic progenitors (4). **j**, Human 12 wpc cerebellum smFISH data for markers of oligodendrocytes and mesodermal cell types. The locations of the regions expanded at the bottom are shown with rectangles (solid line) on the whole section at the top. Black arrows indicate example cells from different cell types. One multiplexed smFISH experiment was performed. astro, astrocyte; COP, committed oligodendrocyte precursor; CTZ, cortical transitory zone; endoth., endothelial; MB, midbrain; NTZ, nuclear transitory zone; OPC, oligodendrocyte progenitor cell; preOPC, precursor of oligodendrocyte progenitor; PWM, prospective white matter; RL, rhombic lip; RP, roof plate; VZ, ventricular zone.

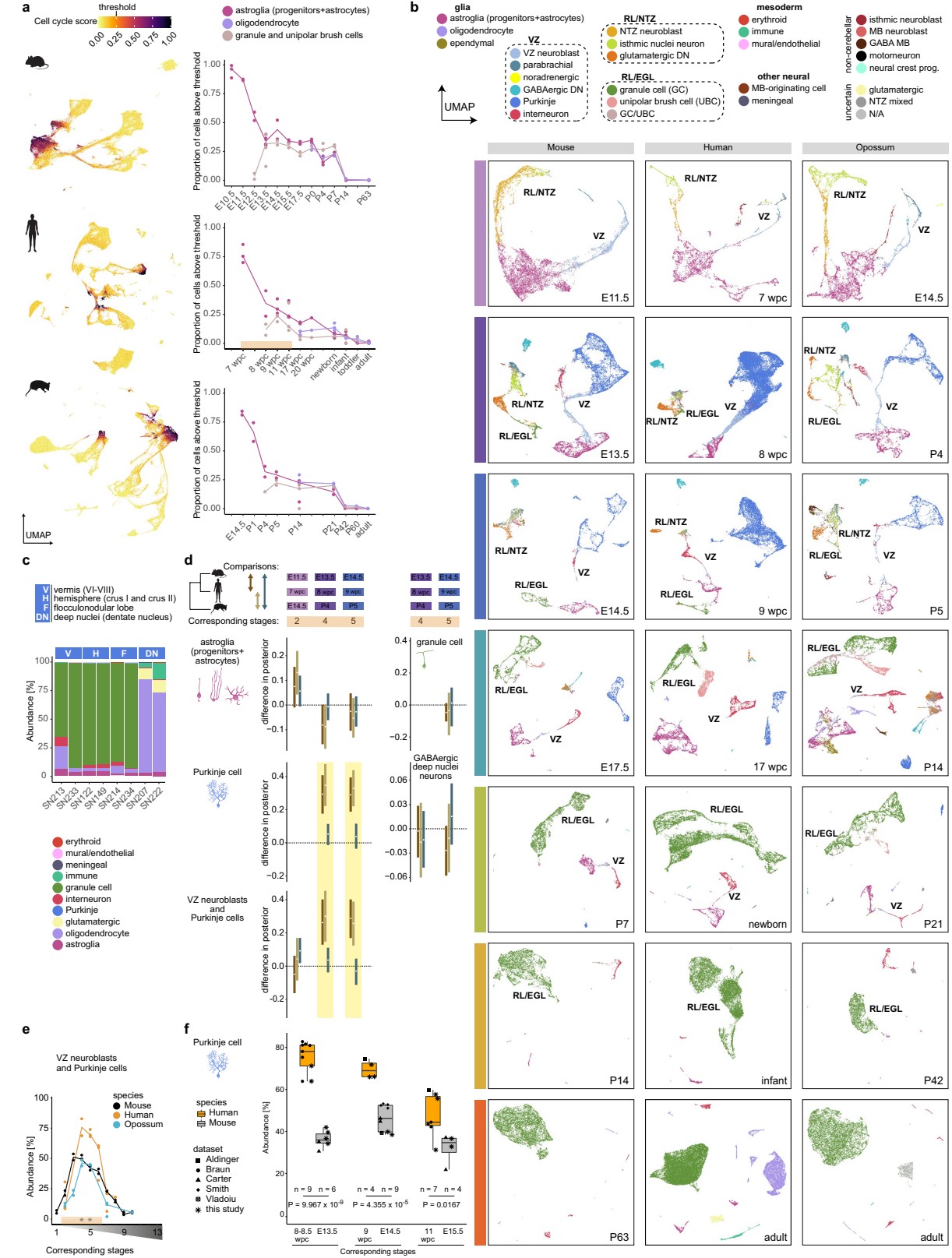

**Extended Data Fig. 7** | See next page for caption.

**Extended Data Fig. 7 | Dynamics of cell type abundances across development.**
**a**, UMAPs of mouse, human and opossum cells coloured by cell cycle score (left) and the proportion of cells (median across biological replicates) above the threshold score value (0.25) among astroglia, oligodendrocytes, and granule and unipolar brush cells (right). The stages are aligned as shown in Fig. 1a. Orange shading marks stages with representative sampling in human. **b**, Individual developmental stage UMAPs of mouse, human and opossum cells coloured by their cell type. Only the stages with correspondences in all studied species are shown. Labels indicate the broad neuronal lineages. **c**, Relative cell type abundances in individual adult human samples from different regions of the cerebellum. **d**, Hierarchical Bayes model analysis of differences in the relative cell type abundances across species at corresponding developmental stages. Difference in posterior (y-axes) shows modelled proportion differences between pairs of species (comparisons); 0 indicates no shift in proportions (dotted line). The modelled differences are summarized as 95% highest density intervals ($HDI_{95}$; lines) for each cell type at developmental stages (x-axes; depicted on top) where at least 50 cells were present. Only corresponding stages with representative sampling in human were considered. Differences in the relative abundances were called (yellow shading) when $HDI_{95}$ of at least two comparisons did not overlap 0 (e.g., a human-specific change is assumed when $HDI_{95}$ of human versus mouse and human versus opossum comparisons does not overlap 0, and $HDI_{95}$ of mouse versus opossum comparison overlaps 0). **e**, Relative abundances of cells annotated as Purkinje cells or VZ neuroblasts across developmental stages. VZ neuroblasts and Purkinje cells were analysed together to exclude the effect of possible biases in the annotation between the three species. Stages are aligned as in Fig. 1a, the line indicates the median of biological replicates, orange shading marks stages with representative sampling in human, and asterisks indicate differences in the relative abundances in human compared to mouse and opossum. **f**, Purkinje cell relative abundances in available human and mouse cerebellum datasets[10,11,13,21,22]. The estimation of abundances is based on the annotations reported in the original studies, except for ref. 22, where cell type annotations were not provided in the original study and were instead transferred from our human dataset (Supplementary Information). *P* values from Welch tests. DN, deep nuclei; EGL, external granule cell layer; MB, midbrain; NTZ, nuclear transitory zone; RL, rhombic lip; UBC, unipolar brush cells; UMAP, Uniform Manifold Approximation and Projection; VZ, ventricular zone; wpc, weeks post conception.

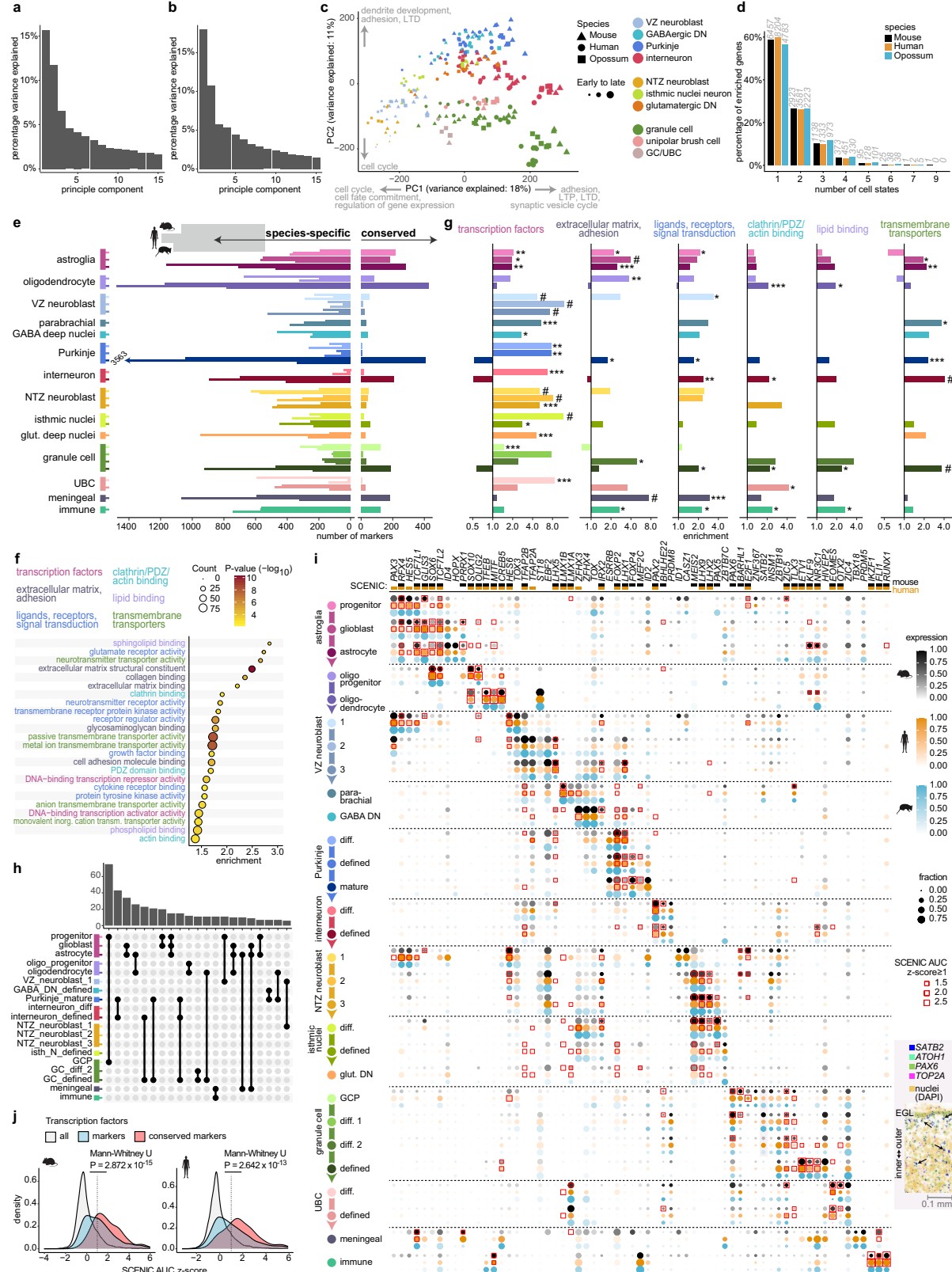

**Extended Data Fig. 8** | See next page for caption.

**Extended Data Fig. 8 | Transcriptional programs and marker genes.**
**a**,**b**, The percentage of variance explained by the first 15 principal components in the global PCA (**a**) and in the PCA of neurons only (**b**). **c**, PCA of neuronal cells based on 10,276 expressed orthologous genes across the three species. Data points represent cell-type pseudobulks for each biological replicate. Examples of enriched gene ontology and pathway categories for the genes loaded to PC1 and PC2 are indicated. **d**, Number of cell states in which genes show expression enrichment in mouse, human and opossum. **e**, Numbers of species-specific and conserved cell state markers among 1:1:1 orthologous genes. **f**, Enriched gene ontology molecular function categories among the conserved markers. Terms are grouped into broad categories as indicated by the colours. **g**, Representation of the broad molecular function categories among the conserved markers of individual cell states. Enrichments were identified using one-sided hypergeometric tests against a background of all 1:1:1 orthologous genes detected in the cell states included in the analysis; *P* values were adjusted for multiple testing using Benjamini-Hochberg method; *$P < 0.05$, **$P < 10^{-2}$, ***$P < 10^{-3}$, #$P < 10^{-6}$. **h**, Shared conserved markers are typically found in states closely related in terms of cell type lineage or maturation status. All groups with more than 5 genes are shown. **i**, Expression and regulon activities of transcription factors that are among the conserved markers, across cell states in mouse (black), human (orange) and opossum (blue). Dot size and colour intensity indicate the fraction of cells expressing each gene and the mean expression level scaled per species and gene, respectively. Colours on top mark the transcription factors for which regulons (i.e. co-expression modules that are retained after pruning for the presence of transcription factor motifs in promoter areas) were built by SCENIC in mouse (black) or human (orange) datasets. Note that activities of individual transcription factors modelled by SCENIC can be compared across cell states but not across species, given that the regulons were built separately for human and mouse. Red rectangles denote high regulon activities (standardized activity score ≥1). For each cell state four highest-ranking transcription factors are shown. Inset shows smFISH signal of *SATB2* in granule cell lineage cells (*PAX6*) in 12 wpc human cerebellum. Arrows denote examples of cells co-expressing *SATB2* and *PAX6*. **j**, Distribution of standardized regulon activity scores in mouse or human cell states for all transcription factors (grey), marker transcription factors (blue), and conserved marker transcription factors (red). Only transcription factors for which regulons were built by SCENIC are included (mouse n = 447, human n = 499). Scores ≥1 (vertical line) were defined as "high". *P* values are from two-sided Mann-Whitney U tests. EGL, external granule cell layer; LTD, long-term depression; LTP, long-term potentiation; PCA, principal components analysis.

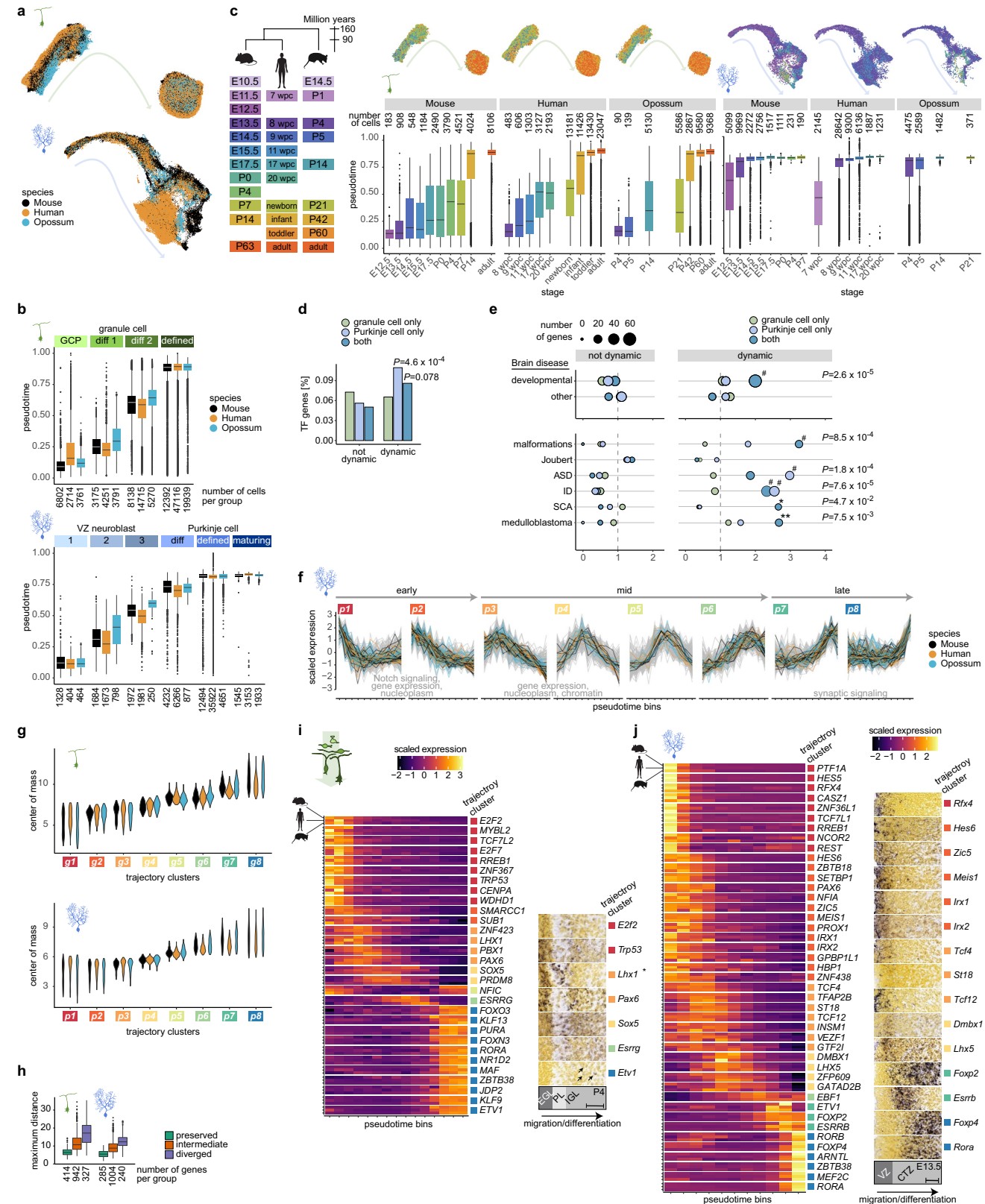

**Extended Data Fig. 9** | See next page for caption.

**Extended Data Fig. 9 | Gene expression trajectories across neuronal differentiation. a**, UMAPs of mouse, human and opossum cells assigned to the granule cell (top) or Purkinje cell (bottom) lineage and integrated across species. Cells are coloured by species. **b**, Pseudotime values across cell state categories for granule (top) and Purkinje (bottom) cell lineage. Number of cells per group is indicated below the axis. **c**, Pseudotime values across developmental stages in the mouse, human and opossum datasets for granule (left) and Purkinje (right) cell lineage. Number of cells per group is indicated above the boxplots. Stage correspondences are shown on the left and the integrated UMAPs plotted per species and coloured by stages at the top. **d**, Percentage of transcription factor genes across gene sets as in Fig. 3e. Adjusted *P* values, two-sided binomial test. **e**, Enrichment of disease-associated genes for genes dynamic or non-dynamic across differentiation of granule cells, Purkinje cells, or both neuron types in all species. Top: inherited brain disease genes[41] were split into two groups based on overlap with developmental disease genes[41]. Bottom: neurodevelopmental and neurodegenerative diseases[13], and malignancies[42] directly linked to cerebellar function and cell types. Adjusted *$P < 0.1$, *$P < 0.01$, #$P < 10^{-4}$, two-sided binomial test. **f**, Clusters of gene expression trajectories across Purkinje cell differentiation. Clusters (p1-p8) are ordered from early to late differentiation based on the mean center-of-mass values of the confident cluster members' trajectories. Strongly preserved trajectories of the orthologues are highlighted with colours. Examples of enriched gene ontology categories for the genes with preserved trajectories are indicated for pairs of clusters. **g**, Centre-of-mass values of individual trajectories across granule (top) and Purkinje (bottom) cell trajectory clusters. **h**, Maximum dynamic time warping aligned distance between the orthologues belonging to different trajectory conservation groups. Number of genes per group is indicated below the axis. **i**, **j**, Expression of transcription factor genes with strongly preserved trajectories across granule (**i**) and Purkinje cell (**j**) differentiation. Scaled expression across pseudotime bins is shown on the left and RNA in situ hybridization data[15] on the right. Areas from sagittal sections of P4 cerebellar cortex in lobule III (**i**) or mouse E13.5 cerebellar primordia (**j**) counterstained with HP Yellow are shown. Schemes of layers in the respective areas are at the bottom. Arrows in **i** point to rare positive cells. *Lhx1* is additionally expressed in Purkinje cells. Scale bars: 50 μm. In **b**, **c** and **h**, boxes represent the interquartile range, whiskers extend to extreme values within 1.5 times the interquartile range from the box, and line denotes the median. ASD, autism spectrum disorders; CTZ, cortical transitory zone; diff, differentiating; EGL, external granule cell layer; ID, intellectual disability; IGL, internal granule cell layer; PL, Purkinje cell layer; SCA, spinocerebellar ataxia; VZ, ventricular zone.

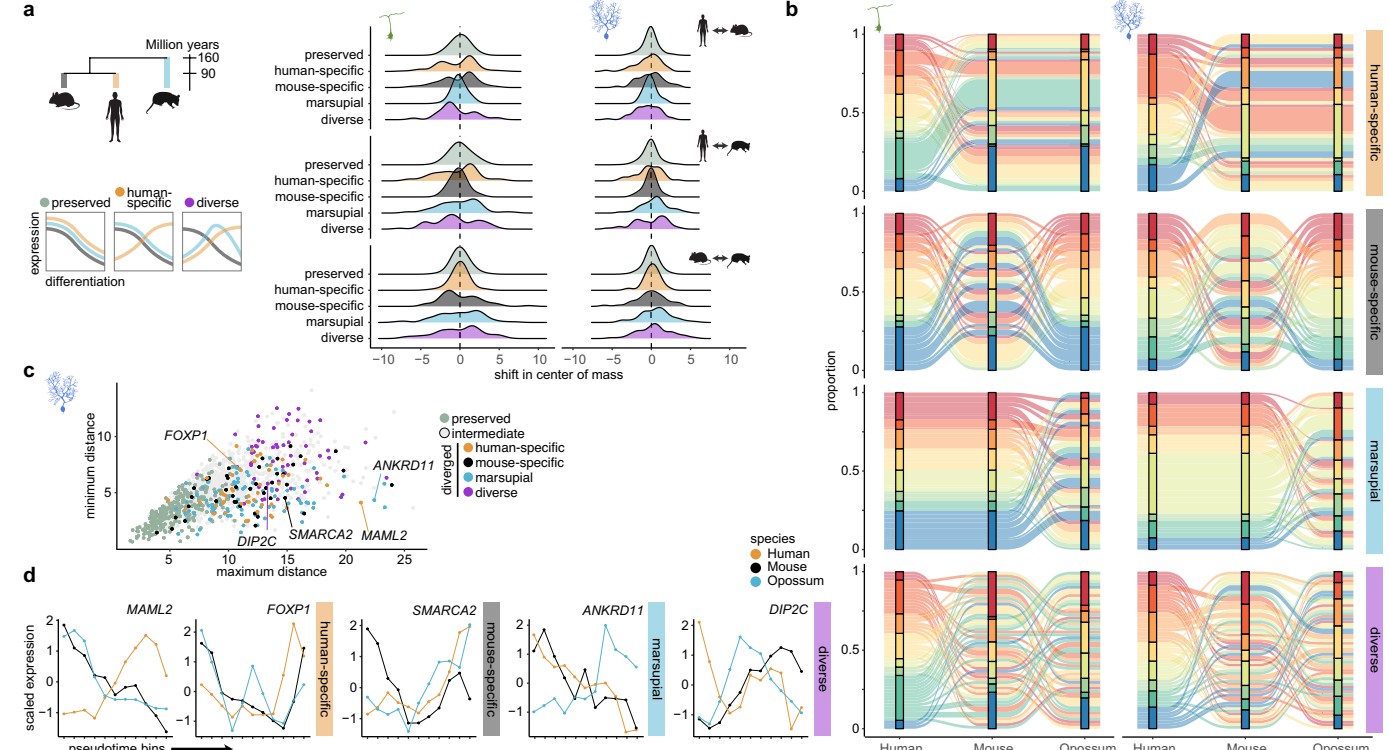

**Extended Data Fig. 10 | Gene expression trajectories across neuronal differentiation. a**, Distribution of the pairwise (human versus mouse, human versus opossum, mouse versus opossum) shifts in the center of mass values of the trajectories that were assigned to preserved, human-specific, mouse-specific, marsupial and diverse patterns groups. The scheme (right) shows examples of a preserved pattern, a change in the human lineage, and a diverse pattern that cannot be assigned to a lineage. The changes in genes that differ between eutherians and opossum (marsupial) cannot be polarized. **b**, Distribution between trajectory clusters for genes with changes in gene expression trajectories during granule cell (left) or Purkinje cell (right) differentiation. **c**, Scatter plot of minimum and maximum pairwise distances between the expression trajectories of orthologues from the three species in Purkinje cell differentiation. High maximum and low minimum distances indicate the strongest lineage-specific trajectory changes. **d**, Examples of genes that display cross-species differences in trajectories during Purkinje cell differentiation.

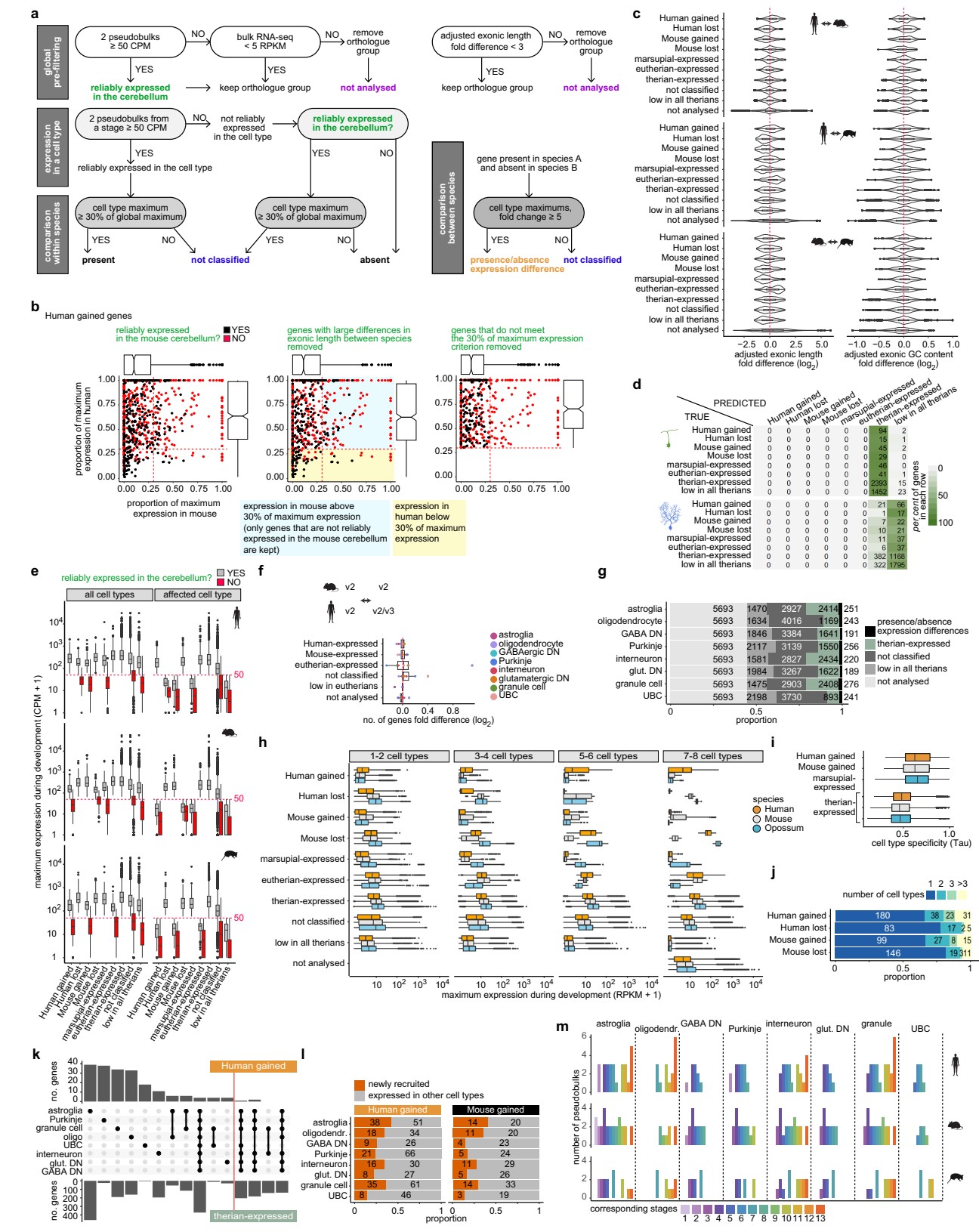

**Extended Data Fig. 11** | See next page for caption.

**Extended Data Fig. 11 | Presence or absence expression differences between species. a**, Scheme on classification of presence or absence expression differences. **b**, Visualization of filtering steps for human gains. The plots show the ratio between the gene's maximum expression within a cell type and the maximum expression of the same gene across all cell types in mouse (x axis) and human (y axis). The red lines indicate the 30% threshold. Initial calls (left), calls after filtering out the genes with large differences in exonic length (middle), and after additionally applying the 30% of maximum expression criterion (right). **c**, Median-adjusted fold difference in exonic length (left) and exonic GC content (right) across species for the genes in different categories assigned based on the presence/absence expression patterns in the cerebellar cell types in the three therian species. Genes below the threshold (<50 CPM) in all species are marked as low in all therians. **d**, Random forest model with exonic length and GC content as predictors are not able to predict the presence or absence expression differences between the species. Granule cell and Purkinje cell calls are shown. **e**, Distribution of expression levels for the genes in different presence/absence categories. Assignments from all eight main neural cell types are included (i.e., each gene is represented eight times). Maximum expression during development among all cell types (15 cerebellar cell types (Supplementary Information); left) or in affected cell type (right) is shown. **f**, Fold difference in the number of genes assigned to different presence/ absence categories using Chromium v2 data only or including also v3 data.

Subsets of human (v2/v3 or v2 only) and mouse (v2 only) datasets were used (Supplementary Information). **g**, Genes assigned to different categories in the presence or absence expression analysis of the three therian species. **h**, Comparison of cerebellar gene expression patterns in the snRNA-seq datasets to the bulk RNA-seq data[19]. Genes were assigned to groups based on the presence/absence expression patterns in the cerebellar cell types in the three therian species. Maximum expression across development, based on the bulk RNA-seq data, is shown. **i**, Cell type-specificity index (Tau) for the genes that are robustly expressed in a single species or all species. Data from all cell types are summarized. The index ranges from 0 (broad expression) to 1 (restricted expression). **j**, Number of cell types in which genes were called as having gained or lost expression in different phylogenetic branches. **k**, Number of genes assigned as human-gained or therian-expressed in different cerebellar cell types and their combinations. Ten cell type groups with the highest number of genes in the human-gained gene set (left from the red line) or therian-expressed gene set are shown. **l**, Number of human-gained and mouse-gained genes that were expressed in other cell types in the cerebellum or newly recruited to the cerebellar transcriptome. **m**, Number of cell type pseudobulk replicates used for expression plots in Fig. 4g,i and Extended Data Fig. 12a,d. In **b**,**e**,**f**,**h**,**i**, boxes represent the interquartile range, whiskers extend to extreme values within 1.5 times the interquartile range from the box, and line denotes the median.

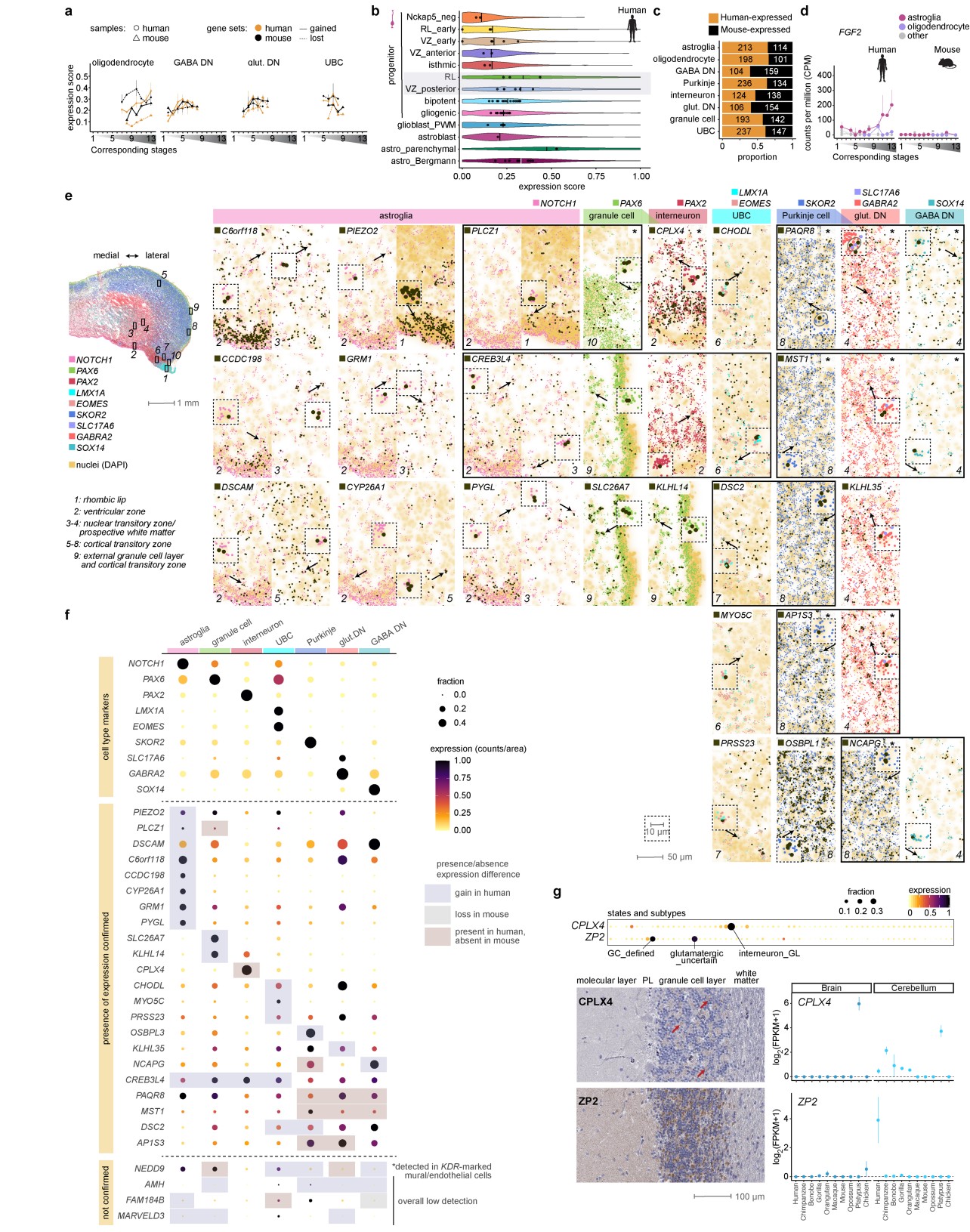

**Extended Data Fig. 12** | See next page for caption.

**Extended Data Fig. 12 | Expression patterns and spatial mapping of genes with presence or absence expression differences. a**, Expression of genes that were gained or lost in the human or mouse lineage in different cerebellar cell types across development. The expression of the genes that were lost in the human lineage were evaluated in the mouse samples, and vice versa. Stages are aligned across species as shown in Fig. 1a; the line indicates the median of biological replicates. **b**, Expression of genes that were gained in the astroglia in the human lineage across astroglia subtypes. The lines show the median of all cells and the dots show the median of each biological replicate that had at least 50 cells in the respective subgroup. **c**, Presence or absence expression differences between mouse and human cerebellar cell types. **d**, Example of a gene that shows a presence or absence expression difference between human and mouse astroglial cells. **e**, Human 12 wpc cerebellum smFISH data for genes with presence or absence expression differences between human and mouse. Asterisks denote expression changes that could not be polarized; other changes were assigned as gains in the human lineage. The locations of the regions 1–10 expanded at the right are shown with rectangles (solid line) on the whole section at the left. mRNA spots are black for the genes with differences, and coloured for the cell type markers. Arrows and insets show example cells where co-expression with the respective cell type marker(s) is detected. **f**, Quantification of the smFISH data in **e**. Cell type labels were transferred based on alignment[83] of the segmented[82] smFISH dataset with the 11 wpc snRNA-seq data (Fig. 1d). Expression levels (mRNA counts) were normalised to segment area and scaled. The cell types where a difference was called are highlighted for each gene. Out of the 26 genes tested, expression of 22 genes in the respective cell type(s) was confirmed by smFISH; 3 genes displayed overall low signal, and expression of one gene remained undetected in a cell type where the change was observed based on snRNA-seq data. **g**, Expression of ZP2 and CPLX4 in the human snRNA-seq dataset from this study (top), in human cerebellar sections as determined by immunohistochemistry (Human Protein Atlas)[45] (left), and in the adult brain and cerebellum from nine mammals and chicken based on bulk RNA-seq data[47] (right). The antibodies used for immunohistochemistry were HPA047627 (CPLX4) and HPA011296 (ZP2). Red arrows point to synaptic staining in the granule cell layer. For brain bulk RNA-seq, prefrontal cortex was sampled for primates, and whole brain, except cerebellum, was sampled for non-primates. GABA DN, GABAergic deep nuclei neurons; glut. DN, glutamatergic deep nuclei neurons; PWM, prospective white matter; RL, rhombic lip; UBC, unipolar brush cell; VZ, ventricular zone.

# Reporting Summary

## Statistics

For all statistical analyses, confirm that the following items are present in the figure legend, table legend, main text, or Methods section.

| n/a | Confirmed | |
|---|---|---|
| ☐ | ☒ | The exact sample size (*n*) for each experimental group/condition, given as a discrete number and unit of measurement |
| ☐ | ☒ | A statement on whether measurements were taken from distinct samples or whether the same sample was measured repeatedly |
| ☐ | ☒ | The statistical test(s) used AND whether they are one- or two-sided<br>*Only common tests should be described solely by name; describe more complex techniques in the Methods section.* |
| ☒ | ☐ | A description of all covariates tested |
| ☐ | ☒ | A description of any assumptions or corrections, such as tests of normality and adjustment for multiple comparisons |
| ☐ | ☒ | A full description of the statistical parameters including central tendency (e.g. means) or other basic estimates (e.g. regression coefficient) AND variation (e.g. standard deviation) or associated estimates of uncertainty (e.g. confidence intervals) |
| ☐ | ☒ | For null hypothesis testing, the test statistic (e.g. *F*, *t*, *r*) with confidence intervals, effect sizes, degrees of freedom and *P* value noted<br>*Give P values as exact values whenever suitable.* |
| ☐ | ☒ | For Bayesian analysis, information on the choice of priors and Markov chain Monte Carlo settings |
| ☒ | ☐ | For hierarchical and complex designs, identification of the appropriate level for tests and full reporting of outcomes |
| ☒ | ☐ | Estimates of effect sizes (e.g. Cohen's *d*, Pearson's *r*), indicating how they were calculated |

*Our web collection on statistics for biologists contains articles on many of the points above.*

## Software and code

Policy information about availability of computer code

| | |
|---|---|
| Data collection | BD FACSDiva 8.0.1 softaware was used for nuclei sorting. |
| Data analysis | Open source software including CellRanger v2 and v3; the R packages (v.3.6.3 and v.4.1.2) tidyverse v1.3.0, SingleCellExperiment v1.6.0, liger v0.4.6, rliger v1.0.0, batchelor v1.0.1, pheatmap80 v1.0.12, ggplot2 v3.3.2, ggExtra v0.10.0, mclust v5.4.3, uwot v0.1.10, Seurat v4.0.6, leidenAlg v1.0.5, HDInterval v0.2.2, dtw v1.20, irlba v2.3.3, SoupX, Harmony v1.0, Mfuzz v2.44.0, ggalluvial v0.12.3, WebGestaltR v0.4.4; the Python (v3.6) packages scanpy v1.5.1, htseq v0.13.5, Scrublet v0.2, Tangram v1.0.3, scikit-learn v1.1.1, pySCENIC; the Julia (v1.6.4) package Baysor v0.5.2; commercial software cellSens (Olympos), Fiji plugin Polylux (V1.9.0., Resolve Biosciences). Custom code is available at https://gitlab.com/kaessmannlab/mammalian-cerebellum |

For manuscripts utilizing custom algorithms or software that are central to the research but not yet described in published literature, software must be made available to editors and reviewers. We strongly encourage code deposition in a community repository (e.g. GitHub). See the Nature Portfolio guidelines for submitting code & software for further information.

## Data

Policy information about availability of data

All manuscripts must include a data availability statement. This statement should provide the following information, where applicable:

- Accession codes, unique identifiers, or web links for publicly available datasets
- A description of any restrictions on data availability
- For clinical datasets or third party data, please ensure that the statement adheres to our policy

The datasets generated in the current study are available in the heiDATA repository, https://doi.org/10.11588/data/QDOC4E. Processed data can be interactively explored at https://apps.kaessmannlab.org/sc-cerebellum-transcriptome. Mouse and human processed data are also available as a CELLxGENE collection at https://

# Field-specific reporting

Please select the one below that is the best fit for your research. If you are not sure, read the appropriate sections before making your selection.

☒ Life sciences ☐ Behavioural & social sciences ☐ Ecological, evolutionary & environmental sciences

For a reference copy of the document with all sections, see nature.com/documents/nr-reporting-summary-flat.pdf

# Life sciences study design

All studies must disclose on these points even when the disclosure is negative.

| | |
|---|---|
| Sample size | No statistical methods were used to predetermine sample size. At least 2 biological replicates were generated for each developmental stage (except human 20 wpc). All samples are listed in Supplementary Table 1, and an overview of the samples is given in Extended Data Fig. 1a. Human sample size was based on the number of individuals available, and comparable sample size was used for mouse and opossum. |
| Data exclusions | Low quality nuclei and mislabeled samples (not cerebellum) were excluded as described in the Methods. |
| Replication | At least 2 biological replicates were generated for each developmental stage (except human 20 wpc). Data from all replicates was included in the final dataset as long as the data quality and sample identity criteria were met (see Data exclusions). |
| Randomization | Randomization was not used in this study. Randomization was not relevant to our study since samples were not allocated into experimental groups. |
| Blinding | Blinding was not relevant to our study. Both data collection and analyses required an understanding of the nature of the sample being collected/analyzed. |

# Reporting for specific materials, systems and methods

We require information from authors about some types of materials, experimental systems and methods used in many studies. Here, indicate whether each material, system or method listed is relevant to your study. If you are not sure if a list item applies to your research, read the appropriate section before selecting a response.

## Materials & experimental systems

| n/a | Involved in the study |
|---|---|
| ☐ | ☒ Antibodies |
| ☒ | ☐ Eukaryotic cell lines |
| ☒ | ☐ Palaeontology and archaeology |
| ☐ | ☒ Animals and other organisms |
| ☐ | ☒ Human research participants |
| ☒ | ☐ Clinical data |
| ☒ | ☐ Dual use research of concern |

## Methods

| n/a | Involved in the study |
|---|---|
| ☒ | ☐ ChIP-seq |
| ☐ | ☒ Flow cytometry |
| ☒ | ☐ MRI-based neuroimaging |

# Antibodies

| | |
|---|---|
| Antibodies used | Lmx1a-Millipore:AB10533-lot:3868680; TBR2/EOMES-Millipore:ABN1687-Lot:Q3076145; TBR2/EOMES- Millipore:AB15894-lot:3090750; HCRTR2-R&D:MAB52461-lot:CBFY0115061; HCRTR2-Alomone Labs:AOR-002.<br><br>Donkey anti-Mouse IgG (H+L) Alexa Fluor 488 (Invitrogen, Cat.No.: A21202), Donkey anti-Mouse IgG (H+L) Alexa Fluor 568 (Invitrogen, Cat.No.: A10037), Donkey anti-Rabbit IgG (H+L) Alexa Fluor 488 (Invitrogen, Cat.No.: A21206), Donkey anti-Rabbit IgG (H+L) Alexa Fluor 568 (Invitrogen, Cat.No.: A10042), and Goat Anti-Chicken IgY (H+L) Alexa Fluor 568 (Abcam, Cat.No.:ab175477). |
| Validation | AB10533: evaluated by Western Blot in mouse testis tissue lysate by the provider; IHC signal in the rhombic lip shown by Yeung et al. 2014 (https://doi.org/10.1523/JNEUROSCI.1330-14.2014).<br>ABN1687: A representative lot detected TBR2 in Immunofluorescence applications (Nelson, B.R., et. al. (2013). J Neurosci. 33(21):9122-39; Hodge, R.D., et. al. (2013). J Neurosci. 33(9):4165-80), in immunohistochemistry applications (Yoon, K.J., et. al. (2008). Neuron. 58(4):519-31).<br>AB15894: a representative lot detected TBR2 in mouse cerebral cortex and mouse cerebellum tissue, as tested by the provider; staining in UBCs shown by Canton-Josh et al. 2022 ( https://doi.org/10.7554/eLife.76912).<br>MAB52461: HCRTR2 was detected in immersion fixed paraffin-embedded sections of human brain (hypothalamus) by the provider.<br>AOR-002: IHC signal in the hypothalamus shown by Parekh et al. 2021 (https://doi.org/10.1038/s41598-021-00522-0). |

# Animals and other organisms

Policy information about studies involving animals; ARRIVE guidelines recommended for reporting animal research

| | |
|---|---|
| Laboratory animals | RjOrl:SWISS and Bl6N mice (Mus musculus), gray short-tailed opossum (Monodelphis domestica). Animals from both sexes were used. Ages of the animals are listed in Supplementary Table 1, and an overview is given in Fig. 1a and Extended Data Fig. 1a. The animals were housed under a 12h/12h dark/light cycle (reversed for opossums) in a temperature (20-24 °C mouse; 24-26 °C opossum) and humidity (40-65% mouse, 60-65% opossum) controlled room with ad libitum access to food and water. |
| Wild animals | This study did not involve wild animals. |
| Field-collected samples | This study did not involve samples collected from the field. |
| Ethics oversight | All animal procedures were performed in compliance with national and international ethical guidelines for the care and use of laboratory animals, and were approved by the local animal welfare authorities: Heidelberg University Interfaculty Biomedical Research Facility (T-63/16, T-64/17, T-37/18, T-23/19), Vaud Cantonal Veterinary Office (No.2734.0) and Berlin State Office of Health and Social Affairs, LAGeSo (T0198/13, ZH104). |

Note that full information on the approval of the study protocol must also be provided in the manuscript.

# Human research participants

Policy information about studies involving human research participants

| | |
|---|---|
| Population characteristics | Human samples were obtained from official scientific tissue banks. The samples come from healthy non-affected individuals defined as normal controls by the corresponding brain bank. Individuals from both sexes were included. For all samples, the sex and age are reported in Supplementary Table 1. |
| Recruitment | Informed consent for the use of tissues for research was obtained in writing from donors or their family. |
| Ethics oversight | The use of human samples was approved by an ERC Ethics Screening panel (associated with ERC Consolidator Grant 615253, OntoTransEvol) and ethics committees in Heidelberg (authorization S-220/2017), North East-Newcastle & North Tyneside (REC reference 18/NE/0290), London-Fulham (REC reference 18/LO/0822), Ministry of Health of Hungary (No.6008/8/2002/ETT) and Semmelweis University (No.32/1992/TUKEB). |

Note that full information on the approval of the study protocol must also be provided in the manuscript.

# Flow Cytometry

## Plots

Confirm that:

☒ The axis labels state the marker and fluorochrome used (e.g. CD4-FITC).

☒ The axis scales are clearly visible. Include numbers along axes only for bottom left plot of group (a 'group' is an analysis of identical markers).

☐ All plots are contour plots with outliers or pseudocolor plots.

☒ A numerical value for number of cells or percentage (with statistics) is provided.

## Methodology

| | |
|---|---|
| Sample preparation | Nuclei were extracted from frozen tissues and stained with Hoechst |
| Instrument | BD FACSAria III |
| Software | BD FACSDiva 8.0.1 |
| Cell population abundance | The nuclei were sorted based on the Hoechst staining. No quantifications were performed based on flow cytometry data. Further selection of barcodes containing a single nucleus was done based on RNA-sequencing data. |
| Gating strategy | To separate nuclei from the cellular debris the gates were set on FSC/SSC and at the excitation wavelength of 405 nm. |

☒ Tick this box to confirm that a figure exemplifying the gating strategy is provided in the Supplementary Information.

