## [Peer Review File · Nature]

Manuscript Title: Cellular development and evolution of the mammalian cerebellum

Reviewer Comments & Author Rebuttals

Reviewer Reports on the Initial Version:

Referees' comments:

Referee #1 (Remarks to the Author):

Sepp and colleagues describe a developmental census of cellular diversity in the cerebellum of human, mouse, and opossum using expression profiles from ~400,000 single nucleus RNA-seq datasets. They report a high degree of conservation of cell types and states, relative proportions, and expression developmental trajectories including a core transcription factor cascade that presumably drives cell fate. This short list of TFs provides a clear path for future experimental work to unravel the molecular mechanisms underlying cell differentiation and maturation. Conserved cell type markers will facilitate the generation of genetic tools to target specific populations for functional studies. Moreover, defining cell type homologies across species enables predictions of spatial distributions and developmental origins of types in human cerebellum based on data from mouse.

The authors also highlight human-specific differences, including an expanded pool of Purkinje neurons in the first trimester, altered expression trajectories, and de novo gene expression in the human or great ape lineages. Some human-specific genes are associated with cerebellar diseases, and the authors provide a companion paper that leverages these reference cell types to associated pediatric tumors with different cerebellar cell populations. Altogether, this manuscript provides insight into the evolution of an understudied region that contains the largest number of neurons in the human brain and provides a resource that will be of broad interest, particularly in genomics, neuroscience, and translational medicine.

While the manuscript is quite comprehensive, there are some additional analyses that would be helpful to include. First, the authors comment on conserved TFs but say little about the other conserved genes that define the core identity of cell types. What are the key ion channels, receptors, or cell adhesion molecules that may be contributing to known cellular properties? Second, do disease genes that show human-specific expression participate in different gene coexpression networks than mouse? Or are they coexpressed with similar genes and expressed in different cell types? Third, the most striking finding is the expansion of Purkinje neurons yet reduced diversity in human (2 vs. 4 subtypes in mouse and opossum). Please comment more on this species difference. If it is driven by differences in sampling, why is this the only cell population that is so dramatically affected? If you integrate data across species, do you see overlap of the human LB and EB cells with the 4 populations in the other species? Or are they only partially overlapping? Do markers of the 4 types have patterned expression in the human Purkinje UMAP?

Other comments:

- Please discuss why there are missing types before 17pcw in human where there should be less chance of sampling bias.

- Human and mouse gains could be inflated based on lower detection of genes in opossum using 10x v2 data (i.e. these should be human or mouse losses not gains). Since you have some v3 opossum data, what is the effect of using v2 vs. v3 on classifying expression changes?

- Fig. 3d - Please comment on why there is a gap between differentiating and mature GCs not Purkinje cells?

- Fig. 3h - Do genes with human-specific trajectories show greater phenotypic constraint vs. genes divergent in mouse or opossum?

Referee #3 (Remarks to the Author):

Manuscript by Sepp et al seeks to reconstruct cerebellar development from embryonic to adult stages with a specific emphasis on evolutionary differences between opossum, mouse, and humans.

Towards this goal, the authors generate almost 400,000 single cells from the developing human, mouse and opossum cerebellum across a wide range of ages. Cross-species alignment of developmental trajectories is the main focus of the analysis. Both data generation and bioinformatic analyses are well executed using well-established methods.

The manuscript is well written and easy to follow, and the dataset is likely to be useful for the community.

My main reservation relates to the issue of novelty. Prior study by Aldinger et al has already performed a very similar analysis in mouse and human. It is true that manuscript by Sepp et al further extends this data by adding additional time points and another species, but the conceptual value added by the analyses seems more incremental due in part to the overall lack of novel biological insights, lessening my enthusiasm for the current study.

Major comments

1. The novelty of the developmental trajectory is particularly lacking based on recent findings in the human and mouse cerebellum development from scRNA-seq and spatial analysis (Aldinger et al.) which is recapitulated here especially in Fig. 2 on the rhombic lip and the divergence of Purkinje cells.
2. Because many of the analyses of cell states are derived from single cell data alone, it will be important to validate some of the newly discovered cell states with FISH/IHC in human tissue.
3. It seems that while the authors include data from postnatal/adult stages, but the analysis is relatively underdeveloped, and lacks biological insights.
4. There is a significant amount of usage of abundance estimates here with the scRNA-seq data which should be treated with caution. Especially considering library imbalance therefore the text should be revised as such with the analysis like in Fig. 1d,e Extended Data Fig7, etc. Statistical methods to support such claims would also be needed, and emerging tools are beginning to enable this for scRNA-seq data.
5. The evolutionary analysis on Fig 4 is highly underdeveloped. It would be interesting to functionalize in-silico the gene divergences between the species. Specifically in Fig, 4b, the human specific genes could be analyzed with GSEA or even simpler with GO. This can also be done for Fig 4g in the context for both mouse and human. In Fig4e, how many of those genes are specific to the cell types? If not, the chart should ideally represent cell-type specific genes with significant expression that are gained in different phylogenetic branches. Candidate genes with divergent expression would have to be validated using in situ hybridization at the very least.

Minor Comments

1. Extended Data Fig. 1e doesn't have a legend.
2. In Extended Data Fig. 2 the pipeline shows fastMNN using 100 dimensions for correction. Why is the dimension set so high, especially when this is done across species so the number of orthologous genes is less than what batch correction takes into account?
3. Were there any functional categories from the GO analysis within the trajectory analysis with divergence between the species (in regards to Fig. 3f and Extended Data Fig. 9g)?
4. Fig. 1c is extremely large with the UMAP taking up a lot of space for each individual species, is there an issue with integrating the entire dataset? The major cell types should still be recapitulated?
5. In Extended Data Fig. 8a,b what are the genes loaded into the first 2 PC since those seem to be contributing to the most of the variance? Otherwise a,b, and c aren't helpful in showing anything. This comment also applies to Fig 3a.
6. In Extended Data Fig. 8g were the TF for the cell types generated using a specific analyses method such as SCENIC? If not, why wasn't it used (or any other published TF activity methods)?
7. In Extended Data Fig. 10 passing some of gene groups found such as the Human gained gene group though GO would help interpret the findings.

Referee #4 (Remarks to the Author):

In this manuscript, Sepp, Leiss, and coauthors present a multispecies single-cell RNA sequencing dataset from developing cerebellum, extracted from three different species: human, mouse, and opossum. The aim of the study is to describe the differences and similarities in the process of cerebellar development across species.

The authors analyze the dataset and provide a classification of cell types present at the different stages of development in the three species. First, the authors approach the problem of determining a correspondence between the cerebellar development of these species to an unprecedented level of granularity. The annotation of the different cell populations across different species is performed carefully and with attention to the literature returning proportions that seem reasonably in line with the cell type composition for mouse and developing human cerebellum established in other studies. While missing a few intermediate states, the dataset can be considered comprehensive for most practical purposes.

The manuscript provides knowledge at a different level of granularity, maybe not all immediately digestible at a first read but useful to consultation (i.e. there are many interesting summary visualization worth consulting in the supplementary figures). Overall, the resource has a tremendous value for understanding cerebellum evolution. The data per se has a high potential to be re-used for other metaanalyses. The analyses performed by the authors are of a high standard, the work of annotation and curation is excellent, and several details of the analysis are innovative and exemplar for the field.

The authors, first, approach the developmental population dynamic of different cell types. Purkinje cells in humans are identified to have a unique mode of expansion, while other cell types' developmental trajectories are revealed as non-conserved. Then, the authors propose a gene-

centric evolutionary analysis of the trajectories. They identify several disease-associated genes that behave differently in human cerebellar development than mouse and opossum. This leads to the impactful conclusion that mouse models might not be as relevant for studying disruptions of cerebellar development as they are believed.

A web resource accompanies the paper; while not perfect (e.g., integrated UMAP of the species is missing), it succeeds in making the data more available for the public.

The only general worry I have about the manuscript is that it is not always clear which of the several points proved is an entirely unprecedented discovery or rather just a systematization/validation of something already known. For example, to my knowledge, the fact that Purkinje cells progenitors in humans go through secondary expansion in the subventricular zone (SVZ), akin to cortical neurons in the human cortex, is well-known in the field. Since my primary expertise is not in cerebellar cell type development, I suggest that an expert with complete knowledge of the literature be consulted to evaluate these aspects.

Overall, given the quality of data, well-delineated analyses, care in the annotation and coherence of the story, I believe the work is of great importance for the developmental neuroscience community, and I can forecast its impact going beyond its core-specific field and constitute a landmark study. Therefore, I am highly supportive of the publication of this work in Nature after appropriate revision.

MAJOR POINTS

The authors identify the correspondence between different stages of cerebellar development based on transcription data and claim that there is no major heterochrony. While overall the approach is reasonable, it would be stronger if supported by morphological data: it would be useful to provide histological micrographs for the cerebellum in all three species and indicating homologous subdomains on different developmental timepoints, in particular for opossum where not much is available.

The time correspondences proposed do not always match the correspondence between different developmental stages identified in previous publications by the same approach. For example, in "Gene expression across mammalian organ development" (by the same collective of authors), mouse developmental stages e13.5 and e14.5 are shown to be the most transcriptionally similar to the opossum P2 stage, while in the current manuscript, the same stages correspond to P4-P5 stage in the opossum. Similarly, a human embryo on 7wpc was shown to have the highest transcriptional correlation with e12.5-e14.5 mouse embryo in the previous work, while in the current work 7wpc human cerebellum corresponds to e11.5 mouse cerebellum. While the differences seem to be minor, there are rapid changes in the mouse embryo with every day of development; therefore, identifying the wrong correspondence might be detrimental for further comparative analysis. This aspect should be at least discussed.

Related to the previous point: looking at the correlation maps between homologous cell types coming from different species (e.g., Extended data fig. 3c, extended data fig. 5e), it is evident that the mature cell types and the progenitors at the beginning of the developmental process have more correlation across species than intermediate progenitors. It would be useful if the author could specify how this should be interpreted. E.g., is this the effect of the incomplete sampling/imperfect matching of the intermediate stages of development between species, or are the intermediate stage progenitors less conserved in evolution?

Related to the sampling strategy: the authors claim to identify certain cell types (e.g., GC_diff2_KCNIP4 in extended data fig. 5, cell type isth_N_SLC5A7 in extended data fig. 4) that are present in mouse and opossum, but not in human. For granule cells cluster, the explanation

could be in the sampling strategy (as authors indicate themselves); however, to make the study stronger in that sense; we think other datasets of cerebellar human development should be taken into consideration. E.g., are those cell types possible to find in the recently published dataset on human cerebellar development ("Spatial and cell type transcriptional landscape of human cerebellar development" Aldinger et al.)?

Similarly, there are some populations that authors find in human developing cerebellum but not in mice/opossum ones. For example, populations 7 and 8 (glutamatergic deep nuclei maturing and mature neurons) in extended data fig.4 are present only in human data, while glutamatergic defined (population 6) neurons are present in all three species. What's the author's explanation for the observation? The authors should provide a pseudotime analysis of human populations to see if clusters 7 and 8 neurons are more mature than cluster 6 neurons. The generation of an integrated UMAP for deep nuclei cells across three species could help determine if populations 7 and 8 are mappings to other populations in mouse/opossum developing cerebellum.

We find the claim that the authors make about evolutionarily conserved trajectories of different cell types development (except for Purkinje cells) to be partially misleading. The authors base this conclusion on the results of the hierarchical Bayesian modeling of different cell types' developmental dynamics. They find a significant difference (comparing across species) to be present only for Purkinje cells; however, this is not in line with the current state of the knowledge in the field. Their model indicates the secondary expansion that Purkinje cells progenitors undergo in cerebellar SVZ; however, granule cells are known to undergo the same process later in the development. If this is not possible to model because of the sampled time points/parts of the tissue, the claim of "conserved evolutionary trajectories for cell types except for Purkinje cells" should be made with caution.

Moreover, oligodendrocytes in humans have intracerebellar origin in contrast to oligodendrocytes in mice and opossum. In that light, the fact that the models for oligodendrocytes developmental dynamics are similar between humans and the other two species is surprising. Based on the UMAPs dedicated to glial development (extended data fig.6d), the difference in oligodendrocytes developmental trajectory between species should be striking.

Could the authors reanalyze this part of the data with this in mind? At least the result hierarchical Bayesian model for granule cells and oligodendrocytes should be visualized. Are there may be differences in migration markers? Finally, It would be interesting to come up with a proxy to estimate whether glial development is more or less evolutionarily conserved than neuronal development.

While the authors use a well-recognized tool for integration (LIGER) between pairs of datasets, the batch effect is clearly still present on integrated UMAPs shown on extended data fig. 2b. Did the authors try other integration algorithms (e.g., Harmony, Seurat integration)? A mini benchmark of this in the supplementary could be of great technical help to the field of evolutionary single-cell analyses.

The authors claim to identify many target genes with unique expression patterns in humans. This is an interesting broad result that should be qualified more to be a relevant finding. What are the genes that are "lost" in humans? It would be extremely helpful to perform a gene ontology or gene set enrichment analysis to understand whether some biological process or pathway is overrepresented in this set.

Finally, regarding all these claims about genes specific or lost in one species. It would be important to have internal positive controls to support these statements. A control would serve the purpose of avoiding false negatives: genes that are close to the detection sensitivity of scRNA-seq and are "dropping out" in the species where they are expressed the least. Moreover, because of evolutionary divergences of sequence at the 3'UTR (note: 10X chromium is a 3' method), some genes might be more or less efficiently reverse-transcribed and amplified for one species

generating false negatives. Therefore, convincing proof a gene is lost in a lineage would have to include a positive controls that are detectable in another cell type in the same species. In addition, at least one of these two further corroborations should be presented: (i) the average abundance distribution of the "lost genes" is not significantly different from the distribution of other, similarly variable genes. (ii) checking that changes sequence and/or exon structure comparison of the 3' part of the gene model between the species cannot explain the "lost genes" (this can be done with a Generalized Linear Model with predictors summary statistics from the 3' part analysis: say GC content, length,...).

MINOR COMMENT

On extended data fig. 5b, the numbers on the UMAP do not correspond to the numbers assigned to the clusters on extended data fig. 5a subplot.

Author Rebuttals to Initial Comments:

Response to the referees

We thank the referees for thoroughly reviewing our manuscript and for the constructive criticism. We have made every attempt to fully address each of their concerns, which we believe resulted in an improved manuscript.

Below we provide a summary of the major changes to the manuscript, followed by the detailed point by point responses to the referees' concerns. To facilitate readability, we use various font emphases and colours, as outlined below:

The comments from the referees are in bold;
our responses are in normal text;
changes to the manuscript are highlighted in yellow.

Page and line numbers refer to the current revised manuscript. Figures exclusively addressed to the reviewers are termed "Reviewer Figures". All references are numbered in the context of this response.

The key revisions can be summarised as follows:

- To substantiate the cell type annotations and the presence/absence expression differences between the species, we performed multiplexed single molecule RNA FISH on a tissue section from human 12 weeks post conception (wpc) cerebellum. The inclusion of this spatial dataset and the associated analyses not only provided a strong validation of our findings, but also enabled detection of some of the "missing" subtypes in human, and yielded novel insights into the spatial distribution of cerebellar cell types.
- We conducted histological stainings of opossum cerebellum and compared its morphology to that of human and mouse to confirm the stage correspondences between species.
- By incorporating the human 12 wpc spatial dataset and reanalyzing an independent snRNA-seq dataset of human fetal cerebellum (Aldinger et al. 2021¹), we have expanded our previous conclusions regarding human Purkinje cell diversity. We now confirm the existence of four developmental subtypes in humans, mirroring the observations in mouse and opossum. Furthermore, our findings reveal a human-specific shift in the prevalence of Purkinje cell subtypes that accompanies the general increase in Purkinje cell relative abundances during critical developmental stages in humans.
- We have thoroughly reassessed the assignments of presence/absence expression differences between the species. As a result, we have incorporated two additional filtering steps aimed at mitigating potential biases associated with cross-species differences in sequence features and genome annotation quality, which could affect the snRNA-seq detection sensitivity. Importantly, while these changes did not alter the main conclusions, the implementation of more stringent criteria significantly enhances the quality and reliability of our classifications.

The new spatial dataset generated during the revision is available in the heiDATA repository via this private link: <https://heidata.uni-heidelberg.de/privateurl.xhtml?token=eed9204c-8e60-41ae-8dde-acc328d29e1>.

CONTENTS

Referee #1	2
SUMMARY	2
MAJOR COMMENTS	2
MINOR COMMENTS	8
Referee #3	13
SUMMARY	13
MAJOR COMMENTS	13
MINOR COMMENTS	27
Referee #4	33
SUMMARY	33
MAJOR COMMENTS	34
MINOR COMMENT	57
References	57

Referee #1

SUMMARY

Sepp and colleagues describe a developmental census of cellular diversity in the cerebellum of human, mouse, and opossum using expression profiles from ~400,000 single nucleus RNA-seq datasets. They report a high degree of conservation of cell types and states, relative proportions, and expression developmental trajectories including a core transcription factor cascade that presumably drives cell fate. This short list of TFs provides a clear path for future experimental work to unravel the molecular mechanisms underlying cell differentiation and maturation. Conserved cell type markers will facilitate the generation of genetic tools to target specific populations for functional studies. Moreover, defining cell type homologies across species enables predictions of spatial distributions and developmental origins of types in human cerebellum based on data from mouse.

The authors also highlight human-specific differences, including an expanded pool of Purkinje neurons in the first trimester, altered expression trajectories, and *de novo* gene expression in the human or great ape lineages. Some human-specific genes are associated with cerebellar diseases, and the authors provide a companion paper that leverages these reference cell types to associated pediatric tumors with different cerebellar cell populations. Altogether, this manuscript provides insight into the evolution of an understudied region that contains the largest number of neurons in the human brain and provides a resource that will be of broad interest, particularly in genomics, neuroscience, and translational medicine.

While the manuscript is quite comprehensive, there are some additional analyses that would be helpful to include.

We thank the reviewer for the positive evaluation of our work. We hope to have addressed the remaining critiques of the reviewer in the revised manuscript.

MAJOR COMMENTS

Major comment 1: First, the authors comment on conserved TFs but say little about the other conserved genes that define the core identity of cell types. What are the key ion channels, receptors, or cell adhesion molecules that may be contributing to known cellular properties?

We agree with the Reviewer that besides the TFs, proteins from other molecular function categories contribute to the definition of cell types. To systematically address this in the revised manuscript, we performed an over-representation analysis on the conserved marker genes from all cell states. The analysis was performed using the WebGestaltR² package. Amongst molecular function categories, this analysis revealed enrichment for extracellular matrix and adhesion proteins, transmembrane transporters (including ion channels), ligands and receptors, TFs, and proteins involved in plasma membrane and vesicle dynamics, in line with the reviewer's suggestion. These enrichments are significant in comparison to all detected genes as well as in comparison to species-specific marker genes ($P < 0.05$, hypergeometric test with BH adjustment). Nevertheless, when looking at individual cell states, TFs showed the highest maximum proportions (79% TF genes among the conserved markers of VZ_neuroblast_2 state; ~12-fold enrichment compared to the background level of 6.6%), compared to other broad molecular function categories that we found enriched among the conserved markers (25% ligands and receptors among the conserved markers of Purkinje_defined state; 5.9-fold enrichment compared to the background level of 4.3%). Given these enrichments and the roles of TFs in cell fate decisions (Arendt et al 2016)³, we still highlight the TFs in the text, and extend our analyses to include the investigation of the activities of the TFs (inferred by SCENIC), an analysis suggested by Referee #3 (see Referee #3 minor comment 6, page 30).

In the revised version of the manuscript, we now highlight the different molecular function categories enriched among the conserved markers of all cell states, and report the significant GO and KEGG terms in Extended Data Fig. 8f,g and Supplementary Table 8. For consistency, we updated the analysis of GO enrichments among the conserved markers of individual cell states, and performed it using the WebGestaltR package (Supplementary Table 8). Additionally, in Supplementary Table 7, which lists all conserved marker genes, we expanded the annotation of genes to encompass not only the TF category, but also other enriched broad molecular function categories (as shown in Extended Data Fig. 8f,g).

Changes to the manuscript

- Main text, page 14, lines 318-323:

Consistently, conserved markers are associated with pertinent gene ontology terms, including ‘neural tube development’ for progenitors, ‘neuron migration’ for differentiating granule cells, and ‘ensheathment of neurons’ for oligodendrocytes (Supplementary Table 8). At the level of molecular functions, the conserved markers are enriched for extracellular matrix and adhesion proteins, transmembrane transporters, ligands and receptors, TFs, and proteins involved in plasma membrane and vesicle dynamics (Extended Data Fig. 8f-g, Supplementary Table 8).

- Extended Data Figures:

Extended Data Fig. 8. Transcriptional programs and marker genes.

e, Numbers of species-specific and conserved cell state markers among 1:1:1 orthologous genes. **f**, Enriched gene ontology molecular function categories among the conserved markers. Terms are grouped into broad categories as indicated by the colours. **g**, Representation of the broad molecular function categories among the conserved markers of individual cell states. Enrichments were identified using hypergeometric tests against a background of all 1:1:1 orthologous genes detected in the cell states included in the analysis; P -values were adjusted for multiple testing using Benjamini-Hochberg method; * $P < 0.05$, ** $P < 10^{-2}$, *** $P < 10^{-3}$, # $P < 10^{-6}$. **h**, Conserved markers shared between cell states. All groups with more than 5 genes are shown. **i**, Expression and regulon activities of TFs that are among the conserved markers, across cell states in mouse (black), human (orange) and opossum (blue). Dot size and colour intensity indicate the fraction of cells expressing each gene and the mean expression level scaled per species and gene, respectively. Colours on top mark the TFs for which regulons (i.e. co-expression modules that are retained after pruning for the presence of TF motifs in promoter areas) were built by SCENIC in mouse (black) or human (orange) datasets. Red rectangles denote high regulon activities (standardised activity score ≥ 1). For each cell state four highest-ranking TFs are shown. Inset shows smFISH signal of SATB2 in granule cell lineage cells (PAX6) in 12 wpc human cerebellum. Arrows denote examples of cells co-expressing SATB2 and PAX6. **j**, Distribution of standardised TF regulon activity scores in mouse or human cell states for all TFs (grey), marker TFs (blue), and conserved marker TFs (red). Only TFs for which regulons were built by SCENIC are included (mouse $n=447$, human $n=499$). Scores ≥ 1 (vertical line) were defined as “high”.

- Methods, page 55, lines 1325-1333:

Gene ontology and pathway enrichment analyses.

.../Functional enrichments among the conserved markers (Extended Data Fig. 8f, Supplementary Table 8) were identified by over-representation analysis against the background of three-way 1:1 orthologues detected in all species ($FDR < 0.05$) using the functional databases *biological process noRedundant*, *molecular function noRedundant*, *cellular component noRedundant*, and *KEGG*. The molecular function terms, enriched among all conserved markers (Extended Data Fig. 8f), were manually grouped into broad categories. For each broad term category, we tested the enrichments among the conserved markers of individual cell states using hypergeometric tests and adjusted P -values using Benjamini-Hochberg method (Extended Data Fig. 8g). The gene lists for the respective terms were extracted from the WebGestalt² databases, the transcription factor list was downloaded from the animal TFDB (v.3.0)⁴.

- Supplementary Tables:

Supplementary Tables 7 and 8 were updated:

- (i) Annotations for the enriched broad molecular function categories are provided (Supplementary Table 7).
- (i) GO and KEGG terms enriched among the conserved markers of all cell states (Supplementary Table 8).
- (ii) The terms enriched among the conserved markers of the individual cell states were updated for consistency (Supplementary Table 7).

Major comment 2: Second, do disease genes that show human-specific expression participate in different gene coexpression networks than mouse? Or are they coexpressed with similar genes and expressed in different cell types?

We thank the Reviewer for bringing this intriguing topic to our attention. In the revised manuscript, we have refined our approach for the identification of genes that display presence/absence expression differences between species (see also Referee #4 major comments 10, page 50). The refined approach is more stringent and additionally includes a separate comparison of human and mouse data only (without polarisation of the changes using the opossum data). This led to the identification of 1,392 genes with presence/absence expression differences between human and mouse. Among these are 26 cerebellum-linked disease genes. Given the space limitation of the current manuscript, we prefer not to include analyses on co-expression networks in this study. Additionally, we believe that a more thorough approach that incorporates information about gene regulatory networks is needed to properly address this question. Based on several seminal studies (Wittkopp et al. 2004, Wilson et al. 2008)^{5,6}, we hypothesise that many of the expression shifts are facilitated by changes in the *cis* regulatory elements, suggesting that these genes often participate in new co-expression networks. Shifts in the expression of an entire set of co-expressed genes would require many changes in *cis* regulatory elements or changes in the expression/activity of *trans* factors, though possible we consider it to be less frequent. To gain insights into the contributions of the different mechanisms, it would be of great interest to identify the *cis* regulatory changes associated with the gene expression shifts in the cerebellum by integrating single-cell measurements of gene expression and chromatin accessibility. Generating and analysing these data is beyond the scope of this manuscript, but it will be interesting to address this question in the future.

Major comment 3: Third, the most striking finding is the expansion of Purkinje neurons yet reduced diversity in human (2 vs. 4 subtypes in mouse and opossum). Please comment more on this species difference. If it is driven by differences in sampling, why is this the only cell population that is so dramatically affected? If you integrate data across species, do you see overlap of the human LB and EB cells with the 4 populations in the other species? Or are they only partially overlapping? Do markers of the 4 types have patterned expression in the human Purkinje UMAP?

We agree with the Reviewer that the unique dynamics of Purkinje cells in human are interesting and worth further investigation. Although we were able to distinguish only two Purkinje subgroups in our snRNA-seq dataset, we did not intend to claim that the diversity of Purkinje subtypes in human is reduced. We lacked the confidence to further split the EB and LB groups into the 4 subtypes we identified in mouse and opossum since the groups expressing the different subtype markers had less distinct expression patterns in the human data. We have included UMAPs that summarise the expression patterns of the markers in the revised manuscript (Extended Data Fig. 3I) and show the individual gene patterns in Reviewer Fig. 1.1a.

Reviewer Figure 1.1. Purkinje subtypes. a, UMAPs showing expression of key marker genes in Purkinje cell lineage cells in mouse and human. Subtypes are indicated with colours on top. b, UMAPs of cells from Purkinje cell lineage aligned across species.

The integration of Purkinje cell data across species lends some support to the presence of all 4 subtypes in human (Reviewer Fig. 1.1b), but the uncertainty remains, possibly due to differences in subtype prevalence in human. The latter is supported by the newly added analyses of an independent snRNA-seq dataset and spatial data for the human 12 wpc cerebellar primordium in the revised manuscript. Specifically, we made use of the human cerebellum dataset by Aldinger et al. 2021¹, which includes 9-20 wpc cerebellum samples. We filtered the Aldinger datasets for high-quality barcodes of the Purkinje cells (Reviewer Fig. 1.2), which enabled detection of all four developmental Purkinje cell subtypes (Fig. 2e). We then calculated the ratio of Purkinje cell numbers in early-born and late-born subtypes in the Aldinger dataset and our datasets, revealing the increased relative abundances of the early-born subtype cells in human compared to mouse and opossum (Fig. 2e). Our spatial dataset of the human cerebellum at 12 wpc, a stage when Purkinje cell generation is completed (Haldipur et al. 2022)⁷, provides further support for this observation (Fig. 2f,g). In the cerebellum section profiled, we found that the majority of the developing Purkinje cells express *FOXP1* and/or *RORB*, whereas *EBF2*, *CDH9* and *ETV1* mark restricted subsets of Purkinje cells.

Quantification of Purkinje cell numbers in the spatial dataset, after imputing subtype labels using the 11 wpc snRNA-seq data, confirmed the observed shift in Purkinje subtype prevalence in human.

In conclusion, all four Purkinje subtypes, which we identified in mouse and opossum, are present in the human cerebellum. The sampling differences, low relative abundances of the late-born subtypes, and larger effects from biological differences between batches in human (e.g., genetic background, exact developmental stage) compared to the other studied species, likely interfere with Purkinje subtype distinction in our human snRNA-seq dataset.

Reviewer Fig. 1.2. Re-analysis of Purkinje cell data from Aldinger et al. 2021¹. Purkinje cell barcodes were extracted from the original dataset, the data was integrated using LIGER, and clustered. Clusters that contain likely doublets/contaminated cells were identified based on high numbers of transcripts and detection of markers of other cell types (*PAX2*, *PAX6*, *SOX2*). After re-integration of the data, the four subtypes shown in Fig. 2e were distinguished based on expression of Purkinje subtype markers.

Changes to the manuscript

- Main text, page 9, lines 212-227:

Based on key marker genes and the correlation of orthologous variable gene expression, we identified the same four developmental Purkinje subtypes in opossum (Fig. 2a,c,d). In human, we reliably distinguished two subgroups (*EBF1/2*-low and -high), but the patterned expression of subtype markers indicated additional diversity (Fig. 2a,c,d; Extended Data Fig. 3l). To investigate this further, we reanalysed an independent snRNA-seq dataset of human fetal (9-20 wpc) cerebellum¹ (Methods), and explored the expression of Purkinje subtype markers in our 12 wpc spatial dataset. These analyses confirmed the presence of all four Purkinje subtypes in the human fetal cerebellum (Fig. 2e,f, Extended Data Fig. 3l,n). In the 12 wpc Purkinje cell compartment (*SKOR2*), early-born Purkinje cell markers *FOXP1* and *RORB* exhibit widespread expression, whereas the late-born Purkinje cell marker *EBF2* was detected in restricted spatial domains, wherein *CDH9*-positive cells located medially and *ETV1*-positive cells laterally, in line with mouse Purkinje cell patterning (Figure 2b,f, Extended Data Fig. 3k). Comparison of the prevalence of early-born and late-born Purkinje cells in the three species revealed a shift in subtype ratios in human, with increased numbers of early-born Purkinje cells detected in human fetal samples from the two snRNA-seq datasets, and in the 12 wpc spatial dataset (Fig. 2e,g). In sum, while Purkinje cell patterning is conserved among mammalian species, the subtype ratios changed in the lineage leading to humans, facilitated by mechanisms that likely involve an augmented generation of early-born Purkinje cells.

- Discussion, page 23, lines 541-548:

The increased abundance of human Purkinje cells is biased towards the early-born subtypes, which in the mouse bear similarities to the adult Aldoc-positive subtypes enriched in the posterior and flocculonodular lobes of the cerebellar hemispheres. Purkinje cells in these regions project to the lateral (dentate) deep nuclei that in the human lineage expanded by selective increase in the numbers of the large-bodied subtype of glutamatergic neurons^{8,9}. Thus, it is tempting to speculate that the biased expansion of the Purkinje cells and large-bodied glutamatergic neurons in the lateral nuclei coincided during the course of human evolution. Additionally, adaptations in these areas have been suggested to support cognitive functions in humans¹⁰.

- Figures

Fig. 2. Spatiotemporally defined Purkinje cell subtypes.

e, Top: UMAP of 14,246 human 9-20 wpc Purkinje cells from a published dataset¹ with cells coloured by subtype. Bottom: ratio of early-born to late-born Purkinje cell numbers in fetal samples. Biological replicates are shown with dots, and lines indicate the median of a stage (9 and 11 wpc). Corresponding developmental stages (Fig. 1a) are shown for mouse and opossum. **f**, Detection of Purkinje cell markers in the 12 wpc human cerebellum by smFISH. The same coronal section is shown in all panels; close-up views are indicated with rectangles; arrows point to double-positive regions. **g**, Mapping of the early- and late-born Purkinje cells in the 12 wpc human cerebellum by alignment of the smFISH data with 11 wpc snRNA-seq data. The pie chart indicates the percentage of cells by group. EB, early-born; LB, late-born.

Extended Data Fig.3. Atlas of the VZ cell types.

k, Spatial distribution of Purkinje subtypes in E15.5 mouse cerebellar primordium based on RNA *in situ* hybridisation data¹¹ for subtype marker genes. Medial and lateral sagittal sections are shown. **l**, UMAPs showing expression of key marker genes in the subtype-assigned Purkinje cells in our mouse, opossum, and human datasets, and in the reanalysed Aldinger *et al.* 2021¹ dataset. Scaled expression of *EBF1* and *EBF2* is shown at the left to highlight the combinatorial patterns; scaled expression of subtype markers *RORB*, *FOXP1*, *CDH9* and *ETV1* is shown at the right with each cell coloured according to the gene that has the highest scaled expression level. For visualisation purposes, the scales were capped at 95th quantile for *RORB*, *FOXP1*, *CDH9*, *EBF1* and *EBF2*, and 99th quantile for *ETV1*. **n**, Dot plot showing expression of key marker genes in the Purkinje subtypes in the reanalysed Aldinger *et al.* 2021¹ dataset.

- Methods, page 34, lines 813-826, and page 35, lines 841-850

External snRNA-seq datasets

Processed data and annotations of adult mouse cerebellum snRNA-seq dataset by Kozareva *et al.*¹² were downloaded from https://singlecell.broadinstitute.org/single_cell/study/SCP795. Processed data and annotations of human fetal cerebellum sn/scRNA-seq dataset by Aldinger *et al.*¹ were downloaded from <https://www.covid19cellatlas.org/aldinger20/>. From the Aldinger *et al.* dataset, we extracted barcodes annotated as Purkinje cells (n=25 711), and used LIGER¹³ (k=10) to perform batch correction and integrate data across stages, assigning batches by *sample_id* and *experiment*. We performed Leiden clustering (leidenAlg¹⁴, version 1.0.5; resolution 0.6), and identified clusters that contained barcodes that likely represent doublets, based on the higher number of counts in these clusters and co-expression of markers of several cell types (*PAX2* for interneurons, *PAX6* for granule cells, *SOX2* for astroglia lineage cells). After excluding the doublet clusters, we performed LIGER integration (k=10) and Leiden clustering (resolution 1.2) on the remaining barcodes (n=14,246, 55% of the barcodes annotated as Purkinje cells in the original dataset). We annotated 15 of the obtained 17 clusters based on the expression of Purkinje subtype markers *RORB*, *FOXP1*, *CDH9*, *ETV1*, *EBF1* and *EBF2*. 732 barcodes (2 clusters, 5.1% of the barcodes) were not assigned to a subtype.

Quantification of cell type abundances and ratios

For Fig. 2e, we calculated the ratio of cell numbers of the early-born Purkinje subtypes (*FOXP1*, *RORB*) and the late-born subtypes (*CDH9*, *ETV1*) in each biological replicate. We only included samples that met the following criteria: (i) they come from fetal stages when Purkinje cell generation is complete (9-20 wpc in human), (ii) the

relative abundances of *Purkinje_maturing* state cells (which were not separated into subtypes) among the *Purkinje_defined* and *Purkinje_maturing* cells is below 5% (our dataset; Extended Data Fig. 3i), (iii) at least 50 subtype-assigned cells are present. In the case of the Aldinger *et al.*¹ dataset we excluded samples that only contained the hemisphere or vermis (n=2), which altogether resulted in the inclusion of 9 samples representing 8 fetal stages (2 replicates for 14 wpc, and 1 replicate for each other stage). For subtype relative abundances presented in Extended Data Fig. 3i,r, 4j and 5h, we required at least 50 cells of the respective cell states to be present in a sample.

MINOR COMMENTS

Minor comment 1: Please discuss why there are missing types before 17 pcw in human where there should be less chance of sampling bias.

(This comment is related to Reviewer 4 Major comment 4, page 37).

Compared to the mouse and opossum datasets, the human dataset lacks the following cell (sub)types: *ependymal* cells, *isth_N_SLC5A7*, *MBO* (midbrain-originating) cells, and *GC_diff_2_OTX2*. Each of these cases is discussed in detail below. Additionally, the differences in Purkinje subtype mapping are discussed above (Reviewer 1 Major Comment 3, page 5).

1. Ependymal cells have low abundance in the mouse (57 cells) and opossum (160 cells) datasets. These cells are likely present in the human dataset in the subcluster “orig.cl_38_9” (Supplementary Table 3). However, this subcluster contains cells from other types, could not be further divided due to low number of cells (114), and was therefore not assigned to a cell type.
2. The subtype *isth_N_SLC5A7* is the least abundant subtype among the isthmic nuclei neurons in the mouse (218 cells) and opossum (246 cells) datasets, and our inability to distinguish this subset in the human dataset could be related to their low numbers. Additionally, the isthmic nuclei neurons are situated at the dissection border (close to the isthmus), implying that variations in dissections could influence their sampling.
3. *MBO* (midbrain-originating) cells were not distinguished in the human dataset, but expression of markers for this cell type was detected in some of the cells annotated as *NTZ_mixed* (orig.cl_16_1, Supplementary Table 3), suggesting that these cells are present in the human cerebellar anlage, but their numbers are relatively low.
4. The subtype *GC_diff_2_OTX2* (*KCNIP4* in the initial manuscript) has the lowest abundance among the differentiating granule cell state *GC_diff_2* (Extended Data Fig. 5h). This subtype best matches the adult subtype enriched in the nodulus (Extended Data Fig. 5k), suggesting that it might be unevenly distributed in the developing cerebellum as well. Given that we only sampled fragments of the human cerebellum from 17 wpc onwards, it remains possible that this subtype is undersampled in our human dataset.

We additionally explored the Aldinger et al. 2021¹ dataset for the types that we did not detect in our human dataset.

1. Ependymal cells are detected in the Aldinger et al. dataset and express *SPAG17* (Reviewer Fig. 1.3a). The disparate coverage of ependymal cells between our dataset and the Aldinger et al. dataset is likely related to sampling differences, such as variance in the developmental stages covered (Reviewer Fig. 3.1a, page 14) and possible deviations in the specific regions included in the dissected tissues, e.g., our samples mostly excluded the choroid plexus and the roof of the 4th ventricle.
- 2.-3. We did not distinguish cells from categories *isth_N_SLC5A7* and *MBO* in the Aldinger et al. dataset (Reviewer Fig. 1.3b). We note that in our mouse and opossum datasets, these cells are mostly sampled at embryonic and early fetal stages, which have limited coverage in the Aldinger et al. dataset.
4. After exclusion of likely doublets/contaminated cells from the Aldinger et al. data, we were able to detect a subgroup of differentiating granule cells that show *OTX2* expression (*GC_diff_2_KCNIP4/OTX2*; Reviewer Fig. 1.3c). To accurately characterise the differentiating granule cell subtypes in human, unbiased sampling of human late fetal and early postnatal cerebellar tissues is required, given that most *GC_diff_2* cells are covered by stages E17.5-P14 in our mouse dataset and P14-P60 in our opossum dataset. The respective stages in human (17 wpc - toddler) are only partially covered by the Aldinger et al. dataset.

Finally, we looked for the cell (sub)types missing in our human snRNA-seq data in the newly added spatial dataset of the human 12 wpc cerebellum.

1. Our spatial gene panel includes *SPAG17* that marks ependymal cells in the mouse and opossum snRNA-seq datasets (Extended Data Fig. 6b) as well as in the Aldinger et al. human dataset (Reviewer Fig. 1.3a). Unfortunately this was not enough to unequivocally identify ependymal cells in the very cell dense ventricular zone of the human 12 wpc cerebellar primordium (Reviewer Fig. 1.4a).

- We detected cells co-expressing *SLC17A6* and *SLC5A7* in the 12 wpc spatial dataset (Extended Data Fig. 4m). These cells are mostly located in the medial part of the NTZ and could represent the *isth_N_SLC5A7*.
- Midbrain-originating cells (*MBO*) are located anteriorly in the developing mouse cerebellum (Reviewer Fig. 1.4b), and we did not track these in the posterior section of the human 12 wpc cerebellum.
- Our snRNA-seq data suggests that *KCNIP4* expression is less variable among the *GC_diff_2* cells in human than in mouse, whereas differential expression of *OTX2* might be more stable across species (Extended Data Fig. 5i). The spatial data from the 12 wpc cerebellum supports this notion – we detected *KCNIP4*-positive granule cells at various locations along the mediolateral axis in the developing cerebellum, whereas *OTX2*-positive granule cells were mostly located to the domain closest to the rhombic lip (Extended Data Fig. 5j). We therefore renamed this category of cells as *GC_diff_2_OTX2* in the revised manuscript. We confirm that our inability to distinguish this group of cells in the human snRNA-seq dataset is related to their relatively low numbers and/or sampling biases, given that a spatially restricted population of *OTX2*-positive differentiating granule cells are detected in the human 12 wpc cerebellum by smFISH.

Reviewer Fig. 1.3. Exploration of the Aldinger et al. dataset for the subtypes that were not discerned in the human dataset produced in this study. The Aldinger et al. dataset was subsetting and re-integrated using LIGER. **a**, Ependymal cells and choroid plexus epithelial cells are present among the glial cells, although some barcodes represent likely doublets/contaminated cells, given that they show high numbers of transcripts, and expression of neuronal markers (*SLC17A6*, *GAD1*). **b**, *isth_N_SLC5A7* and *MBO* (*LEF1*) cells were not discerned among the *SLC17A6*-expressing glutamatergic cells. **c**, Among the granule cells and UBCs, removal of clusters that contain likely doublets/contaminated cells (*PAX2*, *SKOR2*, *SLC1A3*), enabled detection of a subgroup of differentiating granule cells that express *OTX2*.

Reviewer Fig. 1.4. a, Human 12 wpc cerebellum smFISH data for *SPAG17*, a marker of ependymal cells. **b**, Spatial distribution of midbrain-originating cells (MBO) in mouse E15.5 cerebellar primordium based on RNA *in situ* hybridisation data¹¹ for marker genes. Anterior and posterior coronal sections are shown. Arrows indicate the MBO stream.

In the revised manuscript, we have added a paragraph to the Methods to describe all cases when a category was not detected in all three species (including cases when a category is missing in the mouse or opossum dataset). We base these observations on our data only, and hope that the Reviewer agrees that exploration of the Aldinger dataset did not change our conclusions. Additionally, for the *isth_N_SLC5A7* and *GC_diff_2_OTX2* category cells, we include in the manuscript the spatial expression patterns of the respective markers in the 12 wpc human cerebellum, which provides an orthogonal approach for the detection of these subtypes in human (Extended Data Fig. 4m and 5j).

Changes to the manuscript

- Main text, page 10, lines 237-244:

Although the *SLC5A7*-marked subtype was not detected in the human snRNA-seq dataset, we observed cells co-expressing *SLC5A7* and *SLC17A6* in the 12 wpc cerebellum by smFISH (Extended Data Fig. 4m). *...* Differentiating GCs clustered into early and late populations, and in mouse and opossum we additionally detected a distinct *OTX2*-expressing subset (Extended Data Fig. 5f,g). The latter was not distinguished in the human snRNA-seq dataset due to sampling biases, given that we detected *OTX2*-expressing granule cells by spatial mapping in the domain proximal to the rhombic lip in the 12 wpc cerebellum (Extended Data Fig. 5j).

- Methods, page 33, lines 785-810:

Some annotation categories were not detected in all three species (Extended Data Fig. 2c). Out of these, many involve contaminating cell types located at the dissection borders: *progenitor_RP* (roof plate), *motorneuron*, and *neural_crest_progenitor* groups detected only in mouse, *progenitor_MB* and *MB_neuroblast* (midbrain) detected in opossum, *isthmic_neuroblast* detected in human and opossum, and *GABA_MB* (midbrain) detected in human. Some categories were not detected in human likely due to their overall low numbers: *isth_N_SLC5A7* is the least abundant subtype among the isthmic nuclei neurons in the mouse and opossum datasets (less than 250 cells) and these cells are situated at the dissection border; ependymal cells have low abundance in the mouse (57 cells) and opossum (160 cells) datasets, and are likely present in the human subcluster “orig.cl_38_9” (Supplementary Table 3) that, however, also contains cells from other types and was therefore not assigned to a cell type; *MBO* (midbrain-originating cells) were not distinguished in the human dataset, but markers of this cell type were expressed among some of the cells annotated as *NTZ_mixed* (“orig.cl_16_1”, Supplementary Table 3). Sampling of tissue fragments in human could be the reason why we did not distinguish *GC_diff_2_OTX2* in this species. The lower resolution of Purkinje subtype mapping in human could be related to the differences in Purkinje cell sampling, developmental dynamics, and/or subtype prevalence. In the mouse dataset, we did not distinguish subtype *progenitor_VZ_anterior*, likely due to limited resolution, given that it was possible to identify this subpopulation based on snATAC-seq data¹⁵. We also did not detect *preOPCs* in the mouse dataset: this population was mostly sampled from a single stage in human (17 wpc) and opossum (P14), and is in general expected to be scarce in the cerebellum given that more than 90% of oligodendrocytes in the mouse cerebellum originate in the ventral hindbrain¹⁶. The presence of group *glutamatergic_uncertain* in the human dataset only is due to inclusion of data from adult samples microdissected from deep nuclei region. Similarly, inclusion of these samples could underlie the unique detection of oligodendrocyte progenitors subtypes *OPC_early* and *OPC_late* only in the human dataset. Nevertheless, separation of human OPCs into subtypes is supported by previous studies¹⁷. It remains unclear if the human-unique categories *glut_DN_maturing*, *GC_diff_1_early* and *GC_diff_1_late*, and the opossum-unique categories *interneuron_MEIS2* and *ependymal_progenitor* are a result of biological or technical variation.

Minor comment 2: Human and mouse gains could be inflated based on lower detection of genes in opossum using 10x v2 data (i.e. these should be human or mouse losses not gains). Since you have some v3 opossum data, what is the effect of using v2 vs. v3 on classifying expression changes?

We agree that the use of different Chromium versions for data generation is a confounder in our study. However, most of the opossum dataset was generated using the more sensitive v3 chemistry (Extended Data Fig. 1a). Specifically, 24% of the cells in the mouse dataset, 53% of the cells in the human dataset, and 90% of the cells in the opossum dataset were produced with v3 Chromium kits. Our approach to call expression differences between species is in general conservative, with a large grey zone of genes that were not classified (Extended Data Fig. 10e). To specifically study the effects of the v2 versus v3 on the detection of expression changes, we focussed on mouse and human datasets, which contain more v2 data. We selected the developmental stages that included at least 2 replicates with v2 data in both mouse and human (E13.5, E14.5, E15.5, P7, P14 in mouse and corresponding stages in human), and called expression differences using the v2 data only. We then replaced one of the human v2 replicates with a v3 replicate, called again the expression differences, and compared the numbers of genes in different categories between the calls made with v2 data only and the calls made after inclusion of v3 human data. This analysis revealed that the number of genes called as different between mouse and human remained consistent regardless of the Chromium version used (Extended Data Fig. 10h).

We cannot completely exclude the possibility that some of the genes that we call as gained in human are actually losses in mouse, and were not detected in opossum because of technical issues. However, we provide evidence that this is not the case for the majority of the calls we make by studying the expression patterns in the bulk RNA-seq data, which supports the classification of expression changes we made based on the snRNA-seq data (Extended Data Fig. 10i). Additionally, our analyses of functional constraints among the genes that gained or lost expression in a species, supports the results of polarisation as we find that genes with expression gains are under weaker constraint (in human population) compared to the lost genes, whereas both groups have relaxed constraints compared to the genes called as eutherian-expressed (Fig. 4f).

Changes to the manuscript

- Methods, page 53, lines 1298-1306:

We also evaluated the possible effects of using different Chromium versions (v2 and v3) for data production on classifying expression differences. For this we focussed on a subset of human and mouse datasets that have at least 2 biological replicates produced with Chromium v2 (E13.5, E14.5, E15.5, P7, P14 in mouse, and corresponding stages in human). We then replaced one of the human v2 replicates from each stage with a replicate produced with Chromium v3, and called presence/absence expression differences in the two subsets (mouse v2 : human v2, and mouse v2 : human v2/3) using the pipeline described above. Similar numbers of expression differences were detected using the two subsets (Extended Data Fig. 10h), indicating that our approach is robust to differences in Chromium versions.

- Figures

Extended Data Fig. 10. h, Fold difference in the number of genes assigned to different presence/absence categories using Chromium v2 data only or including also v3 data. Subsets of human (v2/v3 or v2 only) and mouse (v2 only) datasets were used (Methods).

Minor comment 3: Fig. 3d - Please comment on why there is a gap between differentiating and mature GCs not Purkinje cells?

The UMAPs in Fig. 3d are calculated on the basis of 50 Harmony-corrected principal components and used to illustrate the cross-species data integration (Harmony) and estimation of diffusion pseudotime (DPT) for the granule cell and Purkinje cell lineages. Although UMAP is a valuable tool for dimensionality reduction, the distances between clusters can be affected by many parameters and do not always reflect the underlying biology. Importantly, we estimated the diffusion pseudotime using the ten-dimensional diffusion map, and not the 2D UMAP. Our datasets contain numerous mature granule cells that allow these cells to cluster separately, whereas the sampling of mature Purkinje cells is more limited. Furthermore, depending on the parameters of dimensionality reduction used for the generation of 2D UMAPs, mature Purkinje cells can appear separated from the other Purkinje lineage cells (Extended Data Fig. 3c), and mature granule cells can appear connected to the other granule cell lineage cells (Reviewer Figure 3.3, page 28). Our interpretations and conclusions are not solely based on the 2D UMAPs. Specifically, to validate the pseudotime estimations, we investigated pseudotime values across cell state categories and across developmental

stages (Extended Data Fig. 9b,c), and studied the spatial patterns of transcription factors expressed at different pseudotime bins (Extended Data Fig. 9h,i). In the revised manuscript, we have added a clarification on the granule cell UMAP embedding to the Methods section.

Changes to the manuscript

- Methods, page 48, lines 1177-1179:

Of note, the UMAP embedding of granule cells (Fig. 3d, Extended Data Fig. 9a,c) shows a gap between the differentiating and defined granule cells but this 2D representation does not influence the estimation of DPT values in the ten-dimensional diffusion map.

Minor comment 4: Fig. 3h - Do genes with human-specific trajectories show greater phenotypic constraint vs. genes divergent in mouse or opossum?

We carried out the analysis suggested by the reviewer. As shown in Reviewer Fig. 1.5, there are no significant differences in the constraint metrics between the genes that show diverged trajectories in different lineages. Also, note that in order to incorporate new data and analyses to the revised manuscript, we shortened the part describing the functional constraints. We now only show the LOEUF scores for the dynamic genes (Fig. 3e) and the genes with presence/absence expression differences (Fig. 4f). Given that the constraint plots for the different trajectory conservation groups were removed, and the limitations of space, we have decided not to include the analysis on species-specific trajectory groups in the manuscript. We hope that the reviewer shares our view that it does not contribute significantly to the main findings of our study.

Reviewer Fig. 1.5. Phenotypic constraints for genes with trajectory changes assigned to different evolutionary lineages. Intolerance to functional mutations in human population (top and middle) or cell lines (bottom) for human genes with trajectory changes in granule cell (left) or Purkinje cell (right) differentiation. The arrows point towards more tolerance. Boxes represent the interquartile range, and whiskers extend to extreme values within 1.5 times the interquartile range from the box. Line denotes the median and notches 95% confidence interval for the median.

Referee #3

SUMMARY

Manuscript by Sepp et al seeks to reconstruct cerebellar development from embryonic to adult stages with a specific emphasis on evolutionary differences between opossum, mouse, and humans. Towards this goal, the authors generate almost 400,000 single cells from the developing human, mouse and opossum cerebellum across a wide range of ages. Cross-species alignment of developmental trajectories is the main focus of the analysis. Both data generation and bioinformatic analyses are well executed using well-established methods. The manuscript is well written and easy to follow, and the dataset is likely to be useful for the community.

My main reservation relates to the issue of novelty. Prior study by Aldinger et al has already performed a very similar analysis in mouse and human. It is true that manuscript by Sepp et al further extends this data by adding additional time points and another species, but the conceptual value added by the analyses seems more incremental due in part to the overall lack of novel biological insights, lessening my enthusiasm for the current study.

We thank the reviewer for the appreciation of our work and the useful comments that helped us to improve the manuscript. We believe that our work, together with the new data and analyses added during the revision, provide a major advance in the field and improve our understanding of cerebellum development, cellular diversity and evolution, as described in detail in our response to the major comments. We hope that the revised manuscript overall better highlights the novelty of our results.

MAJOR COMMENTS

Major comment 1: The novelty of the developmental trajectory is particularly lacking based on recent findings in the human and mouse cerebellum development from scRNA-seq and spatial analysis (Aldinger et al.) which is recapitulated here especially in Fig. 2 on the rhombic lip and the divergence of Purkinje cells.

We thank the reviewer for raising this concern and apologise for not properly highlighting the novelty of our work in the initial manuscript. While the Aldinger et al. 2021¹ work provided insights into the gene expression programs in the human fetal cerebellar cells, our work, for the first time, describes the conserved and diverged aspects of cell type dynamics, diversification, and gene expression patterns in the cerebellum, based on extensive comparisons across three mammalian species. To convey the novelty of our work, we discuss below some of the results of our work in light of the previous observations by Aldinger et al. 2021¹:

1. We present an unprecedented snRNA-seq atlas covering cerebellum development in three mammalian species (human, mouse, opossum). For each species, our dataset includes embryonic, fetal and postnatal samples, which is critical to account for the protracted course of development of the cerebellum. For comparison, the Aldinger dataset is limited to fetal development (Reviewer Fig. 3.1a). Our human dataset includes more than 180 thousand cells compared to less than 70 thousand reported by Aldinger and colleagues. Furthermore, when reprocessing the Aldinger dataset, we noticed that the filtering cutoff (200 UMIs) used by the authors is very lenient, given that the number of high-quality cells we identified after reprocessing using knee-point-based cutoffs was ~39 thousand (Reviewer Fig. 3.1b). For comparison, our human dataset includes ~48 thousand fetal cells (Reviewer Fig. 3.1a). The high number of cells sampled, developmental stages covered, and meticulous data processing enabled us to distinguish 65 separate cell type/state categories in the human dataset, most of which were also detected in the mouse and opossum datasets, thus allowing us to establish a consensus classification of the cellular diversity in the mammalian cerebellum. In comparison, Aldinger and colleagues reported 21 groups. Although we broadly agree with the cell type annotation in Aldinger et al. 2021¹, there are a few notable exceptions (Reviewer Fig. 3.1c). In particular, the cell cluster annotated as molecular layer interneurons (MLI) represents GABAergic deep nuclei neurons as it is marked by *SOX14* and not *PAX2*; the cluster annotated as inhibitory cerebellar nuclei neurons (iCN) comprises various glutamatergic neurons of the nuclear transitory zone, parabrachial, and noradrenergic cells, as these cells express *SLC17A6*, *MEIS2*, *LHX9*, *PAX5*, *LMX1B*, and *PHOX2B*. In the revised manuscript, we additionally provide a spatial dataset of the human 12 wpc cerebellar primordium (see Reviewer 3 Major comment 2, page 17). This new dataset, simultaneously profiling 100 selected genes, corroborates our cell type annotations derived from the snRNA-seq data.

2. Concerning the developmental trajectories of the rhombic lip (RL) derived cell types, the inclusion of the embryonic and postnatal samples in our study, represents a major advantage. First, embryonic data allowed us to map the neurogenesis and diversification of the glutamatergic deep nuclei and isthmic nuclei neurons (NTZ lineage, Extended Data Fig. 4). Second, generation of postnatal data enabled the characterisation of the developmental diversity and the full differentiation trajectory of the granule cells (Fig. 3d-h, Fig. 4a-d, Extended Data Fig. 5,

Extended Data Fig. 9, Extended Data Fig. 10a-b). Given the clear granule cell and NTZ trajectories in our dataset, we were able to explicitly distinguish the unipolar brush cells (UBCs), and even define two developmental UBC subtypes, one of which has not been described previously (UBC_HCRTR2, Extended Data Fig. 5). Aldinger and colleagues reported a partial granule cell trajectory, and could not distinguish UBCs and glutamatergic nuclei neurons based on their data. However, distinguishing these closely related cell types is important, given that they have been implicated as the origin of group 4 medulloblastoma, as also discussed by Aldinger et al. 2021¹.

3. In our manuscript, we thoroughly characterise the Purkinje cell developmental diversity (Fig. 2, Extended Data Fig. 3i-n) and differentiation trajectory (Fig. 3d-h, Extended Data Fig. 9, 10a-d). In mouse and opossum, we identified four developmental subtypes of Purkinje cells, which are specified not only by their birthdate but also birthplace, whereas we could reliably distinguish only two broader groups in human. Given that the Aldinger et al. dataset comprehensively covers the fetal stages, which is a suitable time window for capturing the diverse developmental subtypes of Purkinje cells, we include in the revised manuscript an analysis of the human Purkinje cell subtypes in this dataset. We filtered the Aldinger et al. dataset for high-quality barcodes of the Purkinje cells (Reviewer Fig. 3.2), which enabled detection of all four developmental Purkinje cell subtypes that we described in the mouse and opossum (Fig. 2e, Extended Data Fig. 3l,n). We further calculated the ratio of Purkinje cell numbers in early-born and late-born subtypes in the Aldinger dataset and our datasets, revealing the increased relative abundances of the early-born subtype cells in human compared to mouse and opossum (Fig. 2e). We used the added spatial dataset of the 12 wpc human cerebellum (Fig. 2f,g), to substantiate our findings about human Purkinje cell subtypes and their relative abundances. We believe these observations represent a major advancement compared to the conclusions by Aldinger et al. 2021¹, given that they did not detect transcriptionally distinct Purkinje cell groups.

Reviewer Fig. 3.1. Comparison of the human cerebellum datasets from this study and by Aldinger et al. 2021¹. **a**, Number of nuclei/cells across developmental stages included in the two datasets. Given the limited access to metadata, after reprocessing we were only able to assign the developmental stage for the data from the Aldinger et al. experiments 2 and 3, which collectively contain 93% of the cells in the original dataset. The numbers next to the bars indicate the number of biological replicates (individuals); asterisks denote stages where additional data from experiment 1 was included in the original dataset. **b**, Examples of plots showing cumulative number of UMIs per cell/nucleus for two batches from the Aldinger dataset. **c**, Expression of key marker genes in neuronal cell type categories reported by Aldinger and colleagues. Dot size and colour indicate the fraction of cells expressing each gene and the mean expression level scaled by species and gene, respectively. Based on the expression patterns we suggest that (i) the group annotated as MLI (molecular layer interneurons) comprises GABAergic deep nuclei neurons, given that these cells express *SOX14*; (ii) the group annotated as iCN (inhibitory cerebellar nuclei neurons) comprises various glutamatergic neurons of the nuclear transitory zone, parabrachial, and noradrenergic cells, as these cells express *SLC17A6*, *MEIS2*, *LHX9*, *PAX5*, *LMX1B*, and *PHOX2B*.

4. The major focus of our work is on comparative analyses. We established cross-species correspondences between developmental stages and transcriptomically defined cell type/state categories, evaluated the developmental dynamics of cerebellar cell types in the three species, identified conserved marker genes for all cell states present in the

developing cerebellum, and characterised conserved gene expression trajectories in Purkinje cell and granule cell differentiation. Furthermore, using different approaches, we identified genes with diverged expression patterns between the species. Together, these analyses provide a rich resource of candidate gene sets that underlie core ancestral gene expression programs of cell fate specification in the cerebellum as well as genes that contribute to phenotypic innovations of this brain structure during evolution. Importantly, these analyses were only possible given the dense sampling covering the entire development of the cerebellum in all three species. We argue that our study represents the first comprehensive large-scale evolutionary investigation of mammalian brain development, whereas previous single-cell RNA-sequencing studies of the brain were focused on developmental analyses in single species, or cross-species comparisons of the cellular and molecular architecture of the adult brain. Aldinger and colleagues somewhat compared their dataset of human cerebellum development (Aldinger et al. 2021)¹ to a mouse dataset (Vladoiu et al. 2019)¹⁸, but the analyses were limited to evaluation of cell type annotations and exploration of a few gene expression modules. Based on these analyses, the authors suggest that the human RL cells do not directly map to mouse RL cells (Aldinger et al. 2021)¹. This is in stark contrast to our results, given that we find correspondences between the different groups of RL progenitors and GCP/UBCP cells across all three mammalian species, including the marsupial opossum. We have highlighted this in the revised manuscript, and suggest in the discussion that the observed changes in the anatomy of progenitor zones might be driven by individual gene expression shifts and not the emergence of transcriptomically distinct new cell type(s). We highlight the RL and posterior ventricular zone progenitors present during late embryonic and early fetal development as progenitor groups that express relatively high levels of genes that have gained expression in the human lineage (Extended Data Fig. 11b).

We have now made every effort to improve the presentation of our findings, and hope that the novelty of our work is better conveyed in the revised manuscript.

Reviewer Fig. 3.2. Re-analysis of Purkinje cell data from Aldinger et al. 2021¹. Purkinje cell barcodes were extracted from the original dataset, the data was integrated using LIGER, and clustered. Clusters that contain likely doublets/contaminated cells were identified based on high numbers of transcripts and detection of markers of other cell types (*PAX2*, *PAX6*, *SOX2*). After re-integration of the data, the four subtypes shown in Fig. 2e could be distinguished based on expression of Purkinje subtype markers.

Changes to the manuscript

- Main text, page 9, lines 212-227:

Based on key marker genes and the correlation of orthologous variable gene expression, we identified the same four developmental Purkinje subtypes in opossum (Fig. 2a,c,d). In human, we reliably distinguished two subgroups (*EBF1/2*-low and -high), but the patterned expression of subtype markers indicated additional diversity (Fig. 2a,c,d; Extended Data Fig. 31). To investigate this further, we reanalysed an independent snRNA-seq dataset of human fetal (9-20 wpc) cerebellum¹ (Methods), and explored the expression of Purkinje subtype markers in our 12 wpc spatial dataset. These analyses confirmed the presence of all four Purkinje subtypes in the human fetal cerebellum (Fig. 2e,f, Extended Data Fig. 31,n). In the 12 wpc Purkinje cell compartment (*SKOR2*), early-born Purkinje cell markers *FOXP1* and *RORB* exhibit widespread expression, whereas the late-born Purkinje cell marker *EBF2* was detected in restricted spatial domains, wherein *CDH9*-positive cells located medially and *ETV1*-positive cells laterally, in line with mouse Purkinje cell patterning (Figure 2b,f, Extended Data Fig. 3k). Comparison of the prevalence of early-born and

late-born Purkinje cells in the three species revealed a shift in subtype ratios in human, with increased numbers of early-born Purkinje cells detected in human fetal samples from the two snRNA-seq datasets, and in the 12 wpc spatial dataset (Fig. 2e,g). In sum, while Purkinje cell patterning is conserved among mammalian species, the subtype ratios changed in the lineage leading to humans, facilitated by mechanisms that likely involve an augmented generation of early-born Purkinje cells.

- Discussion, page 23, lines 541-552:

The increased abundance of human Purkinje cells is biased towards the early-born subtypes, which in the mouse bear similarities to the adult *Aldoc*-positive subtypes enriched in the posterior and flocculonodular lobes of the cerebellar hemispheres. Purkinje cells in these regions project to the lateral (dentate) deep nuclei that in the human lineage expanded by selective increase in the numbers of the large-bodied subtype of glutamatergic neurons^{8,9}. Thus, it is tempting to speculate that the biased expansion of the Purkinje cells and large-bodied glutamatergic neurons in the lateral nuclei coincided during the course of human evolution. Additionally, adaptations in these areas have been suggested to support cognitive functions in humans¹⁰. /.../ Considering the apparent absence of new transcriptomically distinct cell types in the human cerebellum, we propose that the previously observed alterations in progenitor zones' anatomy¹⁹ may be attributed to gene expression changes within the mammalian-shared cell types.

- Figures

Fig. 2. Spatiotemporally defined Purkinje cell subtypes.

e, Top: UMAP of 14,246 human 9-20 wpc Purkinje cells from a published dataset¹ with cells coloured by subtype. Bottom: ratio of early-born to late-born Purkinje cell numbers in fetal samples. Biological replicates are shown with dots, and lines indicate the median of a stage (9 and 11 wpc). Corresponding developmental stages (Fig. 1a) are shown for mouse and opossum. **f**, Detection of Purkinje cell markers in the 12 wpc human cerebellum by smFISH. The same coronal section is shown in all panels; close-up views are indicated with rectangles; arrows point to double-positive regions. **g**, Mapping of the early- and late-born Purkinje cells in the 12 wpc human cerebellum by alignment of the smFISH data with 11 wpc snRNA-seq data. The pie chart indicates the percentage of cells by group. EB, early-born; LB, late-born.

Extended Data Fig.3. Atlas of the VZ cell types.

k, Spatial distribution of Purkinje subtypes in E15.5 mouse cerebellar primordium based on RNA *in situ* hybridisation data¹¹ for subtype marker genes. Medial and lateral sagittal sections are shown. **l**, UMAPs showing expression of key marker genes in the subtype-assigned Purkinje cells in our mouse, opossum, and human datasets, and in the reanalysed Aldinger *et al.* 2021¹ dataset. Scaled expression of *EBF1* and *EBF2* is shown at the left to highlight the combinatorial patterns; scaled expression of subtype markers *RORB*, *FOXP1*, *CDH9* and *ETV1* is shown at the right with each cell coloured according to the gene that has the highest scaled expression level. For visualisation purposes, the scales were capped at 95th quantile for *RORB*, *FOXP1*, *CDH9*, *EBF1* and *EBF2*, and 99th quantile for *ETV1*. **n**, Dot plot showing expression of key marker genes in the Purkinje subtypes in the reanalysed Aldinger *et al.* 2021¹ dataset.

- Methods, page 34, lines 812-826, and page 35, lines 841-850

External snRNA-seq datasets

Processed data and annotations of adult mouse cerebellum snRNA-seq dataset by Kozareva *et al.*¹² were downloaded from https://singlecell.broadinstitute.org/single_cell/study/SCP795. Processed data and annotations of human fetal cerebellum sn/scRNA-seq dataset by Aldinger *et al.*¹ were downloaded from

<https://www.covid19cellatlas.org/aldinger20/>. From the Aldinger *et al.* dataset, we extracted barcodes annotated as Purkinje cells (n=25 711), and used LIGER¹³ (k=10) to perform batch correction and integrate data across stages, assigning batches by *sample_id* and *experiment*. We performed Leiden clustering (leidenAlg¹⁴, version 1.0.5; resolution 0.6), and identified clusters that contained barcodes that likely represent doublets, based on the higher number of counts in these clusters and co-expression of markers of several cell types (*PAX2* for interneurons, *PAX6* for granule cells, *SOX2* for astroglia lineage cells). After excluding the doublet clusters, we performed LIGER integration (k=10) and Leiden clustering (resolution 1.2) on the remaining barcodes (n=14,246, 55% of the barcodes annotated as Purkinje cells in the original dataset). We annotated 15 of the obtained 17 clusters based on the expression of Purkinje subtype markers *RORB*, *FOXP1*, *CDH9*, *ETV1*, *EBF1* and *EBF2*. 732 barcodes (2 clusters, 5.1% of the barcodes) were not assigned to a subtype.

Quantification of cell type abundances and ratios

For Fig. 2e, we calculated the ratio of cell numbers of the early-born Purkinje subtypes (*FOXP1*, *RORB*) and the late-born subtypes (*CDH9*, *ETV1*) in each biological replicate. We only included samples that met the following criteria: (i) they come from fetal stages when Purkinje cell generation is complete (9-20 wpc in human), (ii) the relative abundances of *Purkinje_maturing* state cells (which were not separated into subtypes) among the *Purkinje_defined* and *Purkinje_maturing* cells is below 5% (our dataset; Extended Data Fig. 3i), (iii) at least 50 subtype-assigned cells are present. In the case of the Aldinger *et al.*¹ dataset we excluded samples that only contained the hemisphere or vermis (n=2), which altogether resulted in the inclusion of 9 samples representing 8 fetal stages (2 replicates for 14 wpc, and 1 replicate for each other stage). For subtype relative abundances presented in Extended Data Fig. 3i,r, 4j and 5h, we required at least 50 cells of the respective cell states to be present in a sample.

Major comment 2: Because many of the analyses of cell states are derived from single cell data alone, it will be important to validate some of the newly discovered cell states with FISH/IHC in human tissue.

We took several approaches to address this comment. First, we used multiplexed smFISH to spatially map the marker genes of the cell type categories that we identified based on snRNA-seq data. We performed the smFISH on a section of 12 wpc human cerebellum and concomitantly detected spatial expression patterns of ~70 selected marker genes. The results of these analyses are integrated across several figures in the revised manuscript (Fig. 1, 2, 4, Extended Data Fig. 3-6, 8, 11) and provide strong support for our snRNA-seq-based annotations. The added spatial dataset also allowed us to strengthen several of our findings related to the developmental Purkinje cell subtypes (Fig. 2f,g), the “early” subtype of interneurons (Extended Data Fig. 3q), and the existence of the GCP/UBCP cells (Extended Data Fig. 5e), *OTX2*-expressing (*KCNIP4* in the initial manuscript) subtype of differentiating GCs (Extended Data Fig. 5j), UBC subtypes (Extended Data Fig. 5l), and progenitor subtypes (Extended Data Fig. 6i). Furthermore, we aligned the acquired spatial dataset with our snRNA-seq data to transfer the cell type labels, and leveraged the spatial data to validate some of the genes with presence/absence expression differences between species (Extended Data Fig. 11f), as highlighted in our response to Reviewer 3, Major comment 5 on page 24.

Second, we used immunohistochemistry to validate the novel UBC-like subtype (*UBC_HCRTR2*) that we identified based on snRNA-seq data. Given that this UBC subtype was detected in all mammalian species studied, we conducted this experiment using mouse cerebellum sections. We demonstrate that cells co-expressing *LMX1A*, *EOMES* and *HCRTR2* are indeed present in the granule cell layer and exhibit a dendritic brush (Extended Data Fig. 5m,n).

Third, we include a few additional panels showing the expression patterns of marker genes in the mouse using the Allen Developing Mouse Brain Atlas¹¹. These map parabrachial and noradrenergic cells at E15.5 (Extended Data Fig. 3f), Purkinje cell subtypes at E15.5 (to provide a comparison for the human 12 wpc data; Extended Data Fig. 3k), and RL/NTZ cell states at E13.5 (Extended Data Fig. 4e). We argue that for the conserved cell states and types, mouse data can also be used for validation.

Overall, our findings from all approaches strengthen the validity of our snRNA-seq-based annotations and provide additional insights into the spatial distribution of cerebellar cell types.

Changes to the manuscript

- Main Text

Page 4, lines 88-92: To validate the annotation, we performed multiplexed single molecule fluorescence *in situ* hybridisation (smFISH) in the 12 weeks post conception (wpc) human cerebellum and aligned the obtained spatial data with our snRNA-seq data (Methods). This allowed us to map the spatial expression patterns of >70 marker genes and locate the cell types within the cerebellum (Fig. 1d, Extended Data Fig. 3-6, Supplementary Table 5).

Page 10, lines 215-227: To investigate this further, we reanalysed an independent snRNA-seq dataset of human fetal (9-20 wpc) cerebellum¹ (Methods), and explored the expression of Purkinje subtype markers in our 12 wpc spatial

dataset. These analyses confirmed the presence of all four Purkinje subtypes in the human fetal cerebellum (Fig. 2e,f, Extended Data Fig. 3l,n). In the 12 wpc Purkinje cell compartment (*SKOR2*), early-born Purkinje cell markers *FOXP1* and *RORB* exhibit widespread expression, whereas the late-born Purkinje cell marker *EBF2* was detected in restricted spatial domains, wherein *CDH9*-positive cells located medially and *ETV1*-positive cells laterally, in line with mouse Purkinje cell patterning (Figure 2b,f, Extended Data Fig. 3k). Comparison of the prevalence of early-born and late-born Purkinje cells in the three species revealed a shift in subtype ratios in human, with increased numbers of early-born Purkinje cells detected in human fetal samples from the two snRNA-seq datasets, and in the 12 wpc spatial dataset (Fig. 2e,g). In sum, while Purkinje cell patterning is conserved among mammalian species, the subtype ratios changed in the lineage leading to humans, facilitated by mechanisms that likely involve an augmented generation of early-born Purkinje cells.

Page 10, lines 229-232: We distinguished five homologous subtypes of GABAergic interneurons, including an early-born type (*ZFHX4*) that is detected in the forming deep nuclei area in the 12 wpc human cerebellum, and four types that we matched to the transcriptionally-defined adult subtypes with layer-specific localizations in the mouse cerebellar cortex¹² (Extended Data Fig. 3o-t).

Page 10, lines 237-239: Although the *SLC5A7*-marked subtype was not detected in the human snRNA-seq dataset, we observed cells co-expressing *SLC5A7* and *SLC17A6* in the 12 wpc cerebellum by smFISH (Extended Data Fig. 4m).

Page 11, lines 241-244: Differentiating GCs clustered into early and late populations, and in mouse and opossum we additionally detected a distinct *OTX2*-expressing subset (Extended Data Fig. 5f,g). The latter was not distinguished in the human snRNA-seq dataset due to sampling biases, given that we detected *OTX2*-expressing granule cells by spatial mapping in the domain proximal to the rhombic lip in the 12 wpc cerebellum (Extended Data Fig. 5j).

Page 11, lines 252-255: We confirmed the presence of UBCs expressing *TRPC3*, *HCRTR2* or both in the human 12 wpc cerebellum by smFISH, and observed the brush-like phenotype of the the *HCRTR2*-positive cells in the mouse P7 cerebellum by immunohistochemistry (Extended Data Fig. 5l,m). Thus, the *HCRTR2*-expressing subset represents a previously unappreciated, mammalian-conserved UBC subtype.

Page 12, lines 264-272: The spatial mapping of progenitors (*SOX2*, *NOTCH1*, *PAX3*) in the human 12 wpc cerebellum revealed their presence not only in the VZ and RL, but also scattered in the prospective white matter (PWM) and cortical transitory zone, consistent with the marker expression patterns in the E15.5 mouse cerebellum (Fig. 1d, Extended Data Fig. 6h,i). smFISH for the markers of bipotent (*GLIS3*) and gliogenic (*TNC*) progenitors revealed reverse gradients, with *TNC* expressed highly in the ventricular cells within and proximal to the RL, and in the cortical transitory zone, while *GLIS3* was detected in the more distal VZ and the PWM (Extended Data Fig. 6i). In line with the presence of two late progenitor populations and our previous observations in the mouse¹⁵, we identified two glioblast populations in all three species, PWM glioblasts and astroblasts (Fig. S6c,d,f).

- Discussion, page 22, lines 532-533:

We found consensus subtypes for most neural cell types in the developing cerebellum and used multiplexed smFISH to track them in the 12 wpc human cerebellum.

- Figures

Fig. 1. d, Mapping of the main cerebellar cell types in the 12 wpc human cerebellum by alignment of the multiplexed smFISH data with 11 wpc snRNA-seq data.

Fig. 2. Spatiotemporally defined Purkinje cell subtypes. f and g - see page 16 in this response letter.

Extended Data Fig. 3. Atlas of the VZ cell types. **e**, Human 12 wpc cerebellum smFISH data for markers of the VZ cell types. Expression of marker genes of GABAergic deep nuclei neurons (*SOX14*), Purkinje cells (*SKOR2*, *ITPR1*) and interneurons (*PAX2*) is detected in expected domains (left). Only a few cells outside the rhombic lip and a region with artifactual signals (solid line box) are co-labelled by the markers of the parabrachial (*LMX1A*, *LMX1B*; red arrow) and noradrenergic (*PHOX2B*, *LMX1B*; black arrows) cell types in this section (right), which originates from the posterior cerebellum. This is in line with the expression of the parabrachial and noradrenergic cell markers in the anterior cerebellum in mouse, as shown in **f**. **f**, Spatial distribution of parabrachial and noradrenergic cell types in mouse E15.5 cerebellar primordium based on RNA *in situ* hybridisation data¹¹ for marker genes. Anterior and posterior coronal sections are shown. **g**, Human 12 wpc cerebellum smFISH data for markers of the interneuron “early” subtype. Cells co-expressing *PAX2* and *ZFH4* are detected in the region of the nuclear transitory zone.

Extended Data Fig. 4. Atlas of the RL/NTZ cell types.

e, **h**, Spatial distribution of RL/NTZ cell states (**e**) or glutamatergic deep nuclei and isthmic nuclei subtypes (**h**) in mouse E13.5 cerebellar primordium based on RNA *in situ* hybridisation data¹¹ for marker genes. Sagittal sections counterstained with HP Yellow are shown. Coloured arrows indicate the domains expressing markers of the different cell type/state categories; dotted arrows show the direction of the migration from the rhombic lip to the NTZ. In **h**, a schematic summary is shown in the top left panel. **i**, Human 12 wpc cerebellum smFISH data for markers of the glutamatergic deep nuclei. The locations of the regions expanded at the right are shown with rectangles on the whole section at the left. Black arrows indicate *SLC17A6*-positive glutamatergic deep nuclei neurons. Pink arrow indicates *EOMES*-positive unipolar brush cell. Insets (dashed line) show close-ups of individual cells. *LMX1A*, a marker of *glut_DN_P* is detected in a minority of glutamatergic deep nuclei neurons (1); glutamatergic deep nuclei neurons expressing *GABRA2*, enriched in *glut_DN_maturing* cells, dominate the NTZ at 12 wpc (2). **m**, Detection of cells co-expressing *SLC17A6* and *SLC5A7* in the human 12 wpc cerebellum by smFISH.

Extended Data Fig. 5. Atlas of the RL/EGL cell types. **e, j, l**, Human 12 wpc cerebellum smFISH data for markers of GC and UBC states (**e**), GC subtypes (**j**), and UBC subtypes (**l**). The locations of the regions expanded at right are shown with rectangles (solid line) on the whole section at left. Arrows indicate cells with specific expression patterns as described in the legends. Insets (dashed line) show close-ups of individual cells. In **e**, the heatmap shows the scaled density of mRNA spots in different rhombic lip compartments, as proposed by Yeung et al. in mouse²⁰ and by Haldipur et al. in human¹⁹. **m, n**, Detection of HCRTR2 in unipolar brush cells by immunohistochemistry. The HCRTR2, EOMES and LMX1A were detected by indirect immunofluorescence (**m**) or Immuno-SABER (**n**). The HCRTR2 antibodies used for immunohistochemistry were MAB52461 (**m**) and AOR-002 (**n**). Arrows point to HCRTR2-positive and -negative UBCs, as specified in the legend. Dotted circles highlight HCRTR2-positive cells with brush morphology. The fields shown are from the lobule X granule cell layer of P7 mouse.

Extended Data Fig. 6. Atlas of the glial cell types. **h**, Spatial distribution of astroglia lineage cells in mouse E15.5 cerebellar primordium based on RNA *in situ* hybridisation data¹¹ for marker genes. Sagittal sections counterstained with HP Yellow are shown. The regions in nuclear transitory zone (1) and cortical transitory zone (2) shown with rectangles at left are expanded at right, and highlight marker expression outside the VZ and RL. Coloured arrows indicate expression domains along the ventricular zone (including the VZ of the rhombic lip). **i**, Human 12 wpc cerebellum smFISH data for markers of astroglia lineage. The locations of the regions expanded at the right are shown with rectangles (solid line) on the whole section at the left. Coloured arrows indicate expression domains along the

ventricular zone (including the VZ of the rhombic lip). Black arrows indicate proliferative progenitor cells. Insets (dashed line) show close-ups of regions or individual cells. The same regions and cells are shown in top and bottom panels. *SLIT2*, a marker of RL progenitors is expressed in the the rhombic lip VZ (1); progenitors expressing *KIRREL2*, a marker of VZ progenitors, are present in the VZ and adjacent subventricular zone (2); progenitors expressing *GLIS3*, enriched in bipotent progenitors, are detected in the PWM/NTZ (3), progenitors in the forming Purkinje cell layer in CTZ express *TNC*, a marker of gliogenic progenitors (4). **j**, Human 12 wpc cerebellum smFISH data for markers of oligodendrocytes and mesodermal cell types. The locations of the regions expanded at the bottom are shown with rectangles (solid line) on the whole section at the top. Black arrows indicate example cells from different cell types.

Extended Data Fig. 8. Transcriptional programs and marker genes.

i, /.../ Inset shows smFISH signal of *SATB2* in granule cell lineage cells (*PAX6*) in 12 wpc human cerebellum. Arrows denote examples of cells co-expressing *SATB2* and *PAX6*.

- Methods, page 40, lines 959-993

Multiplexed single molecule *in situ* hybridisation data generation and analyses

The spatial mapping was performed using an available 12-week post-conception human cerebellum sample provided by the Human Developmental Biology Resource (HDBR; UK). The cerebellum was dissected, mounted in Tissue-Tek OCT compound (Sakura Finetek) and frozen in isopentane cooled to its freezing point. 10 μm coronal cryosections were collected on Molecular Cartography coverlips (Resolve Biosciences). The probeset targeted 100 genes (Supplementary Table 5) manually selected based on our snRNA-seq data to cover markers of the cerebellar cell (sub)types and states, and selected genes with presence/absence expression differences between human and mouse (see *Presence/absence expression differences*). Among the latter we selected genes that are expressed in the human respective cell type in the 11-17 wpc developmental time window based on snRNA-seq data, and for which absence in the mouse respective cell type is supported by public *in situ* hybridisation data^{11,21} (Supplementary Table 5). Molecular Cartography probe sets were designed and the data was produced as a service at Resolve Biosciences, Monnheim. smFISH data was visualised in Fiji using the PolyLux plugin (V1.9.0., Resolve Biosciences).

We performed cell segmentation based on the detected transcripts considering joint likelihood of transcriptional composition and cell morphology using the Baysor²² (v0.5.2) Julia (v1.6.4) library. The following parameters were set for each sample: “min-molecules-per-gene=1”, “min-molecules-per-cell=3”, “scale=25.0”, “scale-std=’25%’”, “estimate-scale-from-centers=true”, “min-molecules-per-segment=2”. The sampling step of Baysor was run with “new-component-weight=0.2” and “new-component-fraction=0.3”. Importantly, only segments that passed the 0.95 confidence level, as assigned by Baysor, were considered as cells. A custom R script was used to convert the Baysor output into a classical scRNA-seq raw dataset. The resulting data was integrated with the 11 wpc human snRNA-seq data (excluding cells from categories *GABA_MB* and *NTZ_mixed*) using the Python (v3.9) library of Tangram²³ (v1.0.3). For imputation of metadata, Tangram function “project_cell_annotation” was applied with default parameters. We imputed metadata separately at two levels: cell type and precisest label, which combines cell states and subtype information. Each segmented cell was assigned to the highest-scoring metadata category.

For the presentation of imputed cell type labels in Fig. 1d and Extended Data Fig. 11f, we required at least 15 transcripts and an area of 500 pixels (9.522 μm^2) per segment, and filtered out outliers that had the transcript numbers or area larger than the sum of the 3rd quantile and 1.5 times the interquartile range ($n_{\text{transcripts}} > \sim 140$; $\text{area} > \sim 7040$ pixels (134 μm^2)). The median metadata prediction score (Tangram) of the selected segments was 0.072 in case of cell type imputation and 0.050 in case of precisest label imputation. The prediction scores varied per group, for instance among the more abundant cell types (>1000 segments) the median score was the highest for the predicted Purkinje cell segments (0.084) and the lowest for GC/UBC segments (0.042). Therefore, we standardised the prediction scores per group, and further filtered for the segments with a prediction score above the 1st quantile. Additionally, for the predicted precisest labels (Purkinje subtypes; Figure 2g), we required the segments to have a concordant cell type assignment (Purkinje cell).

- Supplementary tables

Supplementary Table 5. List of genes analysed by multiplexed single molecule fluorescence *in situ* hybridisation.

Major comment 3: It seems that while the authors include data from postnatal/adult stages, but the analysis is relatively underdeveloped, and lacks biological insights.

To study the protracted course of development of the cerebellum, we sampled embryonic, fetal and postnatal stages. Postnatal samples were integrated into all analyses. These samples were particularly important for the characterisation of:

- diversity of developing granule cells (Extended Data Fig. 5);

- diversity of interneurons (Extended Data Fig. 3o-t);
- astrocyte diversity (Extended Data Fig. 6);
- full differentiation trajectory of the granule cells (Fig. 3d-h, Extended Data Fig. 9);
- identification of genes with changes in expression trajectories during granule cell differentiation (Fig. 4a-d, Extended Data Fig. 10a,b)
- identification of genes that display presence/absence expression differences: many of the identified genes are expressed at late stages (Fig. 4e-g).

Taken together, postnatal samples form an inherent part of our study, and many of our findings are reliant on them.

Major comment 4: There is a significant amount of usage of abundance estimates here with the scRNA-seq data which should be treated with caution. Especially considering library imbalance therefore the text should be revised as such with the analysis like in Fig. 1d,e Extended Data Fig7, etc. Statistical methods to support such claims would also be needed, and emerging tools are beginning to enable this for scRNA-seq data.

Single cell/nucleus RNA-seq datasets contain information on sample composition, but the estimates of cell type proportions are influenced by both biological and technical (e.g., sampling differences) sources of variability. Thus, we agree with the Reviewer that the abundance estimates should be interpreted with caution. To stabilize the abundance estimates and detect statistically credible shifts in cell type proportions, we used a statistical method that is based on Bayesian hierarchical modelling. Although independently developed, our approach is similar to the method recently developed by Phipson et al. 2022²⁴ (propeller) that performed well in an independent comparison of statistical methods for cell type composition analysis (Simmons 2022²⁵). A Bayesian framework was also implemented by Büttner et al. 2021²⁶ (scCODA) and shown to improve the performance compared to other methods, particularly when only a few biological replicates are available²⁶. We apologise if our description of the statistical method was not clear in the initial manuscript; we did our best to improve the presentation in the revised manuscript.

Briefly, for each species, stage and cell type the relative abundances were modelled using a binomial model. The true proportion of a given cell type within each species and stage was treated as an unobserved variable, which we modelled as a normal distribution. The mean of this normal distribution is drawn from a species and stage specific Student's T distribution with uninformative priors. The standard deviation is drawn from a wide exponential distribution with $\lambda = 1$. These settings set a broad prior to the model. We then test for differences using the Region Of Plausible Equality (ROPE) method. If the 95% highest density interval (HDI) of the posterior differences overlaps 0, we assume no differences in the cell type proportions between the two compared species; if the HDI does not overlap 0, we assume that the proportions are different. We additionally require at least two comparisons (out of the three: human vs mouse, human vs opossum, mouse vs opossum) to show differences in proportions.

The above methods look for differences in proportions based on discrete clusters or cell type labels. Besides these, alternative methods exist that evaluate the ratios of cells in neighbourhoods on a k-nearest neighbour graph (lochNESS²⁷, Milo²⁸, DA-seq²⁹). However, these methods heavily rely on integration of datasets and have so far been applied to within-species comparisons only. Given that even after extensive batch correction, species-effects are still present in the integrated manifolds (Extended Data Fig. 2b, d), we opted to perform the abundance comparisons at a coarse level only, and made use of our carefully curated cell type level annotations.

Taken together, in light of the Reviewer's comment, and also Referee #4 major comments 6 and 7 (page 45 and 47), we have now improved the presentation of the methodology and the results related to the comparisons of cell type abundances across species. Specifically:

- (i) We improved the methods section on the Bayes modelling of cell type abundances.
- (ii) We expanded the Main Text to include all detected differences in the relative cell type abundances between the species at the stages with representative sampling (7-11 wpc in human). We do not highlight the Purkinje cell relative abundance differences in the conclusion of the section "Atlases of cerebellum development across mammals".
- (iii) We toned down the related statements in the Summary and in the Discussion.
- (iv) In Figure 1f and Extended Data Fig. 7d,e we highlight the stages with comparable sampling across species and mark the stages where differences in cell type abundances were detected.
- (v) Extended Data Fig. 7d of the revised manuscript includes posterior difference plots for five cell types - astroglia, GABAergic deep nuclei neurons, granule cells and interneurons are now shown in addition to Purkinje cells.

Changes to the manuscript

- Summary, page 2, lines 38-40:

Our cross-species analyses revealed largely conserved developmental dynamics of cell type generation, except for Purkinje cells, where we observed an expansion of early-born subtypes in the human lineage.

- Main text, page 7, lines 159-172:

We applied hierarchical Bayesian modelling to compare the relative abundances of individual cerebellar cell types across matched developmental stages between species (Methods). We focused on early developmental stages that are less influenced by sampling differences (Fig. 1f, Extended Data Fig. 7d). The most striking difference we observed was a ~2-fold higher relative abundance of Purkinje cells in human compared to mouse and opossum at two stages when their relative abundances peak during development (8-9 wpc in human; Fig. 1f). The difference remains statistically significant even when additionally considering the VZ neuroblasts,

[This has been redacted]

(Fig. 1f, Extended Data Fig. 7d-e). /.../ Altogether, our snRNA-seq atlases provide a comprehensive view of the cerebellar cell types in mammals and show that the developmental sequence and timing of their generation have been largely conserved for at least 160 million years.

- Discussion, page 23, lines 536-538:

The most notable difference is the significantly higher relative abundance of Purkinje cells during early fetal development in human, which is possibly linked with the expansion of neuronal progenitor pools in the human cerebellum in the same time window¹⁹.

- Methods, page 35, lines 852-878:

Bayes modelling of cell type abundance differences

To test for differences in cell type abundances (Extended Data Fig. 7d), we applied a Bayesian hierarchical model that accounts for species-specific, biological and technical variability. For each species, stage and cell type the relative abundances were modelled using a binomial model. The true proportion of a given cell type within each species and stage was treated as an unobserved variable, which we modelled as a normal distribution to simulate the expected distribution of biological relative abundances of the given cell type. The mean of this normal distribution is drawn from a species and stage specific Student's T distribution with uninformative priors. The standard deviation is drawn from a wide exponential distribution with $\lambda = 1$. These settings set a broad prior to the model. Specifically, we modelled the number of cells of a certain cell type i as follows:

$$\begin{aligned} y_i &\sim \text{Binomial}(N, \alpha_i) \\ \alpha_i &\sim \text{Normal}(\alpha_{0i}, \sigma_i), 0 \leq \alpha_{0i} < 1 \\ \alpha_{0i} &\sim \text{StudentT}(1, 1.5, 1) \\ \sigma_i &\sim \exp(1), \sigma_i > 0 \end{aligned}$$

where N is the total cell count and y_i represents the relative abundances of an investigated cell type within a distinct biological replicate. The hyperparameter α_i estimates the species-specific proportion of a certain cell type. α_i is sampled from a species and stage specific normal distribution. Only biological replicates with more than 50 cells of the target cell type were subjected to the fitting. The model was fitted using RStan³⁰ (v2.19.3, R, *sampling* function; *iter*=4,000, *control*=list(*adapt_delta*=0.99)). We then computed the pairwise differences of α_0 between the species, using the simulated posterior distributions. We evaluated the 95% highest density interval (HDI³¹, *HDIInterval*, v0.2.2, R) of the results and tested for differences using the Region Of Plausible Equality (ROPE) method. If the HDI of at least two comparisons (pairwise between the three species) did not overlap 0, we assumed a difference in cell type proportion.

Of note, although independently developed, our approach is similar to the method recently established by Phipson et al. 2022²⁴ (propeller) that performed well in an independent comparison of statistical methods for cell type composition analysis²⁵. Bayesian framework was also implemented by Büttner et al. 2021²⁶ (scCODA) and shown to improve the performance compared to other methods, particularly when only a few biological replicates are available. We opted not to use scCODA²⁶, since it requires selection of a reference cell type.

- Figures:

Fig. 1. e,f, Relative cell type abundances across developmental stages in the whole datasets (e) or amongst the cerebellar cells (f). For adult human, only data from cerebellar lobes is included. /.../ In

f stages are aligned as in a, the line indicates the median of biological replicates, orange shading marks stages with representative sampling in human, and asterisks indicate differences in the relative abundances in human compared to mouse and opossum.

Extended Data Fig. 7. d, Hierarchical Bayes model analysis of differences in the relative cell type abundances across species at corresponding developmental stages. Difference in posterior (y-axis) shows modelled proportion differences between pairs of species (comparisons); 0 indicates no shift in proportions (dotted line). The modelled differences are summarised as 95% highest density intervals (HDI₉₅; lines) for each cell type at developmental stages (x-axis; depicted on top) where at least 50 cells were present. Only corresponding stages with representative sampling in human were considered. Differences in the relative abundances were called

(yellow shading) when HDI₉₅ of at least two comparisons did not overlap 0 (e.g., a human-specific change is assumed when HDI₉₅ of human vs mouse and human vs opossum comparisons does not overlap 0, and HDI₉₅ of mouse vs opossum comparison overlaps 0). **e**, Relative abundances of cells annotated as Purkinje cells or VZ neuroblasts across developmental stages.

VZ neuroblasts and Purkinje cells were analysed together to exclude the effect of possible biases in the annotation between the three species. Stages are aligned as in Fig. 1a, the line indicates the median of biological replicates, orange shading marks stages with representative sampling in human, and asterisks indicate differences in the relative abundances in human compared to mouse and opossum.

Major comment 5: The evolutionary analysis on Fig 4 is highly underdeveloped. It would be interesting to functionalize in-silico the gene divergences between the species. Specifically in Fig, 4b, the human specific genes could be analyzed with GSEA or even simpler with GO. This can also be done for Fig 4g in the context for both mouse and human. In Fig4e, how many of those genes are specific to the cell types? If not, the chart should ideally represent cell-type specific genes with significant expression that are gained in different phylogenetic branches. Candidate genes with divergent expression would have to be validated using in situ hybridization at the very least.

We performed an over-representation analysis (WebGestaltR²) on the genes that show trajectory changes (Fig. 4b) or presence/absence expression differences between species (Fig. 4e). In the revised manuscript we report the enrichments in Supplementary Table 11, and highlight the categories enriched among the genes with human-specific trajectory changes and presence/absence expression differences in the main text.

In the interest of space, we have moved the plot shown in the initial manuscript in Fig. 4e to Extended Data Fig. 10k. Our analyses of presence/absence expression differences are not focused on the cell type-specific genes, but we found that the genes that were called as expressed in a cell type in only one of the three species (human gained, mouse gained, marsupial-expressed) are more cell type-specific compared to genes that were called as expressed in a cell type in all species (therian-expressed; Extended Data Fig. 10j). However, the majority of the genes in the human gained or mouse gained categories are expressed in one or more of the other cell types in the cerebellum (Extended Data Fig. 10m). Overall, this suggests widespread repurposing of genes at the cell type level.

We used multiplexed smFISH to detect 26 genes with divergent expression patterns on a section of the human 12 wpc cerebellum. In this gene panel, we only included genes for which the absence of expression in mouse cerebellar cell types was supported by public *in situ* hybridisation data from the Allen Developing Mouse Brain Atlas¹¹ and/or GenePaint²¹ (Supplementary Table 5). To rule out any potential technical reasons for this absence of signal in mouse *in situ* hybridisation data, we prioritised genes that were detected in other cerebellar cell types and/or tissues in the mouse. Out of the 26 genes we tested in human, 22 could be confirmed as expressed in the respective cell types, 3 had overall low detection by smFISH, and in one case a gene was not detected in one of the cell types where a gain was called (Extended Data Fig. 11e,f, Supplementary Table 5). Thus, the presence/absence expression differences we identified based on snRNA-seq data, are enriched for genes that exhibit concordant patterns in *in situ* hybridisation data, providing support for the robustness and reliability of our findings.

Changes to the manuscript

- Summary, page 2, lines 42-45:

However, we also identified many orthologous genes that gained or lost expression in cerebellar neural cell types in one of the species, or evolved new expression trajectories during neuronal differentiation, indicating widespread gene repurposing at the cell type level.

- Main text, page 18, lines 417-421:

In each lineage, only a few (1-4) genes have changed trajectories in both cell types, suggesting that changes in regulatory programs are largely cell type-specific. Nevertheless, genes with human-specific changes in either granule cells or Purkinje cells share enrichments for functions related to synaptic membrane and glutamatergic synapse (FDR<0.05, Supplementary Table 11).

- Main text, page 19, lines 443-448:

Although most presence/absence expression differences were called in a single cell type, expression gains often involve genes that were already expressed in other neural cell types in the cerebellum (Extended Data Fig. 10k-m), suggesting evolutionary repurposing of genes between the cell types. Functional enrichments among the genes with expression differences include sensory perception and myofilament for genes that gained expression in human oligodendrocytes or astroglia, respectively (FDR<0.05, Supplementary Table 11).

- Main text, page 20, lines 468-484:

To substantiate the detected presence/absence expression differences, we spatially mapped 26 of these genes in the 12 wpc human cerebellum, focussing on genes for which absence of expression in mouse is supported by public *in situ* hybridisation data^{11,21} (Supplementary Table 5). Visualisation of smFISH signals and quantification of the expression levels in cells labelled based on integration with our 11 wpc snRNA-seq data, confirmed the co-expression of 22 genes with the respective cell type markers (Fig. 4j, Extended Data Fig. 11e,f). For instance, *PIEZO2*, *PLCZ1* and *DSCAM* were detected in *NOTCH1*-positive progenitors, and *CPLX4* in *PAX2*-marked interneurons. We further explored the available human immunohistochemistry data³² to map the human-present genes that show cell type-specific gene expression in the adult cerebellum. /.../ Thus, by employing orthogonal datasets, we validated a subset of the detected presence/absence expression differences.

- Methods, page 54, lines 1320-1341:

Gene ontology and pathway enrichment analyses.

For gene ontology and pathway enrichment analyses we used the WebGestaltR² package (version 0.4.4) and mouse functional databases of gene ontology and KEGG pathways as provided by WebGestalt (daily build accessed on 01.14.2019). /.../ Functional enrichments among the genes with preserved trajectories assigned to different trajectory classes (Fig. 3g, Extended Data Fig. 9e) and among the genes with trajectory changes assigned to different lineages (Supplementary Table 11) were identified by over-representation analysis against the background of dynamic orthologous genes (FDR < 0.1). Functional enrichments among the genes with presence/absence expression differences in cerebellar cell types (Supplementary Table 11) were identified by over-representation analysis against the background of 7,047 orthologous genes included in the analysis (see *Presence/absence expression differences*; FDR < 0.1). Databases *biological process*, *molecular function*, *cellular component*, and *KEGG* were used.

Multiplexed single molecule *in situ* hybridisation data generation and analyses, see page 21 in this response letter.

- Supplementary Table:

Supplementary Table 11. Gene ontology terms associated with genes that exhibit trajectory changes or presence/absence (P/A) expression differences.

- Figures

Fig. 4. j, Human 12 wpc cerebellum smFISH data for *PIEZO2*, *PLCZ1* and *DSCAM*. Co-expression with *NOTCH1* (progenitors), *KIRREL2* (VZ) or *SLIT2* (RL) is observed. The locations of the regions expanded at the right are shown with rectangles on the whole section at the left.

Extended Data Fig. 11. e, Human 12 wpc cerebellum smFISH data for genes with presence/absence expression differences between human and mouse. Asterisks denote expression changes that could not be polarised; other changes were assigned as gains in the human lineage. The locations of the regions 1-10 expanded at the right are shown with rectangles (solid line) on the whole section at the left. mRNA spots are black for the genes with differences, and coloured for the cell type markers. Arrows and insets show example cells where co-expression with the respective cell type marker(s) is detected. **f**, Quantification of the smFISH data in **e**. Cell type labels were transferred based on alignment²³ of the segmented²² smFISH dataset with the 11 wpc snRNA-seq data (Fig. 1d). Expression levels (mRNA counts) were normalised to segment area and scaled. The cell types where a difference was called for each gene. Out of the 26 genes tested, expression of 22 genes in the respective cell type(s) was confirmed by smFISH; 3 genes displayed overall low signal, and expression of one gene remained undetected in a cell type where the change was observed based on snRNA-seq data.

MINOR COMMENTS

Minor comment 1: Extended Data Fig. 1e doesn't have a legend.

The colours in Extended Fig. 1f (1e in initial manuscript) are the same as in panel a of the same figure. For clarity, we have now added a separate scale and updated the figure legend.

Changes to the manuscript

- Extended Data Figures:

Extended Data Fig. 1. f, Uniform Manifold Approximation and Projection (UMAP) of 115,282 mouse, 180,956 human and 99,498 opossum cells coloured by their developmental stage. Colours indicate the matched stages as shown in panel a and Fig. 1a. The broad neuronal lineages are shown with arrows. EGL, external granule cell layer; NTZ, nuclear transitory zone; RL, rhombic lip; VZ, ventricular zone.

Minor comment 2: In Extended Data Fig. 2 the pipeline shows fastMNN using 100 dimensions for correction. Why is the dimension set so high, especially when this is done across species so the number of orthologous genes is less than what batch correction takes into account?

We used LIGER combined with fastMNN to integrate data across species. The number of dimensions is determined already in the LIGER integration step. In LIGER, dimensionality reduction is based on integrative non-negative matrix factorization (iNMF) that results in interpretable dimensions, such that a single dimension of the space often captures a particular cell type (Liu et al. 2020)³³. In our datasets, covering cerebellum development in three species, we distinguished 25 cell types divided into 43 cell states, and for 12 cell states, we further split the cells into 48 subtypes (Fig. 1b-c, Extended Data Fig. 2c). Altogether this results in ~80 categories and for several of these categories differentiation continuums are observed. Given the complexity of our datasets, we argue that using 100 dimensions for data integration (LIGER) and correction (fastMNN) is appropriate. Of note, for cross-species data integration we used 1:1 orthologous genes detectable in all batches and variable across cells: n=6,101 for mouse/human; n=5,019 for mouse/opossum; and n=3,742 for global integration.

While somewhat arbitrarily set, the high-dimensional (k=100) LIGER embeddings have allowed data integration but leave enough space for the cells to cluster “freely”. We have observed that using more components (iNMF factors) rarely hurts the integration and data interpretation, whereas using too few components merges cell types that could be resolved with a higher number of components (Reviewer Fig. 3.3).

Reviewer Fig. 3.3. Integration of mouse data using different numbers of iNMF components (k). The major cell types (granule cells, astroglia, Purkinje cells) are well separated in the UMAP resulting from integration with $k=25$ but the discrimination of many other cell types (e.g. UBC, glutamatergic DN, isthmic nuclei neurons; denoted with the arrow) is improved by increasing the number of components.

Minor comment 3: Were there any functional categories from the GO analysis within the trajectory analysis with divergence between the species (in regards to Fig. 3f and Extended Data Fig. 9g)?

We performed an over-representation analysis on the genes that showed species-specific (*human-specific*, *mouse-specific*, *marsupial*) or *diverse* expression trajectories along the granule cell of Purkinje cell differentiation (Fig. 4a-c, Extended Data Fig. 10a-c) as described in our response to Reviewer 3 Major comment 5 (page 24). We prefer not to repeat this analysis for the groups of genes with *diverged* trajectories, which combine genes with species-specific and diverse trajectories.

Minor comment 4: Fig. 1c is extremely large with the UMAP taking up a lot of space for each individual species, is there an issue with integrating the entire dataset? The major cell types should still be recapitulated?

We prefer to present the individual species UMAPs in Fig. 1c, since these best display the dataset in two-dimensional space and have the highest level of resolution. The latter is in line with the high number of cell state markers that are unique to a species (Fig. 3b,c) and previous comparative studies, where the resolution of alignment was shown to be limited by species variation (Bakken et al. 2021)³⁴.

Although not optimally presented in the initial manuscript, we did integrate the entire dataset and used the 100-dimensional global embedding to estimate the pseudoages and stage correspondences (Extended Data Fig. 2f). To better highlight this analysis and facilitate visualisation of the integrated dataset, we now additionally provide

- (i) plots with global 2D UMAPs coloured by species or broad cell type lineage (Extended Data Fig. 2d);
- (ii) 3D UMAP accessible in the web resource (<https://apps.kaessmannlab.org/sc-cerebellum-transcriptome>).

These visualisations demonstrate good cross-species alignment of the broad lineages and cell types in the integrated embeddings. We opted not to display the cell type level alignment using the 2D UMAPS, given that there is overcrowding in the data-dense regions in this plot (Reviewer Fig. 3.4).

Reviewer Fig. 3.4. Integrated two-dimensional UMAP of mouse, human and opossum cells coloured by cell type.

Changes to the manuscript

- Methods, page 30, lines 728-731:

For visualisation purposes, we used the aligned embeddings to compute 2D and 3D UMAP coordinates for the cross-species integrated datasets (uwot 0.1.10, R³⁵, Extended Data Fig. 2b,d; <https://apps.kaessmannlab.org/sc-cerebellum-transcriptome>).

- Extended Data Figures:

Extended Data Fig. 2. d, Integrated UMAP of mouse, human and opossum cells coloured by species or broad cell type lineage. We used 1:1 orthologous genes detectable in all batches and variable across cells (n=3,742).

Minor comment 5: In Extended Data Fig. 8a,b what are the genes loaded into the first 2 PC since those seem to be contributing to the most of the variance? Otherwise a,b, and c aren't helpful in showing anything. This comment also applies to Fig 3a.

We thank the reviewer for this comment. In the revised manuscript, we now include Supplementary Table 6 that reports the genes loaded into the first principal components presented in Fig. 3a and Extended Data Fig. 8a-c. Additionally, we performed gene set enrichment analyses to characterise the PC-loaded genes and include examples of the top enriched terms in the respective figure panels.

Changes to the manuscript

- Main text, page 14, lines 303-307:

The two first principal components order samples by developmental stage and split glial and neuronal cells, while the third component further separates the neuronal types (Fig. 3a, Extended Data Fig. 8a, Supplementary Table 6). In a separate PCA only of neurons, the first component orders samples by developmental stage and the second separates neuronal types (Extended Data Fig. 8b,c, Supplementary Table 6).

- Methods, page 54, lines 1319-1324:

Gene ontology and pathway enrichment analyses.

For gene ontology and pathway enrichment analyses we used the WebGestaltR² package (version 0.4.4) and mouse functional databases of gene ontology and KEGG pathways as provided by WebGestalt (daily build accessed on 01.14.2019). The terms reported for principal components analyses (Fig. 3a and Extended Data Fig. 8c) were identified as enriched (FDR<0.1) by gene set enrichment analyses based on gene loadings to principal components, and databases gene ontology *biological process noRedundant* and *KEGG*.

- Figures:

Fig. 3. a, Principal components analysis based on 10,276 orthologous genes expressed in all species. Data points represent cell type pseudobulks for each biological replicate. Examples of enriched gene ontology and pathway categories for the genes loaded to components 1-3 are indicated.

Extended Data Fig. 8. c, PCA of neuronal cells based on 10,276 expressed orthologous genes across the three species. Data points represent cell type pseudobulks for each biological replicate. Examples of enriched gene ontology and pathway categories for the genes loaded to PC1 and PC2 are indicated.

- Supplementary Information:

Supplementary Table 6. Gene loadings on principal components 1-15 for the global (PC) and neurons-only (nPC) principal components analyses.

Minor comment 6: In Extended Data Fig. 8g were the TF for the cell types generated using a specific analyses method such as SCENIC? If not, why wasn't it used (or any other published TF activity methods)?

Extended Data Fig. 8i (8g in the initial version of the manuscript) shows the expression of transcription factors that are among the conserved set of cell state marker genes. We initially decided to focus on TF expression and not activity given that the estimation of TF activities would greatly benefit from profiling chromatin accessibility (snATAC-seq) in addition to gene expression (snRNA-seq). Chromatin accessibility data would allow the prediction of enhancers active in different cell states and inclusion of these regions in the gene regulatory space for TF motif discovery (as applied by SCENIC³⁶). Methods that can be applied to gene expression data only (e.g. SCENIC³⁷) are limited to detection of TF motifs around the promoters. Currently, snATAC-seq data covering cerebellum development is available only for mouse¹⁵.

The reviewer's comment motivated us to nevertheless carry out SCENIC analyses on our mouse and human snRNA-seq datasets (cisTarget databases are available for both of these species, but not for opossum) and include the results in the revised manuscript (Extended Data Figure 8i-j, Supplementary Table 9). However, we only use it to provide further support to our analyses based on conservation of TF expression patterns. We find that TF markers that show conserved expression specificity have higher TF regulon activities in the respective cell states than TF markers that are not conserved in their expression specificity (Extended Data Figure 8j). This is in line with the notion that expression specificity conservation is a relevant criterion to nominate TFs that play a role in cell type specification during cerebellum development. In-depth comparisons of TF regulons and gene regulatory networks across mammalian species would require generation of comprehensive snATAC-seq datasets for additional species besides the mouse, and this is an important avenue of research for future studies.

Of note, we also extend the analysis of conserved marker genes to cover other key genes besides the TFs, as suggested by Referee #1. See Referee #1 Major comment 1, page 2.

Changes to the manuscript

- Main text, page 15, lines 335-339:

Among all TF markers in mouse and human, conservation of expression specificity is associated with higher expression levels of their predicted target genes in the respective cell states as revealed by SCENIC³⁷ modelling

(Extended Data Fig. 8i-j, Supplementary Table 9). Thus, the identified conserved TF code provides a shortlist of candidates for the functional elucidation of the mechanisms underlying cell type specification in the cerebellum.

• Figures:

Extended Data Fig. 8. i, Expression and regulon activities of TFs that are among the conserved markers, across cell states in mouse (black), human (orange) and opossum (blue). Dot size and colour intensity indicate the fraction of cells expressing each gene and the mean expression level scaled per species and gene, respectively. Colours on top mark the TFs for which regulons (i.e. co-expression modules that are retained after pruning for the presence of TF motifs in promoter areas) were built by SCENIC in mouse (black) or human (orange) datasets. Red rectangles denote high regulon activities (standardised activity score ≥ 1). For each cell state four highest-ranking TFs are shown. Inset shows smFISH signal of SATB2 in granule cell lineage cells (PAX6) in 12 wpc human cerebellum. Arrows denote examples of cells co-expressing SATB2 and PAX6. **j**, Distribution of standardized TF regulon activity scores in mouse or human cell states for all TFs (grey), marker TFs (blue), and conserved marker TFs (red). Only TFs for which regulons were built by SCENIC are included (mouse n=447, human n=499). Scores ≥ 1 (vertical line) were defined as “high”.

• Methods, page 55, lines 1343-1353:

Inference of transcription factor regulon activities

Transcription factor activities in mouse and human were estimated using the pySCENIC^{37,38} pipeline. For computational purposes we subsampled the human dataset by keeping two (SN021 and SN105) out of the nine 8 wpc libraries (9581 out of 39 300 cells), resulting in a dataset of 151 237 cells. The whole mouse dataset was used. The cisTarget databases *500bpUp100Dw* (500 bp upstream and 100 bp downstream of the transcription start site (TSS)) and *TSS+/-10kb* (10 kb upstream and downstream of the TSS) for mouse (mm10) and human (hg38), as well as the transcription factor to motif tables (version 9) were retrieved from the cisTarget website (<https://resources.aertslab.org/cistarget/>). Gene regulatory networks were inferred using pySCENIC *grn* with the default settings. pySCENIC *ctx* was run with `min_genes = 5`, and *auCell* with the default settings. For each transcription factor, regulon activities (AUC scores) were z-scored across cell states.

- Supplementary Information:

Supplementary Table 9. Standardised transcription factor regulon activity scores across cell states in mouse and human.

Minor comment 7: In Extended Data Fig. 10 passing some of gene groups found such as the Human gained gene group though GO would help interpret the findings.

We performed an over-representation analysis (WebGestaltR²) on the genes that show presence/absence expression differences between species. In the revised manuscript we report the enrichments in Supplementary Table 11, and highlight the categories enriched among the genes that gained expression in human in the main text.

Changes to the manuscript

- Main text, page 19, line 446-448

Functional enrichments among the genes with expression differences include sensory perception and myofilament for genes that gained expression in human oligodendrocytes or astroglia, respectively (FDR<0.05, Supplementary Table 11).

- Methods, page 54, lines 1320-1341:

For gene ontology and pathway enrichment analyses we used the WebGestaltR² package (version 0.4.4) and mouse functional databases of gene ontology and KEGG pathways as provided by WebGestalt (daily build accessed on 01.14.2019). /.../ Functional enrichments among the genes with presence/absence expression differences in cerebellar cell types (Supplementary Table 11) were identified by over-representation analysis against the background of 7,047 orthologous genes included in the analysis (see *Presence/absence expression differences*; FDR < 0.1). Databases *biological process*, *molecular function*, *cellular component*, and *KEGG* were used.

- Supplementary Table:

Supplementary Table 11. Gene ontology terms associated with genes that exhibit trajectory changes or presence/absence (P/A) expression differences.

Referee #4

SUMMARY

In this manuscript, Sepp, Leiss, and coauthors present a multispecies single-cell RNA sequencing dataset from developing cerebellum, extracted from three different species: human, mouse, and opossum. The aim of the study is to describe the differences and similarities in the process of cerebellar development across species.

The authors analyze the dataset and provide a classification of cell types present at the different stages of development in the three species. First, the authors approach the problem of determining a correspondence between the cerebellar development of these species to an unprecedented level of granularity. The annotation of the different cell populations across different species is performed carefully and with attention to the literature returning proportions that seem reasonably in line with the cell type composition for mouse and developing human cerebellum established in other studies. While missing a few intermediate states, the dataset can be considered comprehensive for most practical purposes.

The manuscript provides knowledge at a different level of granularity, maybe not all immediately digestible at a first read but useful to consultation (i.e. there are many interesting summary visualization worth consulting in the supplementary figures). Overall, the resource has a tremendous value for understanding cerebellum evolution. The data per se has a high potential to be re-used for other meta-analyses. The analyses performed by the authors are of a high standard, the work of annotation and curation is excellent, and several details of the analysis are innovative and exemplar for the field.

The authors, first, approach the developmental population dynamic of different cell types. Purkinje cells in humans are identified to have a unique mode of expansion, while other cell types' developmental trajectories are revealed as non-conserved. Then, the authors propose a gene-centric evolutionary analysis of the trajectories. They identify several disease-associated genes that behave differently in human cerebellar development than mouse and opossum. This leads to the impactful conclusion that mouse models might not be as relevant for studying disruptions of cerebellar development as they are believed.

- A web resource accompanies the paper; while not perfect (e.g., integrated UMAP of the species is missing), it succeeds in making the data more available for the public.
- The only general worry I have about the manuscript is that it is not always clear which of the several points proved is an entirely unprecedented discovery or rather just a systematization/validation of something already known. For example, to my knowledge, the fact that Purkinje cells progenitors in humans go through secondary expansion in the subventricular zone (SVZ), akin to cortical neurons in the human cortex, is well-known in the field. Since my primary expertise is not in cerebellar cell type development, I suggest that an expert with complete knowledge of the literature be consulted to evaluate these aspects.

Overall, given the quality of data, well-delineated analyses, care in the annotation and coherence of the story, I believe the work is of great importance for the developmental neuroscience community, and I can forecast its impact going beyond its core-specific field and constitute a landmark study. Therefore, I am highly supportive of the publication of this work in Nature after appropriate revision.

We are grateful to the reviewer for the highly positive appreciation of our work and for their comments, which helped us to improve the manuscript.

- Regarding the web resource, we have updated the app to improve its usability. To facilitate direct comparisons between the species, we now enable simultaneous plotting of the UMAPs and dot plots of the three species. (Note that the small adjustments in cell type annotation/vocabulary that we introduced during the revision are not reflected in the web resource to keep it compatible with the preprinted version till publication.) We chose to mainly use individual species UMAPs in the online resource since these better display each dataset in two-dimensional space and have the highest level of resolution. The latter is in line with the high number of cell state markers that are unique to a species (Fig. 3b,c) and previous comparative studies, where the resolution of alignment was shown to be limited by species variation (Bakken et al. 2021)³⁴. Nevertheless, we now provide an option to view the 3D UMAP embedding via the web resource.
- We apologise for the unclarity concerning the novel discoveries of our work. We hope to have overall improved the presentation of our findings in the revised manuscript. We would also like to note that so far it has not been directly demonstrated that the SVZ progenitors in the developing human cerebellum give rise to Purkinje cells. It is generally accepted that there are two populations of “basal” (detached from the ventricular zone) progenitors in the

mammalian cerebellum: the radial gliogenic progenitors (i.e., developing Bergmann glia) that give rise to astrocytes, and bipotent progenitors in the prospective white matter that give rise to interneurons and astrocytes (Buffo and Rossi 2013³⁹, Leung and Li 2018⁴⁰, Cerrato et al. 2018⁴¹). Work from the group of Kathleen Millen has provided evidence for the expansion of the progenitor zones in the human cerebellum (Haldipur et al. 2019)¹⁹, and our work demonstrated an increase in the relative abundances of Purkinje cells during the time of their neurogenesis. Although we hypothesise that these processes might be causally linked, experimental evidence is lacking. Fate mapping, for example using human cerebellar organoids or organotypic slice cultures, will be important to establish the lineage relationships between the SVZ progenitors and the cell (sub)types in the cerebellum. These approaches have already proven to be feasible in the context of human cerebral cortex (Allen et al. 2022; He et al. 2022)^{42,43}.

MAJOR COMMENTS

Major comment 1. The authors identify the correspondence between different stages of cerebellar development based on transcription data and claim that there is no major heterochrony. While overall the approach is reasonable, it would be stronger if supported by morphological data: it would be useful to provide histological micrographs for the cerebellum in all three species and indicating homologous subdomains on different developmental timepoints, in particular for opossum where not much is available.

We agree with the reviewer. We now provide histological data for the developing opossum cerebellum across 7 stages and make direct comparisons to the cerebellar morphology in mouse and human using the data from the Allen Developing Mouse Brain Atlas and the HDBR histological atlas (<https://hdbratlas.org/histology.html>). We refer to Haldipur et al. 2019¹⁹ and Ábrahám et al. 2001⁴⁴ for additional comparisons to human cerebellar development. Importantly, the added histological data supports the stage correspondences estimated from our snRNA-seq data.

Changes to the manuscript

- Main text, page 5, lines 99-102:

The estimated stage correspondences are supported by the morphological characteristics of the developing cerebellum in the three species, and agree with the correspondences previously established by jointly considering multiple somatic organs⁴⁵ (Extended Data Fig. 2h-k).

- Methods, page 39, lines 937-956:

Histology

For cryosections, whole heads (E14.5, P1, P4, P5, P21) or dissected cerebella (P42) of opossums were mounted in Tissue-Tek OCT compound (Sakura Finetek) and frozen in isopentane cooled to its freezing point. 12 µm sagittal cryosections were collected on SuperFrost Plus slides (Thermo Scientific), fixed in 4% PFA in PBS for 12 minutes, washed 3 times with PBS and incubated in 70% ethanol at 4°C overnight. The sections were rehydrated and permeabilized in 0.2% Triton X-100 in PBS for 20 minutes and stained with DAPI and NeuroTrace 530/615 red fluorescent Nissl (both 1:300 in PBS) from the BrainStain imaging kit (Thermo Fisher Scientific) for 20 minutes at room temperature. After washing with 0.2% Triton X-100 in PBS for 3x10 minutes, the sections were mounted in Prolong Diamond Antifade Mountant (Thermo Fisher Scientific). Stitched Z-stack fluorescence images were acquired on Olympus CellSens widefield microscope equipped with 10x 0.4 NA and 20x 0.75 NA objectives, and motorized XY-stage and Z-drive. Extended Focus Imaging projection was calculated using the cellSens software.

For FFPE sections, whole heads of P4 and P14 opossums were fixed in 4% formalin and transferred to 70% ethanol for storage. After decalcification with EDTA (25% v/v), the specimens were embedded in paraffin and sectioned at 6 µm using a sliding microtome (Leica SM2010 R). Sections were stained with Heidenhain's AZAN. Microphotography was done with a LEICA camera (DFC490) mounted on a ZEISS Axioskop equipped with 1.25X and 2.5X objectives, employing the standard LEICA Application Suite (LAS X) for image capturing.

The mouse histology images are from the Allen Developing Mouse Brain Atlas¹¹. Human images are from the HDBR Atlas⁴⁶⁻⁴⁸.

- Figures

Extended Data Fig. 2. h-j. Comparison of the developing cerebellum structures in mouse (h), opossum (i) and human (j). Mouse images are from the Allen Developing Mouse Brain Atlas¹¹. The sagittal sections were stained with HP yellow or Nissl. For opossum, sagittal sections were prepared from E14.5-P21 heads and P42 cerebellum. Sections from fresh-frozen or FFPE samples were stained with DAPI and NeuroTrace Nissl, or with Azan, respectively. Human images are from the HDBR Atlas⁴⁶⁻⁴⁸. 7 wpc and 8 wpc (CS23) sagittal sections were stained with hematoxylin and eosin; *LMX1A* RNA was probed in the 18 wpc sagittal section, counterstained with fast green. Arrowheads indicate the cerebellar ventricular zone, arrows denote the rhombic lip, n and c label the NTZ and CTZ. The stages are

numbered as in Fig. 1a. At E11.5/E14.5 (mouse/opossum) the cerebellar primordium is dominated by the cell-dense neuroepithelium. NTZ and CTZ are first visible at E12.5/P1 and E13.5/P4, respectively. EGL and developing PL are discerned at E17.5/P14. P7/P21 is characterized by a thick EGL, which shrinks but is still present at P14/P42. Similarly in human, at 7 wpc the cerebellar primordium is dominated by the cell-dense neuroepithelium; CTZ is visible at 8 wpc; EGL and PL are discerned at 18 wpc. Newborns are characterized by a thick EGL, which gradually shrinks but is still present in infants after 8 months of postnatal development⁴⁴.

Major comment 2: The time correspondences proposed do not always match the correspondence between different developmental stages identified in previous publications by the same approach.

- For example, in "Gene expression across mammalian organ development" (by the same collective of authors), mouse developmental stages e13.5 and e14.5 are shown to be the most transcriptionally similar to the opossum P2 stage, while in the current manuscript, the same stages correspond to P4-P5 stage in the opossum.
- Similarly, a human embryo on 7 wpc was shown to have the highest transcriptional correlation with e12.5-e14.5 mouse embryo in the previous work, while in the current work 7 wpc human cerebellum corresponds to e11.5 mouse cerebellum.

While the differences seem to be minor, there are rapid changes in the mouse embryo with every day of development; therefore, identifying the wrong correspondence might be detrimental for further comparative analysis. This aspect should be at least discussed.

We thank the referee for raising this concern and admit that this part was not presented in enough detail in the initial manuscript. The exact stage correspondences across species depend on the sampling scheme, which were slightly different in the current study and in Cardoso-Moreira et al. 2019⁴⁵. To provide a clear overview, we have added a new panel (k) to the Extended Data Fig. 2, which shows the sampling scheme and the estimated correspondences in the two studies. Overall, the correspondences established in the two studies agree and the shifts are explained by differences in sampling. To illustrate this, we discuss the examples mentioned by the referee below.

- According to Cardoso-Moreira et al. mouse stage E13.5 corresponds to P2 in the opossum and mouse stage E14.5 corresponds to P4 in the opossum.

The current study did not include the P2 stage in the opossum and out of the included stages mouse E13.5 best matches to opossum P4 (and not to P1).

In the current study we additionally sampled P5 opossum (not included in the Cardoso-Moreira dataset) and found P5 to match mouse E14.5 better than P4 using all three approaches (Extended Data Fig. 2e-g).

- According to Cardoso-Moreira et al. human stage 7 wpc corresponds to E12.5-E13.5 in the mouse. In the Cardoso-Moreira dataset the 7 wpc age group includes Carnegie stages 19-21, and the 8 wpc age group Carnegie stages 22-23 and late 8 week.

In the current study the 7 wpc includes Carnegie stages 18-19, and the 8 wpc includes only Carnegie stage 22, i.e. both stage groups are represented by younger samples than in the Cardoso-Moreira study. This likely explains why the current study matches 8 wpc (Carnegie stage 22) to E13.5 and 7 wpc (Carnegie stages 18-19) to E11.5.

In light of the slight shifts in correspondences introduced by sampling differences, we updated the text in the revised manuscript to clearly state that the correspondences are estimated among the stages sampled in the current study.

Changes to the manuscript

- Main text, page 4, lines 97-102:

By combining these approaches, we infer, for instance, that among the stages sampled the cerebellum of a newborn human most closely resembles that of a one week old mouse and a three week old opossum (Fig. 1a). The estimated stage correspondences are supported by the morphological characteristics of the developing cerebellum in the three species, and agree with the correspondences previously established by jointly considering multiple somatic organs⁴⁵ (Extended Data Fig. 2h-k).

- Methods, page 39, lines 933-935:

We note that the estimated stage correspondences are dependent on the sampling scheme and, although they overall agree with previous studies⁴⁵, should not be interpreted as absolute best matches.

- Extended Data Figures:

Extended Data Fig. 2. k. Comparison of the sampling and stage correspondences in this study and in Cardoso-Moreira et al. 2019.⁴⁵ Human samples representing 4-8 wpc may include samples from several Carnegie stages. The correspondences estimated in both studies globally agree. The shifts are explained by differences in sampling, e.g. in this study 8 wpc in human is represented by CS22 and best matches to E13.5 in mouse, whereas in Cardoso-Moreira et al. 2019 8 wpc stage group includes samples from CS22 to late 8 week and matches to E14.5 in mouse.

* The green and grey shaded areas indicate the examples discussed above (shown to the reviewers only).

Major comment 3: Related to the previous point: looking at the correlation maps between homologous cell types coming from different species (e.g., Extended data fig. 3c, extended data fig. 5e), it is evident that the mature cell types and the progenitors at the beginning of the developmental process have more correlation across species than intermediate progenitors. It would be useful if the author could specify how this should be interpreted. E.g., is this the effect of the incomplete sampling/imperfect matching of the intermediate stages of development between species, or are the intermediate stage progenitors less conserved in evolution?

In these analyses we calculated Spearman's correlation coefficients between orthologous variable gene expression profiles. Limiting the comparisons to the variable genes is important to focus on the signals that distinguish the tightly related cell type/state categories, but we are reluctant to make conclusions about the differences in evolutionary conservation based on these restricted gene sets. Additionally, we agree with the Reviewer's suggestion that the analyses might be affected by imperfect matching between the developmental stages we sampled, and between the individual subtype/state categories defined in each species. The latter is especially true for the cell states that form a differentiation continuum (e.g., VZ neuroblasts 1-3) as we cannot exclude slight shifts in the borders of these categories between the species. We thank the Reviewer for pointing out this potentially misleading interpretation of the correlation maps, and added a note on this to the Methods section of the revised manuscript.

Changes to the manuscript

- Methods, page 46, lines 1126-1131:

Importantly, these analyses should not be used to make conclusions about the differences in evolutionary conservation of the individual categories, given that the comparisons are limited to the intersect of highly variable genes, and might be affected by imperfect matching between the developmental stages we sampled as well as between the individual subtype/state categories defined in each species. The latter is especially true for the cell states that form a differentiation continuum as we cannot exclude slight shifts in the borders of these categories between the species.

Major comment 4: Related to the sampling strategy: the authors claim to identify certain cell types (e.g., *GC_diff2_KCNIP4* in extended data fig. 5, cell type *isth_N_SLC5A7* in extended data fig. 4) that are present in mouse and opossum, but not in human. For granule cells cluster, the explanation could be in the sampling strategy (as authors indicate themselves); however, to make the study stronger in that sense; we think other datasets of cerebellar human development should be taken into consideration. E.g., are those cell types possible to find in the recently published dataset on human cerebellar development ("Spatial and cell type transcriptional landscape of human cerebellar development" Aldinger et al.)?

(This comment is related to Reviewer 1 Minor comment 1, page 8, and Reviewer 1 Major comment 3, page 5).

Compared to mouse and opossum datasets, the human dataset lacks the following cell (sub)types: *ependymal* cells, *isth_N_SLC5A7*, *MBO* (midbrain-originating) cells, and *GC_diff_2_KCNIP4/OTX2*. Additionally, in the human dataset we distinguished 2 groups of developing Purkinje cells as opposed to 4 subtypes identified in mouse and opossum. We took different approaches to elucidate the reasons why these categories were not distinguished in our human dataset. We first discuss each of these cases in the context of our own snRNA-seq dataset:

1. Ependymal cells have low abundance in the mouse (57 cells) and opossum (160 cells) datasets. These cells are likely present in the human dataset in the subcluster "orig.cl_38_9" (Supplementary Table 3). However, this subcluster contains cells from other types, could not be further divided due to low number of cells (114), and was therefore not assigned to a cell type.
2. The subtype *isth_N_SLC5A7* is the least abundant subtype among the isthmic nuclei neurons in the mouse (218 cells) and opossum (246 cells) datasets, and our inability to distinguish this subset in the human dataset could be related to their low numbers. Additionally, the isthmic nuclei neurons are situated at the dissection border (close to the isthmus), implying that variations in dissections could influence their sampling. Of note, we did detect *SLC5A7* expression in a subpopulation of human *isth_N_diff* cells (orig.cl_41_9).
3. *MBO* (midbrain-originating) cells were not distinguished in the human dataset, but expression of markers for this cell type was detected in some of the cells annotated as *NTZ_mixed* (orig.cl_16_1, Supplementary Table 3), suggesting that these cells are present in the human cerebellar anlage, but their numbers are relatively low.
4. The subtype *GC_diff_2_KCNIP4/OTX2* has the lowest abundance among the differentiating granule cell state *GC_diff_2* (Extended Data Fig. 5h). In the mouse, this subtype best matches the adult subtype enriched in the nodulus (Extended Data Fig. 5k), suggesting that it might be unevenly distributed in the developing cerebellum as well. Given that we only sampled fragments of the human cerebellum from 17 wpc onwards, it remains possible that this subtype is underrepresented in our human dataset.
5. There are several factors that likely interfere with Purkinje subtype distinction in our human snRNA-seq dataset: differences in sampling, larger effects from biological differences between batches in human (e.g., genetic background, exact developmental stage) compared to other studied species, and the low relative abundance of the late-born Purkinje cells. In the revised manuscript, we expand the analyses of human Purkinje subtypes, and use independent datasets (Aldinger et al. 2021 and our 12 wpc spatial dataset) to show that all four subtypes are present in the human cerebellum, though with shifted ratios (see below).

Next, as suggested by the Reviewer, we explored the Aldinger et al. 2021¹ dataset for the types that we did not detect in our human dataset. Although, we broadly agree with the cell type annotation provided in Aldinger et al. 2021¹, there are a few exceptions that involve GABAergic deep nuclei neurons (annotated originally as molecular layer interneurons) and glutamatergic deep nuclei neurons and other neuron types located close to/in the nuclear transitory zone (annotated originally as inhibitory deep nuclei neurons; Reviewer Fig. 4.1). We filtered the datasets for relevant cell types based on the expression of marker genes: (1) glial cells (*PAX3* and *SOX10*), (2-3) glutamatergic cell expressing *SLC17A6*, (4) granule cells and UBCs (*PAX6* and *ATOX1*), and (5) Purkinje cells (*SKOR2*). We then applied LIGER to re-integrate each of the subsets individually across the samples in the Aldinger dataset, and studied the expression of markers of the (1) ependymal cells, (2) *isth_N_SLC5A7*, (3) *MBO*, (4) *GC_diff_2_KCNIP4/OTX2*, or (5) Purkinje cell subtypes.

Reviewer Fig. 4.1. Expression of key marker genes in neuronal cell type categories reported in Aldinger et al. 2021¹. Dot size and colour indicate the fraction of cells expressing each gene and the mean expression level scaled by species and gene, respectively. Based on the expression patterns we suggest that (i) the group annotated as MLI (molecular layer interneurons) comprises GABAergic deep nuclei neurons, given that these cells express *SOX14*; (ii) the group annotated as iCN (inhibitory cerebellar nuclei neurons) comprises various glutamatergic neurons of the nuclear transitory zone, parabrachial, and noradrenergic cells, as these cells express *SLC17A6*, *MEIS2*, *LHX9*, *PAX5*, *LMX1B*, and *PHOX2B*.

1. Ependymal (*SPAG17*) and choroid plexus epithelial cells (*SPAG17*, *TTR*) are detected in the Aldinger et al. dataset (Reviewer Fig. 4.2a). The disparate coverage of these cells between our dataset and the Aldinger et al. dataset is likely related to sampling differences, such as variance in the developmental stages covered (Reviewer Fig. 3.1a, page 14) and possible deviations in the specific regions included in the dissected tissues, e.g., our samples mostly excluded the choroid plexus and the roof of the 4th ventricle.
- 2.-3. We did not distinguish cells from categories *isth_N_SLC5A7* and *MBO (LEF1)* in the Aldinger et al. dataset (Reviewer Fig. 4.2b). We note that in our mouse and opossum datasets, these cells are mostly sampled at embryonic and early fetal stages, which have limited coverage in the Aldinger et al. dataset.
4. After exclusion of likely doublets/contaminated cells from the Aldinger et al. data, we were able to detect a subgroup of differentiating granule cells that show *OTX2* expression (*GC_diff_2_KCNIP4/OTX2*; Reviewer Fig. 4.2c). To accurately characterise the differentiating granule cell subtypes in human, unbiased sampling of human late fetal and early postnatal cerebellar tissues is required, given that most *GC_diff_2* cells are covered by stages E17.5-P14 in our mouse dataset and P14-P60 in our opossum dataset. The respective stages in human (17 wpc - toddler) are only partially covered by the Aldinger et al. dataset.
5. We filtered the Aldinger et al. dataset for high-quality barcodes of the Purkinje cells (Reviewer Fig. 4.2d), which enabled detection of all four developmental Purkinje cell subtypes that we described in mouse and opossum (Fig. 2e). The Aldinger et al. dataset comprehensively covers the fetal stages, which is a suitable time window for capturing the diverse developmental subtypes of Purkinje cells. We further calculated the ratio of Purkinje cell numbers in early-born and late-born subtypes in the Aldinger dataset and our datasets, revealing increased relative abundances of the early-born subtype cells in human compared to mouse and opossum (Fig. 2e).

Reviewer Fig. 4.2. Exploration of the Aldinger et al. dataset for the cell (sub)types that were not discerned in the human dataset produced in this study. The Aldinger et al. dataset was subsetted and re-integrated using LIGER. **a**, Ependymal cells and choroid plexus epithelial cells are present among the glial cells, although some barcodes represent likely doublets/contaminated cells, given that they show high numbers of transcripts, and expression of neuronal markers (*SLC17A6*, *GAD1*). **b**, *isth_N_SLC5A7* and *MBO* (*LEF1*) cells were not discerned among the *SLC17A6*-expressing glutamatergic cells. **c**, Among the granule cells and UBCs, removal of clusters that contain likely doublets/contaminated cells (*PAX2*, *SKOR2*, *SLC1A3*), enabled detection of a subgroup of differentiating granule cells that express *OTX2*. **d**, After removing the clusters that contain likely doublets/contaminated cells (*PAX2*, *PAX6*, *SOX2*) from the Purkinje cell subset, the four subtypes shown in Fig. 2e could be distinguished. See text for details. Ast, astrocytes; OPC, oligodendrocyte progenitor cells.

Finally, we looked for the cell (sub)types missing in our human snRNA-seq data in the newly added spatial dataset of the human 12 wpc cerebellum.

1. Our spatial gene panel includes *SPAG17* that marks ependymal cells in the mouse and opossum snRNA-seq datasets (Extended Data Fig. 6b) as well as in the Aldinger et al. human dataset (Reviewer Fig. 4.2a). Unfortunately this was not enough to unequivocally identify ependymal cells in the very cell dense ventricular zone of the human 12 wpc cerebellar primordium (Reviewer Fig. 4.3a).
2. We detected cells co-expressing *SLC17A6* and *SLC5A7* in the 12 wpc spatial dataset (Extended Data Fig. 4m). These cells are mostly located in the medial part of the NTZ and could represent the *isth_N_SLC5A7*.
3. Midbrain-originating cells (MBO) are located anteriorly in the developing mouse cerebellum (Reviewer Fig. 4.3b), and we did not track these in the posterior section of the human 12 wpc cerebellum.
4. Our snRNA-seq data suggests that *KCNIP4* expression is less variable among the *GC_diff_2* cells in human than in mouse, whereas differential expression of *OTX2* might be more stable across species (Extended Data Fig. 5i). The spatial data from the 12 wpc cerebellum supports this notion – we detected *KCNIP4*-positive granule cells at various locations along the mediolateral axis in the developing cerebellum, whereas *OTX2*-positive granule cells were mostly located to the domain closest to the rhombic lip (Extended Data Fig. 5j). We therefore renamed this category of cells as *GC_diff_2_OTX2* in the revised manuscript. We confirm that our inability to distinguish this group of cells in the human snRNA-seq dataset is related to their relatively low numbers and/or sampling biases, given that a spatially restricted population of *OTX2*-positive differentiating granule cells are detected in the human 12 wpc cerebellum by smFISH.
5. Our spatial dataset of the human cerebellum at 12 wpc, a stage when Purkinje cell generation is completed (Haldipur et al. 2022)⁷, provides support for the presence of the 4 developmental subtypes in human (Fig. 2f). In the cerebellum section profiled, we found that the majority of the developing Purkinje cells express *FOXP1* and/or *RORB*, whereas *EBF2*, *CDH9* and *ETV1* mark restricted subsets of Purkinje cells, in line with the higher relative abundances of the early-born compared to late-born Purkinje cells. Furthermore, quantification of Purkinje cell numbers in the spatial dataset by imputing subtype labels based on alignment with the 11 wpc snRNA-seq data, confirmed the shift in Purkinje subtype prevalence in human (Fig. 2g).

In the revised manuscript, we have added the results of the exploration of the Purkinje cell subtypes in the Aldinger et al. dataset and the 12 wpc spatial dataset. For other “missing” cell (sub)types, we have added a paragraph to the Methods to describe all cases when a category was not detected in all three species (including cases when a category is missing in the mouse or opossum dataset). We base these observations on our data only, and hope that the Reviewer agrees that exploration of the Aldinger dataset did not change our conclusions. Additionally, for the *isth_N_SLC5A7* and *GC_diff_2_OTX2* category cells we include in the manuscript the spatial expression patterns of the respective markers in the 12 wpc human cerebellum, which provides an orthogonal approach for the detection of these subtypes in human.

Reviewer Fig. 4.3. Spatial mapping of markers of ependymal and midbrain-originating cells. a, Human 12 wpc cerebellum smFISH data for *SPAG17*, a marker of ependymal cells. **b**, Spatial distribution of midbrain-originating cells (MBO) in mouse E15.5 cerebellar primordium based on RNA *in situ* hybridisation data (Allen Developing Mouse Brain Atlas¹¹) for marker genes. Anterior and posterior coronal sections are shown. Arrows indicate the MBO stream.

Changes to the manuscript

- Main text

Page 9, lines 212-227: Based on key marker genes and the correlation of orthologous variable gene expression, we identified the same four developmental Purkinje subtypes in opossum (Fig. 2a,c,d). In human, we reliably distinguished two subgroups (*EBF1/2*-low and -high), but the patterned expression of subtype markers indicated additional diversity (Fig. 2a,c,d; Extended Data Fig. 3l). To investigate this further, we reanalysed an independent snRNA-seq dataset of human fetal (9-20 wpc) cerebellum¹ (Methods), and explored the expression of Purkinje subtype markers in our 12 wpc spatial dataset. These analyses confirmed the presence of all four Purkinje subtypes in the human fetal cerebellum (Fig. 2e,f, Extended Data Fig. 3l,n). In the 12 wpc Purkinje cell compartment (*SKOR2*), early-born Purkinje cell markers *FOXP1* and *RORB* exhibit widespread expression, whereas the late-born Purkinje cell marker *EBF2* was detected in restricted spatial domains, wherein *CDH9*-positive cells located medially and *ETV1*-positive cells laterally, in line with mouse Purkinje cell patterning (Figure 2b,f, Extended Data Fig. 3k). Comparison of the prevalence of early-born and late-born Purkinje cells in the three species revealed a shift in subtype ratios in human, with increased numbers of early-born Purkinje cells detected in human fetal samples from the two snRNA-seq datasets, and in the 12 wpc spatial dataset (Fig. 2e,g). In sum, while Purkinje cell patterning is conserved among mammalian species, the subtype ratios changed in the lineage leading to humans, facilitated by mechanisms that likely involve an augmented generation of early-born Purkinje cells.

Page 10, lines 237-244: Although the *SLC5A7*-marked subtype was not detected in the human snRNA-seq dataset, we observed cells co-expressing *SLC5A7* and *SLC17A6* in the 12 wpc cerebellum by smFISH (Extended Data Fig. 4m). /.../ Differentiating GCs clustered into early and late populations, and in mouse and opossum we additionally detected a distinct *OTX2*-expressing subset (Extended Data Fig. 5f,g). The latter was not distinguished in the human snRNA-seq dataset due to sampling biases, given that we detected *OTX2*-expressing granule cells by spatial mapping in the domain proximal to the rhombic lip in the 12 wpc cerebellum (Extended Data Fig. 5j).

- Discussion, page 23, lines 541-548:

The increased abundance of human Purkinje cells is biased towards the early-born subtypes, which in the mouse bear similarities to the adult Aldoc-positive subtypes enriched in the posterior and flocculonodular lobes of the cerebellar hemispheres. Purkinje cells in these regions project to the lateral (dentate) deep nuclei that in the human lineage expanded by selective increase in the numbers of the large-bodied subtype of glutamatergic neurons^{8,9}. Thus, it is tempting to speculate that the biased expansion of the Purkinje cells and large-bodied glutamatergic neurons in the lateral nuclei coincided during the course of human evolution. Additionally, adaptations in these areas have been suggested to support cognitive functions in humans¹⁰.

- Methods

Page 33, lines 785-810: Some annotation categories were not detected in all three species (Extended Data Fig. 2c). Out of these, many involve contaminating cell types located at the dissection borders: *progenitor_RP* (roof plate), *motoneuron*, and *neural_crest_progenitor* groups detected only in mouse, *progenitor_MB* and *MB_neuroblast* (midbrain) detected in opossum, *isthmio_neuroblast* detected in human and opossum, and *GABA_MB* (midbrain) detected in human. Some categories were not detected in human likely due to their overall low numbers: *isth_N_SLC5A7* is the least abundant subtype among the isthmio nuclei neurons in the mouse and opossum datasets (less than 250 cells) and these cells are situated at the dissection border; ependymal cells have low abundance in the mouse (57 cells) and opossum (160 cells) datasets, and are likely present in the human subcluster “orig.cl_38_9” (Supplementary Table 3) that, however, also contains cells from other types and was therefore not assigned to a cell type; *MBO* (midbrain-originating cells) were not distinguished in the human dataset, but markers of this cell type were expressed among some of the cells annotated as *NTZ_mixed* (“orig.cl_16_1”, Supplementary Table 3). Sampling of tissue fragments in human could be the reason why we did not distinguish *GC_diff_2_OTX2* in this species. The lower resolution of Purkinje subtype mapping in human could be related to the differences in Purkinje cell sampling, developmental dynamics, and/or subtype prevalence. In the mouse dataset, we did not distinguish subtype *progenitor_VZ_anterior*, likely due to limited resolution, given that it was possible to identify this subpopulation based on snATAC-seq data¹⁵. We also did not detect *preOPCs* in the mouse dataset: this population was mostly sampled from a single stage in human (17 wpc) and opossum (P14), and is in general expected to be scarce in the cerebellum given that more than 90% of oligodendrocytes in the mouse cerebellum originate in the ventral hindbrain¹⁶. The presence of group *glutamatergic_uncertain* in the human dataset only is due to inclusion of data from adult samples microdissected from deep nuclei region. Similarly, inclusion of these samples could underlie the unique detection of oligodendrocyte progenitor subtypes *OPC_early* and *OPC_late* only in the human dataset. Nevertheless, separation of human OPCs into subtypes is supported by previous studies¹⁷. It remains unclear if the human-unique categories *glut_DN_maturing*, *GC_diff_1_early* and *GC_diff_1_late*, and the opossum-unique categories *interneuron_MEIS2* and *ependymal_progenitor* are a result of biological or technical variation.

Page 34, lines 813-826: External snRNA-seq datasets

Processed data and annotations of adult mouse cerebellum snRNA-seq dataset by Kozareva *et al.*¹² were downloaded from https://singlecell.broadinstitute.org/single_cell/study/SCP795. Processed data and annotations of human fetal cerebellum sn/scRNA-seq dataset by Aldinger *et al.*¹ were downloaded from <https://www.covid19cellatlas.org/aldinger20/>. From the Aldinger *et al.* dataset, we extracted barcodes annotated as Purkinje cells (n=25 711), and used LIGER¹³ (k=10) to perform batch correction and integrate data across stages, assigning batches by *sample_id* and *experiment*. We performed Leiden clustering (leidenAlg¹⁴, version 1.0.5; resolution 0.6), and identified clusters that contained barcodes that likely represent doublets, based on the higher number of counts in these clusters and co-expression of markers of several cell types (*PAX2* for interneurons, *PAX6* for granule cells, *SOX2* for astroglia lineage cells). After excluding the doublet clusters, we performed LIGER integration (k=10) and Leiden clustering (resolution 1.2) on the remaining barcodes (n=14,246, 55% of the barcodes annotated as Purkinje cells in the original dataset). We annotated 15 of the obtained 17 clusters based on the expression of Purkinje subtype markers *RORB*, *FOXP1*, *CDH9*, *ETV1*, *EBF1* and *EBF2*. 732 barcodes (2 clusters, 5.1% of the barcodes) were not assigned to a subtype.

Page 35, lines 841-850: Quantification of cell type abundances and ratios

For Fig. 2e, we calculated the ratio of cell numbers of the early-born Purkinje subtypes (*FOXP1*, *RORB*) and the late-born subtypes (*CDH9*, *ETV1*) in each biological replicate. We only included samples that met the following criteria: (i) they come from fetal stages when Purkinje cell generation is complete (9-20 wpc in human), (ii) the relative abundances of *Purkinje_maturing* state cells (which were not separated into subtypes) among the *Purkinje_defined* and *Purkinje_maturing* cells is below 5% (our dataset; Extended Data Fig. 3i), (iii) at least 50 subtype-assigned cells are present. In the case of the Aldinger *et al.*¹ dataset we excluded samples that only contained the hemisphere or vermis (n=2), which altogether resulted in the inclusion of 9 samples representing 8 fetal stages (2 replicates for 14 wpc, and 1 replicate for each other stage). For subtype relative abundances presented in Extended Data Fig. 3i,r, 4j and 5h, we required at least 50 cells of the respective cell states to be present in a sample.

• Figures

Fig. 2. e, Top: UMAP of 14,246 human 9-20 wpc Purkinje cells from a published dataset¹ with cells coloured by subtype. Bottom: ratio of early-born to late-born Purkinje cell numbers in fetal samples. Biological replicates are shown with dots, and lines indicate the median of a stage (9 and 11 wpc). Corresponding developmental stages (Fig. 1a) are shown for mouse and opossum. f, Detection of Purkinje cell markers in the 12 wpc human cerebellum by smFISH. The same coronal section is shown in all panels; close-up views are indicated with rectangles; arrows point to double-positive regions. g, Mapping of the early- and late-born Purkinje cells in the 12 wpc human cerebellum by alignment of the smFISH data with 11 wpc snRNA-seq data. The pie chart indicates the percentage of cells by group. EB, early-born; LB, late-born.

The pie chart indicates the percentage of cells by group. EB, early-born; LB, late-born.

Extended Data Fig.3. k, Spatial distribution of Purkinje subtypes in E15.5 mouse cerebellar primordium based on RNA *in situ* hybridisation data¹¹ for subtype marker genes. Medial and lateral sagittal sections are shown. l, UMAPs showing expression of key marker genes in the subtype-assigned Purkinje cells in our mouse, opossum, and human datasets, and in the reanalysed Aldinger *et al.* 2021¹ dataset. Scaled expression of *EBF1* and *EBF2* is shown at the left to highlight the combinatorial patterns; scaled expression of subtype markers *RORB*, *FOXP1*, *CDH9* and *ETV1* is shown at the right with each cell coloured according to the gene that has the highest scaled expression level. For visualisation purposes, the scales were capped at 95th quantile for *RORB*, *FOXP1*, *CDH9*, *EBF1* and *EBF2*, and 99th quantile for *ETV1*. n, Dot plot showing expression of key marker genes in the Purkinje subtypes in the reanalysed Aldinger *et al.* 2021¹ dataset.

Major comment 5: Similarly, there are some populations that authors find in human developing cerebellum but not in mice/opossum ones. For example, populations 7 and 8 (glutamatergic deep nuclei maturing and mature neurons) in extended data fig.4 are present only in human data, while glutamatergic defined (population 6) neurons are present in all three species. What's the author's explanation for the observation? The authors should provide a pseudotime analysis of human populations to see if clusters 7 and 8 neurons are more mature than cluster 6 neurons. The generation of an integrated UMAP for deep nuclei cells across three species could help determine if populations 7 and 8 are mappings to other populations in mouse/opossum developing cerebellum.

We are grateful to the reviewer for this valuable comment, which helped us to refine our annotation. We assigned the label *glut_DN_mature* (population 8 in the original manuscript) based on the samples these cells originated from. Specifically, the samples in question were microdissected from the adult human dentate nucleus (lateral deep nucleus). We also identified a few markers (*NDUFA4*, *ATP5E*, *FAU*) that mapped to the deep nuclei in the Allen Mouse Brain Atlas (Reviewer Fig. 4.4a). Nevertheless, these cells also express many genes that are known to mark granule cells (*PAX6*, *SLC17A7*). Therefore, we admit that this population of cells may at least partially represent contaminating granule cells present in the microdissected sample. Given these unclaritys, we have now relabeled these cells as “*glutamatergic_uncertain*” and removed them from the RL/NTZ lineage. This population is not present in the mouse and opossum datasets, given that we did not sample glutamatergic deep nuclei region separately in these species. Importantly, this change in cell type annotation does not influence any downstream analyses since we only included categories and stages that were covered in all species.

The *glut_DN_maturing* cells (population 7) were originally annotated based on the unique markers expressed (*LMO1* *GABRG3*) and their origination from later developmental stages compared to the *glut_DN_defined* population. To further evaluate the differentiation status of these populations, we performed the pseudotime analysis as suggested by the reviewer. By comparing the pseudotime values across cell state categories among the RL/NTZ lineage cells, we found that the human *glut_DN_maturing* population has indeed slightly higher pseudotime values compared to *glut_DN_defined* population (Reviewer Fig. 4.4b,c, Extended Data Fig. 4f). This finding is consistent with our suggested order of differentiation for the two groups. To search for possible *glut_DN_maturing* cells in the mouse and opossum datasets, we integrated the data of the RL/NTZ broad lineage cells (NTZ neuroblasts, glutamatergic deep nuclei neurons, and isthmic nuclei neurons) from the three species. This analysis demonstrated that human *glut_DN_maturing* cells have minimal overlap with the mouse or opossum cells (Reviewer Fig. 4.4d), whereas other categories show expected overlap between all three species (note that *isth_N_Slc5a7* cells have no overlap in human, in accordance with their absence in the human dataset). However, these comparisons across species might be affected by differences in resolution: in all species we have an unresolved category called *NTZ_mixed* that contain cells from different cell types located at the nuclear transitory zone, and it remains possible that mouse and opossum cells corresponding to *glut_DN_maturing* are hidden in this group of cells.

Among the genes enriched in the *glut_DN_maturing* cells are genes coding for GABA-A receptor subunits (*GABRA2*, *GABRG3*) and neuronal activity-induced transcription factor *JUN* (Reviewer Fig. 4.4e). Future studies that more extensively sample the developing deep nuclei neurons, are needed to determine whether the exclusive detection of this population in the human dataset is due to sampling biases or represents a true biological shift in the GABA signalling in the developing glutamatergic deep nuclei neurons. The latter possibility is intriguing given that Purkinje cells, the major source of GABA in the cerebellum, have different dynamics in human.

We revised the Extended Fig. 4 and made corresponding updates to other figures in the manuscript to reflect the changes in annotation. However, due to space constraints and the uncertainties related to the *NTZ_mixed* cells, we have decided not to include the integration analysis in the manuscript as we do not believe it significantly contributes to the coherence of the paper. Nevertheless, we have included a plot depicting marker gene expression across NTZ cell states (Extended Data Fig. 4b), a panel on spatial mapping of the *glut_DN* state markers in the mouse E13.5 cerebellum (Extended Data Fig. 4e), the pseudotime analysis (Extended Data Fig. 4f), and detection of *glut_DN* markers in the human 12 wpc cerebellum (Extended Data Fig. 4l). Of note, for consistency we removed the cell state descriptor “mature”, and use the descriptor “maturing” also for the most mature state of Purkinje cells.

Reviewer Fig. 4.4. Annotation of glutamatergic deep nuclei neurons. **a**, Spatial expression patterns for markers used to annotate population 8 (*glut_DN_mature*, now *glutamatergic_uncertain*) in the initial manuscript. Data from the Allen Mouse Brain Atlas. Insets show close-ups of the deep nuclei region. Note that granule cell layer shows darker background due to high cell density, but no *in situ* hybridisation signal in the granule cells is detectable. Scale bars 0.25 mm. **b,c**, UMAPs of human RL/NTZ cells coloured by their state (**b**) or diffusion pseudotime values (**c**). Boxplots in **c** show distributions of the pseudotime values across cell states. **d**, UMAPs showing integration of RL/NTZ cells using Harmony. Top: cells from all species; bottom: cells from human, mouse and opossum plotted separately. Cell state and subtype colours as in **b**. In **b** and **d**, *glut_DN_maturing* category cells are highlighted in red. **e**, Expression of selected markers of *glut_DN_maturing* cells in RL/NTZ cells from human, mouse and opossum.

Changes to the manuscript

• Figures

Extended Data Fig. 4.

a, [...] In human, we distinguished a *LMO1*-marked population of glutamatergic deep nuclei neurons that likely represents a more mature cell state (see b-f and l). **b, i, Expression of key marker genes in the RL/NTZ cell states (**b**) or subtypes (**i**) in mouse, human and opossum. Dot size and colour indicate the fraction of cells expressing each gene and the mean expression level scaled per species and gene, respectively. **e, h,** Spatial distribution of RL/NTZ cell states (**e**) or glutamatergic deep nuclei and isthmus nuclei subtypes (**h**) in mouse E13.5 cerebellar primordium based on RNA *in situ* hybridisation data¹¹ for marker genes. Sagittal sections counterstained with HP Yellow are shown. Coloured arrows indicate the domains expressing markers of the different cell type/state categories; dotted arrows show the direction of the migration from the rhombic lip to the NTZ. **f,** UMAP of human RL/NTZ cells coloured by their pseudotime values, and distribution of pseudotime values across cell state categories. Colours and numbers as in **a**. **l,** Human 12 wpc cerebellum smFISH data for markers of the glutamatergic deep nuclei. The locations of the regions expanded at the right are shown with rectangles on the whole section at the left. Black arrows indicate *SLC17A6*-positive glutamatergic deep nuclei neurons. Pink arrow indicates *EOMES*-positive unipolar brush cell. Insets (dashed line) show close-ups of individual cells. *LMX1A*, a marker of *glut_DN_P* is detected in a minority of glutamatergic deep nuclei neurons (1); glutamatergic deep nuclei neurons expressing *GABRA2*, enriched in *glut_DN_maturing* cells, dominate the NTZ at 12 wpc (2).**

- Methods, page 32, lines 775-805

As a result of the label transfer and curation procedures 97% and 94% of the cells in the human (not considering the removed brainstem library) and opossum datasets were specified at the level of cell state, out of which 47% and 40%, respectively, were additionally assigned to a subtype (Supplementary Table 3-4). 1.1% of human cells and 2.6% of opossum cells belonged to subclusters that contain cells from different cell types located at the nuclear transitory zone (*NTZ_mixed*). 0.6% of human cells belonged to subclusters that contain glutamatergic neurons with uncertain identity originating from adult deep nuclei-enriched samples, and expressing markers of deep nuclei neurons (*NDUFA4*, *FAU*, *ATP5E*) and granule cells (*PAX6*, *SLC17A7*; *glutamatergic_uncertain*). [...] The presence of group *glutamatergic_uncertain* in the human dataset only is due to inclusion of data from adult samples microdissected from deep nuclei region.

Major comment 6: We find the claim that the authors make about evolutionarily conserved trajectories of different cell types development (except for Purkinje cells) to be partially misleading. The authors base this conclusion on the results of the hierarchical Bayesian modeling of different cell types' developmental dynamics. They find a significant difference (comparing across species) to be present only for Purkinje cells; however, this is not in line with the current state of the knowledge in the field. Their model indicates the secondary expansion that Purkinje cells progenitors undergo in cerebellar SVZ; however, granule cells are known to undergo the same process later in the development. If this is not possible to model because of the sampled time points/parts of the tissue, the claim of "conserved evolutionary trajectories for cell types except for Purkinje cells" should be made with caution.

We thank the Reviewer for raising this issue as it helped us to improve the analyses and conclusions related to the relative abundances and developmental dynamics of the cerebellar cell types. We agree with the Reviewer that the statement about the conservation of developmental dynamics of cerebellar cell types is partially misleading, given that our analyses have several limitations, as discussed below.

For each developmental stage, our datasets only sample a subset of cells present in the cerebellum. Therefore, our estimations of cell type abundances rely on proportions. If the relative abundances of one cell type increase, it means the proportions of other cell type(s) must decrease. In case statistically credible shifts in cell type proportions between species are detected for more than one cell type, it might be difficult to pinpoint which of these changes are biologically meaningful. With this notion in mind, we expanded the Bayes modelling to five cell types in the cerebellum, focussing on the developmental stages with representative sampling in human (Extended Data Fig. 7d). Purkinje cells still stand out, as their relative abundances in human are increased compared to mouse and opossum at two consecutive stages (8-9 wpc in human).

[This has been redacted]

As pointed out by the Reviewer, intermediate progenitors have been identified in both the ventricular zone (VZ) and the rhombic lip (RL) of the developing human cerebellum¹⁹. We propose that the increased relative abundances of Purkinje cells in human, at the time of Purkinje cell generation, could be related to the expansion of VZ progenitor zones. We agree with the Reviewer that similar mechanisms (expanded RL progenitor zones) could influence the production of granule cells. However, it remains unclear whether this mechanism would only keep the granule cell numbers balanced (compensation for the increased numbers of Purkinje cells), or also lead to higher proportions of granule cells (overcompensation). Besides the RL progenitors, the extent of transit amplification of the granule cell progenitors in the external granule cell layer (EGL) heavily influences granule cell numbers, and this process is

regulated by the signals from the underlying Purkinje cell layer. Dissection of these mechanisms would require quantification of cell numbers at the level of cell states/subtypes and dedicated experimental work in future studies.

Given that the estimation of abundances in the human dataset is affected by differences in sampling at late stages, we prefer to keep the conclusions about the shifts in relative abundances at the level of cell types and focused on early developmental stages. In light of the Reviewer's comment, and also Referee #1 major comment 4 (page 22) and Referee #4 Major comment 7 (page 47), we have made the following changes to the manuscript:

- (i) We expanded the Main Text, Fig. 1e, and Extended Data Fig. 7d to compare the relative abundances of 5 cell types at early developmental stages.
- (ii) We modified the conclusion of the section "Atlases of cerebellum development across mammals" - instead of "conserved developmental dynamics", we now refer to "conserved developmental sequence and timing".
- (iii) We toned down the related statements in the Summary and in the Discussion.

Changes to the manuscript

- Summary, page 2, lines 38-40

Our cross-species analyses revealed largely conserved developmental dynamics of cell type generation, except for Purkinje cells, where we observed an expansion of early-born subtypes in the human lineage.

- Main text, page 7, lines 159-172:

We applied hierarchical Bayesian modelling to compare the relative abundances of individual cerebellar cell types across matched developmental stages between species (Methods). We focused on early developmental stages that are less influenced by sampling differences (Fig. 1f, Extended Data Fig. 7d). The most striking difference we observed was a ~2-fold higher relative abundance of Purkinje cells in human compared to mouse and opossum at two stages when their relative abundances peak during development (8-9 wpc in human; Fig. 1f). The difference remains statistically significant even when additionally considering the VZ neuroblasts.

[This has been redacted]

(Fig. 1f, Extended Data Fig. 7d-e). This change in Purkinje cell dynamics in the human lineage could be related to differences in developmental durations between species and/or the unique presence of basal progenitors in the human cerebellum¹⁹ that may serve as an additional pool of Purkinje cell progenitors. Altogether, our snRNA-seq atlases provide a comprehensive view of the cerebellar cell types in mammals and show that the developmental sequence and timing of their generation have been largely conserved for at least 160 million years.

- Discussion, page 23, lines 536-538:

The most notable difference is the significantly higher relative abundance of Purkinje cells during early fetal development in human, which is possibly linked with the expansion of neuronal progenitor pools in the human cerebellum in the same time window¹⁹.

- Figures:

Fig. 1. e,f. Relative cell type abundances across developmental stages in the whole datasets (e) or amongst the cerebellar cells (f). For adult human, only data from cerebellar lobes is included. /.../ In f stages are aligned as in a, the line indicates the median of biological replicates, orange shading marks stages with representative sampling in human, and asterisks indicate differences in the relative abundances in human compared to mouse and opossum.

[This has been redacted]

Extended Data Fig. 7e,d. (see next page)

Extended Data Fig. 7. d, Hierarchical Bayes model analysis of differences in the relative cell type abundances across species at corresponding developmental stages. Difference in posterior (y-axis) shows modelled proportion differences between pairs of species (comparisons); 0 indicates no shift in proportions (dotted line). The modelled differences are summarised as 95% highest density intervals (HDI₉₅; lines) for each cell type at developmental stages (x-axis; depicted on top) where at least 50 cells were present. Only corresponding stages with representative sampling in human were considered. Differences in the relative abundances were called (yellow shading) when HDI₉₅ of at least two comparisons did not overlap 0 (e.g., a human-specific change is assumed when HDI₉₅ of human vs mouse and human vs opossum comparisons does not overlap 0, and HDI₉₅ of mouse vs opossum comparison overlaps 0). **e**, Relative abundances of cells annotated as Purkinje cells or VZ neuroblasts across developmental stages. VZ neuroblasts and Purkinje cells were analysed together to exclude the effect of possible biases in the annotation between the three species. Stages are aligned as in Fig.1a, the line indicates the median of biological replicates, orange shading marks stages with representative sampling in human, and asterisks indicate differences in the relative abundances in human compared to mouse and opossum.

Major comment 7: Moreover, oligodendrocytes in humans have intracerebellar origin in contrast to oligodendrocytes in mice and opossum. In that light, the fact that the models for oligodendrocytes developmental dynamics are similar between humans and the other two species is surprising. Based on the UMAPs dedicated to glial development (extended data fig.6d), the difference in oligodendrocytes developmental trajectory between species should be striking.

Could the authors reanalyze this part of the data with this in mind? At least the result hierarchical Bayesian model for granule cells and oligodendrocytes should be visualized. Are there may be differences in migration markers? Finally, It would be interesting to come up with a proxy to estimate whether glial development is more or less evolutionarily conserved than neuronal development.

We apologise for the confusion raised on the origin of oligodendrocytes in the cerebellum. To the best of our knowledge, this has not been studied in human or other mammals besides the rodents. In mouse, most of the oligodendrocyte progenitor cells (OPCs) in the cerebellum originate from the ventral rhombomere 1, and 6% are born in the cerebellum (Hashimoto et al. 2016)¹⁶.

We pinpointed two problems in the original manuscript. First, we admit that the sentence on preOPCs in the Main text was misleading (“*preOPCs are relatively abundant in opossum, rare in human, and are not detected in mouse, in accordance with the extra-cerebellar origin of most oligodendrocytes in mouse*¹⁶.”). We have removed this sentence from the revised manuscript, and instead provide a clarification on the absence of *preOPC* category cells in the mouse dataset in the Methods section (see *Changes to manuscript* below). Second, for Extended Data Fig. 6d we did not denote that the human dataset includes adult samples dissected from the cerebellar deep nuclei and white matter region, in addition to the tissue fragments from different lobes of the cerebellar cortex (Extended Data Fig. 1a). This considerably increases the sampling of oligodendrocyte lineage cells in the human adult data (Extended Data Fig. 7c). Analogous samples were not included in the mouse and opossum datasets, where all adult data comes from dissections that contain half of the entire cerebellum (Supplementary Table 1). Plots showing the relative abundances of the cell (sub)types (Fig. 1e,f Extended Data Fig. 6c) exclude the human samples dissected from the cerebellar deep nuclei and white matter region, given that the samples from the different lobes better represent the overall cell type compositions of the cerebellum (Extended Data Fig. 7c).

We could not evaluate the differential expression of migration markers. In general, developmental heterogeneity of OPCs is less pronounced than that of neurons. Recently, regional patterning of the OPCs along the anteroposterior axis of the neural tube has been described (Braun et al. bioRxiv)⁴⁹, but not much is known about the dorsoventrally differentially expressed genes in the hindbrain OPCs.

As described in the previous section (Referee #4 major comment 6), the revised manuscript incorporates hierarchical Bayesian modelling for five cell types (Extended Data Fig. 7d). Fair comparisons of oligodendrocyte relative abundances were not possible, given that they mostly originate from the developmental stages, which do not have representative sampling in human (newborn and adult). Nevertheless, to get initial insights, we performed Bayesian modelling for the oligodendrocyte lineage cells (Reviewer Fig. 4.5). Again, the human samples enriched for the cerebellar deep nuclei and white matter were excluded from the analysis. No consistent differences in oligodendrocyte relative abundances between the species were detected. Nevertheless, given the limitations mentioned, we cannot make fair comparisons between the developmental dynamics of glial and neuronal cells.

Reviewer Fig. 4.5. Comparison of oligodendrocyte abundances across species. Relative abundances across developmental stages amongst the cerebellar cells and hierarchical Bayes model analysis of differences in the relative abundances at corresponding developmental stages. For adult human, only data from cerebellar lobes is included. Orange shading marks stages with representative sampling in human. Difference in posterior shows modelled proportion differences between pairs of species (comparisons); 0 indicates no shift in proportions (dotted line). The modelled differences are summarised as 95% highest density intervals (HDI₉₅; lines) at developmental stages where at least 50 cells were present in all species.

Changes to the manuscript

- Main text, page 6, lines 139-140:

Additionally, in human and opossum we detected a small population of cells intermediate between astroglial progenitors and OPCs that likely represents the preOPC state⁵⁰ (*EGFR*; Extended Data Fig. 6c,d,f).

- Methods, page 33, lines 801-804:

We also did not detect *preOPCs* in the mouse dataset: this population was mostly sampled from a single stage in human (17 wpc) and opossum (P14), and is in general expected to be scarce in the cerebellum given that more than 90% of oligodendrocytes in the mouse cerebellum originate in the ventral hindbrain¹⁶.

- Figure Legend, page 8, lines 182-183:

Fig. 1. e,f, Relative cell type abundances across developmental stages in the whole datasets (e) or amongst the cerebellar cells (f). For adult human, only data from cerebellar lobes is included.

- Extended Data Figure Legend, page 73, lines 1655-1662:

Extended Data Fig. 6. c, Relative abundances of astroglia subtypes, ependymal progenitors and preOPCs across developmental stages. Colours are as in a; astroglial cells not assigned to a subtype are in grey. Stages are aligned as in Fig. 1a. Human adult samples dissected from the deep nuclei region were excluded. **d,** Uniform Manifold Approximation and Projection (UMAP) of 28,486 mouse, 32,897 human and 20,742 opossum glial cells coloured by their subtype or state. Colours and numbers as in a. Progenitors not assigned to a subtype are in grey. Mouse roof plate progenitors and human preOPCs are low in numbers and not discernible in this UMAP. Inclusion of human adult samples dissected from the deep nuclei region explains the high numbers of oligodendrocytes in the human UMAP.

Major comment 8: While the authors use a well-recognized tool for integration (LIGER) between pairs of datasets, the batch effect is clearly still present on integrated UMAPs shown on extended data fig. 2b. Did the authors try other integration algorithms (e.g., Harmony, Seurat integration)? A mini benchmark of this in the supplementary could be of great technical help to the field of evolutionary single-cell analyses.

Integrating data of developmental trajectories across species is a challenging task (Luecken et al. 2022)⁵¹. It is worth noting that data integration was one of the early steps in our study, performed when many methods were not yet available or inadequate to handle datasets as large as ours.

We employed LIGER to integrate data within a single species across developmental stages and biological replicates, given that other methods we explored (Seurat) resulted in over-merging the data from different developmental stages and did not preserve the developmental trajectories.

For cross-species integration of the full datasets, we relied on orthologous genes, and combined LIGER with fastMNN correction. We used pairwise integrations for transferring cell type labels, and integrated data from all three species to estimate pseudoages (Extended Data Fig. 2b,d,f). While we acknowledge that our integrated embeddings may still exhibit species effects (which may arise from batch effects and/or biological differences), we want to emphasise that our downstream analyses primarily rely on discrete cell type/state categories that we meticulously curated after conducting label transfer based on cross-species integration. Similarly, the pseudoage estimation was one of the three approaches we used for defining stage correspondences across species, and, importantly, it agreed with the other two methods (comparing transcriptomes and cell state composition; Extended Data Fig. 2e-g).

We additionally used Harmony to integrate cells from granule and Purkinje cell lineages from the three species (Fig. 3d, Extended Data Fig. 9a), and these embeddings formed the basis for pseudotime analyses. We employed several approaches to ensure that the alignments were in line with the known biology of these cell types (Extended Data Fig. 9b,c,h,i). Taken together, we are confident that our conclusions are not reliant on the specific method used for cross-species integration.

We concur with the Reviewer's suggestion that our dataset is suitable for benchmarking integration algorithms, but we feel that conducting a conclusive benchmarking analysis is beyond the scope of our current work. Furthermore, recent studies have already systematically evaluated scRNA-seq data integration algorithms for cross-species integration (Luecken et al. 2022³¹, Song et al. bioRxiv³²), and will inform future studies in the field of evolutionary single-cell analyses. Nevertheless, in Reviewer Fig. 4.6, we provide an example of an integrated embedding generated using Harmony, which we consider overall similar to the embedding obtained by combining LIGER with fastMNN.

Reviewer Fig. 4.6. Integration of full human, mouse and opossum datasets using different computational tools.

Major comment 9: The authors claim to identify many target genes with unique expression patterns in humans. This is an interesting broad result that should be qualified more to be a relevant finding. What are the genes that are "lost" in humans? It would be extremely helpful to perform a gene ontology or gene set enrichment analysis to understand whether some biological process or pathway is overrepresented in this set.

We performed an over-representation analysis (WebGestalt²) on the genes that show trajectory changes (Fig. 4b) or presence/absence expression differences between species (Fig. 4e). Functional enrichments are limited among the genes with presence/absence expression differences, which was expected, given that we call expression differences within cell types across all developmental stages and the number of genes in each group is not high (especially after revising the calls, see Referee 4, Major comment 10, page 50). Nevertheless, a few overrepresented categories were identified (Supplementary Table 11), including sensory perception among genes that gained expression in human oligodendrocytes and myofilament among genes that gained expression in human astroglia.

Changes to the manuscript

- Main text

Page 18, lines 417-421: In each lineage, only a few (1-4) genes have changed trajectories in both cell types, suggesting that changes in regulatory programs are largely cell type-specific. Nevertheless, genes with human-specific changes in

either granule cells or Purkinje cells share enrichments for functions related to synaptic membrane and glutamatergic synapse (FDR<0.05, Supplementary Table 11).

Page 19, lines 446-448: Functional enrichments among the genes with expression differences include sensory perception and myofilament for genes that gained expression in human oligodendrocytes or astroglia, respectively (FDR<0.05, Supplementary Table 11).

- Methods, page 54, lines 1319-1341:

For gene ontology and pathway enrichment analyses we used the WebGestaltR² package (version 0.4.4) and mouse functional databases of gene ontology and KEGG pathways as provided by WebGestalt (daily build accessed on 01.14.2019). /.../ Functional enrichments among the genes with preserved trajectories assigned to different trajectory classes (Fig. 3g, Extended Data Fig. 9e) and among the genes with trajectory changes assigned to different lineages (Supplementary Table 11) were identified by over-representation analysis against the background of dynamic orthologous genes (FDR < 0.1). Functional enrichments among the genes with presence/absence expression differences in cerebellar cell types (Supplementary Table 11) were identified by over-representation analysis against the background of 7,047 orthologous genes included in the analysis (see *Presence/absence expression differences*; FDR < 0.1). Databases *biological process*, *molecular function*, *cellular component*, and *KEGG* were used.

- Supplementary Table:

Supplementary Table 11. Gene ontology terms associated with genes that exhibit trajectory changes or presence/absence (P/A) expression differences.

Major comment 10: Finally, regarding all these claims about genes specific or lost in one species. It would be important to have internal positive controls to support these statements. A control would serve the purpose of avoidings false negatives: genes that are close to the detection sensitivity of scRNA-seq and are "dropping out" in the species where they are expressed the least. Moreover, because of evolutionary divergences of sequence at the 3'UTR (note: 10X chromium is a 3' method), some genes might be more or less efficiently reverse-transcribed and amplified for one species generating false negatives. Therefore, convincing proof a gene is lost in a lineage would have to include a positive controls that are detectable in another cell type in the same species. In addition, at least one of these two further corroborations should be presented:

- (i) the average abundance distribution of the "lost genes" is not significantly different from the distribution of other, similarly variable genes.
- (ii) checking that changes sequence and/or exon structure comparison of the 3' part of the gene model between the species cannot explain the "lost genes" (this can be done with a Generalized Linear Model with predictors summary statistics from the 3' part analysis: say GC content, length,...).

We thank the Reviewer for this comment as it helped us to improve the analyses of presence/absence expression differences between the species. Below we separately discuss the two issues raised by the reviewer - (1) internal/within species controls and (2) cross-species differences in orthologous gene sequence/structure.

(1) We would first like to note that already in our initial manuscript we considered the possibility that a gene might remain undetected in a species due to technical limitations of the snRNA-seq method. Specifically, we used published bulk RNA-sequencing data⁴⁵ covering the development of the cerebellum to evaluate the genes that were not detected in any cerebellar cell type in our datasets. We excluded orthologous gene groups that were not detected (< 50 CPM) in the cerebellum snRNA-seq but reliably expressed (>5 RPKM) based on bulk RNA-seq data.

The majority of genes that we called as absent in a species are detected (i.e., reach 50 CPM) in at least one other cell type of the same species (Reviewer Fig. 4.7a). Although the maximum expression (across all pseudobulks per biological replicate and cell type, including additional cell types besides the 8 neural cell types (Methods)) of genes that were called as "lost" in at least one cell type are not as high as the maximum expression of genes that were called as "gained" in at least one cell type (Reviewer Fig. 4.7b, Extended Data Fig. 10g), we argue that this is biologically plausible given that the modular organisation of gene regulatory control might allow lowly expressed genes to more easily lose expression in developmentally related cell types. Nevertheless, to reduce the ambiguity for the genes that are close to the detection threshold (50 CPM), we further considered the relative expression levels of the genes within each species. For this, we calculated the ratio between the gene's maximum expression within a cell type and the maximum expression of the same gene across all cell types in each species. Most of the genes we called as present in a species-specific manner reach at least 30% of their global maximum expression in the cell type where the call was made, whereas most of the genes called as absent in a species-specific manner are below 30% of their maximum

expression levels (Reviewer Fig. 4.7c). Genes that don't meet the 30% criterion represent less confident calls as they often involve expression changes in all cell types and are therefore more likely to be affected by technical biases related to differences in detection sensitivities between the species. Thus, we have now added the 30% relative expression criterion to our pipeline for calling presence/absence expression differences (Extended Data Fig. 10e). However, this criterion is not implemented for the genes that are not detected in any cell type in the cerebellum (these genes were benchmarked against the bulk RNA-sequencing data, as explained above).

(2) Although Chromium Single Cell 3' GEX is a 3' method, many of the enzymatic reactions applied in the preparation of the libraries involve full-length transcripts (reverse transcription, cDNA amplification). We therefore evaluated the effects of both, full gene (exons) and 3'UTR characteristics on the detection of gene expression. We estimated the correlations between fold differences in gene expression and the fold differences in length or GC content of orthologous genes across pairs of species. This analysis revealed that gene expression differences do not correlate with exonic length differences or GC content differences between the orthologues (Reviewer Fig. 4.7d). Nevertheless, among the genes we called as expressed in a species-specific manner, some degree of length-bias can be detected (Reviewer Fig. 4.7e, Extended Data Fig. 10f). Thus, while differences in exon lengths are not globally predictive of differences in expression levels, genes with extreme differences in exon lengths are more likely to be called as specific-specific gains or losses of expression. We note that the observed length biases could have both biological and technical sources. However, to be conservative with our calls we decided to exclude all genes that show large differences in exonic length between the species. Specifically, we first calculated pairwise median-adjusted differences in exonic lengths of all orthologous. Median-adjustment is necessary to consider global differences in exonic length between the species (global shifts are accounted for by the CPM normalisation). We only considered orthologous gene groups for which the pairwise adjusted exonic length fold differences were less than 3 (Extended Data Fig. 10e). We report the exonic length and GC content differences among the final presence/absence differences we called in Extended Data Fig. 10f. Additionally, we tested, as suggested by the reviewer, if expression differences could be predicted based on exonic length and GC content differences. We found that random forest models were not able to predict the presence/absence expression differences between the species (Reviewer Fig. 4.7f), indicating that cross-species differences in exonic length and GC content of the genes are not driving our calls.

The additional filtering steps reduced the number of genes included in the analysis (7,047 vs 9,189), and the total number of genes with presence/absence expression differences identified (1,077 vs 1,889). For instance, on average 57% (range 49-68%) of the human gains called in each cell type in the initial manuscript were kept; 22% (16-30%) did not meet the relative expression criterion, and 25% (18-31%) had large differences in the exonic lengths between the species. Most of these exonic length differences stem from the shorter length of the orthologues annotated in the opossum genome. To increase the number of genes considered in this analysis, we conducted a separate pairwise comparison of human and mouse. This allowed us to evaluate 8,620 genes, and identify 1,392 genes with presence/absence expression differences between the two species (Extended Data Fig. 11c, Supplementary Table 12). This additional analysis improved our ability to map cerebellum-associated disease genes that have different expression patterns in cerebellar cell types in human and mouse, the most common model organism in biomedical studies.

Finally, we would like to highlight that addition of the multiplexed smFISH-based spatial dataset of the human 12 wpc cerebellum allowed us to directly detect expression of a subset of the genes with cross-species expression differences in human cerebellar cells. In this dataset, we only included genes for which the absence of expression in mouse cerebellar cell types was supported by public *in situ* hybridisation data from the Allen Developing Mouse Brain Atlas¹¹ and/or GenePaint²¹ (Supplementary Table 5). To rule out any potential technical reasons for this absence of signal in mouse *in situ* hybridisation data, we prioritised genes that were detected in other cerebellar cell types and/or tissues in mouse. Out of the 26 genes we tested in human, 22 could be confirmed as expressed in the respective cell types, 3 had overall low detection by smFISH, and in one case a gene was not detected in one of the cell types where a gain was called (Extended Data Fig. 11e,f, Supplementary Table 5). Thus, the presence/absence expression differences we identified based on snRNA-seq data, are enriched for genes that exhibit concordant patterns in *in situ* hybridisation data, providing additional support for the robustness and reliability of our findings.

Reviewer Fig. 4.7. Presence/absence gene expression differences. **a**, Detection of genes, which were called absent in at least one cell type in a species, in other cell types in the cerebellum. **b**, Distribution of expression levels for the genes in different presence/absence categories as assigned in the initial manuscript. Assignments from all eight main neural cell types are included (i.e., each gene is represented 8 times). Maximum expression during development among all cell types (15 cerebellar cell types (Methods); left) or in affected cell type (right) is shown. The same plot for the final assignments is shown in Extended Data Fig. 10g. **c**, Proportion of maximum expression for human gains.

The ratio between the gene's maximum expression within a cell type and the maximum expression of the same gene across all cell types was calculated. The lines indicate the 30% threshold. The plot is shown for the initial calls (top), after filtering out the genes with large differences in exonic length (middle), and after additionally applying the 30% of maximum expression criterion (final calls; bottom). **d**, Correlations between pairwise differences in expression and pairwise differences in exonic or 3'UTR length or GC content. **e**, Fold difference in exonic or 3'UTR length (left) and GC content (right) across species for the genes in different categories as assigned in the initial manuscript. Similar plots for the final assignments are shown in Extended Data Fig. 10f. **f**, Random forest model with exonic length and GC content as predictors. Granule cell and Purkinje cell calls are shown.

Changes to the manuscript

- Summary, page 2, lines 42-45:

However, we also identified many orthologous genes that gained or lost expression in cerebellar neural cell types in one of the species, or evolved new expression trajectories during neuronal differentiation, indicating widespread gene repurposing at the cell type level.

- Main text

Page 19, lines 435-438: Out of the 7,062 orthologous genes included in this analysis 1,077 (15.3%) displayed presence/absence expression differences in at least one cell type. After polarizing the changes we found, on average, 62 gains and 19 losses in the human lineage, and 33 gains and 31 losses in the mouse lineage per cell type (Fig. 4e, Supplementary Table 12).

Page 20, lines 461-476: We then asked whether genes associated with cerebellum-linked diseases show presence/absence expression differences between human and mouse, the most common model organism in biomedical studies. In this analysis we additionally considered genes for which polarisation using opossum data was not possible (Methods, Extended Data Fig. 10e), and identified 1,392 genes (16.1% of 8,620) with expression differences between the two eutherian species (Extended Data Fig. 11c, Supplementary Table 12). Among these are 26 disease-associated genes. For instance, autism and Down syndrome-associated *DSCAM* gained expression in human astroglia (Fig. 4i), and *FGF2*, implicated in pilocytic astrocytoma, is expressed in human but not mouse astroglia and oligodendrocytes (Extended Data Fig. 11d). To substantiate the detected presence/absence expression differences, we spatially mapped 26 of these genes in the 12 wpc human cerebellum, focussing on genes for which absence of expression in mouse is supported by public *in situ* hybridisation data^{11,21} (Supplementary Table 5). Visualisation of smFISH signals and quantification of the expression levels in cells labelled based on integration with our 11 wpc snRNA-seq data, confirmed the co-expression of 22 genes with the respective cell type markers (Fig. 4j, Extended Data Fig. 11e,f). For instance, *PIEZO2*, *PLCZ1* and *DSCAM* were detected in *NOTCH1*-positive progenitors, and *CPLX4* in *PAX2*-marked interneurons. We further explored the available human immunohistochemistry data³² to map the human-present genes that show cell type-specific gene expression in the adult cerebellum.

Page 21, lines 482-484: Thus, by employing orthogonal datasets, we validated a subset of the detected presence/absence expression differences.

- Discussion, page 23, lines 555-557:

Furthermore, we found presence/absence expression differences between the species for all neural cell types, a subset of which we validated using multiplexed smFISH, and detected shifts in the gene expression trajectories during Purkinje and granule cell differentiation.

- Methods

Page 40, lines 959-968: The spatial mapping was performed using an available 12-week post-conception human cerebellum sample provided by the Human Developmental Biology Resource (HDBR; UK). The cerebellum was dissected, mounted in Tissue-Tek OCT compound (Sakura Finetek) and frozen in isopentane cooled to its freezing point. 10 μ m coronal cryosections were collected on Molecular Cartography coverlips (Resolve Biosciences). The probeset targeted 100 genes (Supplementary Table 5) manually selected based on our snRNA-seq data to cover markers of the cerebellar cell (sub)types and states, and selected genes with presence/absence expression differences between human and mouse (see *Presence/absence expression differences*). Among the latter we selected genes that are expressed in the human respective cell type in the 11-17 wpc developmental time window based on snRNA-seq data, and for which absence in the mouse respective cell type is supported by public *in situ* hybridisation data^{11,21} (Supplementary Table 5).

Page 51, lines 1239-1262: We further considered possible biases stemming from evolutionary divergence of the sequences and differences in gene annotation quality between the species (GC content, length), which could affect the

efficiency of reverse transcription, PCR-amplification, and/or mapping of the sequences. We determined the maximum expression of each gene across all pseudobulks in each species, and compared the cross-species fold differences in gene's maximum expression to differences in gene length and GC content. We did not observe any strong correlation between fold difference in a gene's maximum expression and fold difference in its median-adjusted exonic length (human-mouse Pearson's $r=0.07$, human-opossum $r=0.11$, mouse-opossum $r=0.06$), 3'UTR length (human-mouse $r=0.07$), exonic GC content (human-mouse $r=-0.01$, human-opossum $r=-0.01$, mouse-opossum $r=-0.02$) or 3'UTR GC content (human-mouse $r=0.01$). Nevertheless, in our initial analyses we observed that genes we called as expressed in a species-specific manner tend to be longer (exonic length) in the species where expression is present compared to their orthologous in species where expression is absent, indicating some length bias in detection sensitivities in the Chromium snRNA-seq data, as also noted previously⁵³. Therefore, to reduce the effects of these biases in our cross-species comparisons, we removed the orthologous gene groups that show major differences in exonic length between the species. For this, we first calculated pairwise median-adjusted differences in exonic lengths of orthologous genes. Median-adjustment is necessary to consider global differences in exonic length between the species (global shifts are accounted for by the CPM normalisation). We only considered orthologous gene groups for which the pairwise adjusted exonic length fold differences were less than 3 (Extended Data Fig. 10e). In the comparative analyses of all three species (human, mouse, opossum), 5,693 genes were affected by the technical limitations related to gene detection and/or annotation in at least one of the species and removed, whereas 7,062 1:1 orthologous genes were kept in the downstream analysis. In a separate comparison of human and mouse only, 4,135 genes were removed because of these technical limitations, and 8,620 genes were kept in the downstream analysis.

Page 52, lines 1267-1276: Next, in each species we determined the maximum expression of each gene per cell type across all matched stages, and calculated the ratio to overall maximum expression within the species, and pairwise fold differences between the orthologues from different species (Extended Data Fig. 10e). The within-species comparison serves to reduce the effects of possible biases in the detection sensitivities between the species. We called a gene "present", when it was reliably expressed in the cell type and its maximum expression levels in the cell type reached 30% of maximum expression levels across all cell types (Extended Data Fig. 10e). We called a gene "absent", when it was not reliably expressed in the cell type and its maximum expression levels in a cell type were below 30% of maximum expression levels across all cell types or the gene was not reliably expressed in the cerebellum (not expressed in any cell type).

Page 53, lines 1289-1306: In a comparison of human and mouse only, the genes with presence/absence expression differences were classified as *mouse-expressed* and *human-expressed*.

For each pair of species, we compared exonic length and GC content ratios among the genes within different presence/absence call classes, and used random forest models with exonic length and GC content as predictors to test if these could explain the presence/absence differences we called based on expression. As noted above, we detected some exonic length bias within the calls (Extended Data Fig. 10f). The length bias, however, could have both biological and technical sources, and random forest models were not able to predict the presence/absence expression differences between the species based on exonic length and GC content of the genes, indicating that these are not driving our calls. We also evaluated the possible effects of using different Chromium versions (v2 and v3) for data production on classifying expression differences. For this we focussed on a subset of human and mouse datasets that have at least 2 biological replicates produced with Chromium v2 (E13.5, E14.5, E15.5, P7, P14 in mouse, and corresponding stages in human). We then replaced one of the human v2 replicates from each stage with a replicate produced with Chromium v3, and called presence/absence expression differences in the two subsets (mouse v2 : human v2, and mouse v2 : human v2/3) using the pipeline described above. Similar numbers of expression differences were detected using the two subsets (Extended Data Fig. 10h), indicating that our approach is robust to differences in Chromium versions.

- Figures

Fig.4. j, Human 12 wpc cerebellum smFISH data for *PIEZO2*, *PLCZ1* and *DSCAM*. Co-expression with *NOTCH1* (progenitors), *KIRREL2* (VZ) or *SLIT2* (RL) is observed. The locations of the regions expanded at the right are shown with rectangles on the whole section at the left.

Extended Data Fig. 10. **e**, Scheme on classification of presence/absence expression differences. **f**, Median-adjusted fold difference in exonic length (left) and exonic GC content (right) across species for the genes in different categories assigned based on the presence/absence expression patterns in the cerebellar cell types in the three therian species. Genes below the threshold (< 50 CPM) in all species are marked as low in all therians. **g**, Distribution of expression levels for the genes in different presence/absence categories. Assignments from all eight main neural cell types are included (i.e., each gene is represented 8 times). Maximum expression during development among all cell types (15 cerebellar cell types (Methods); left) or in affected cell type (right) is shown.

Extended Data Fig. 11. (above and on the next page) **c**, Presence/absence expression differences between mouse and human cerebellar cell types. **e**, Human 12 wpc cerebellum smFISH data for genes with presence/absence expression differences between human and mouse. Asterisks denote expression changes that could not be polarised; other changes were assigned as gains in the human lineage. The locations of the regions 1-10 expanded at the right are shown with rectangles (solid line) on the whole section at the left. mRNA spots are black for the genes with differences, and coloured for the cell type markers. Arrows and insets show example cells where co-expression with the respective cell type marker(s) is detected. **f**, Quantification of the smFISH data in **e**. Cell type labels were transferred based on alignment²³ of the segmented²² smFISH dataset with the 11 wpc snRNA-seq data (Fig. 1d). Expression levels (mRNA counts) were normalised to segment area and scaled. The cell types where a difference was called are highlighted for each gene. Out of the 26 genes tested, expression of 22 genes in the respective cell type(s) was confirmed by smFISH; 3 genes displayed overall low signal, and expression of one gene remained undetected in a cell type where the change was observed based on snRNA-seq data.

• Supplementary Tables:

(i) Added table: **Supplementary Table 5**. List of genes analysed by multiplexed single molecule fluorescence *in situ* hybridisation.

(i) Modified tables: Characterization of orthologous genes for expression trajectories during Purkinje and granule cell differentiation, and for presence/absence (P/A) expression differences in cerebellar neural cell types across species are now reported in two separate tables. We include in these tables the metrics that were used for making the calls.

Supplementary Table 10. Characterization of orthologous genes for expression trajectories during Purkinje and granule cell differentiation.

Supplementary Table 12. Presence/absence (P/A) expression differences in cerebellar neural cell types across species.

MINOR COMMENT

Minor comment 1: On extended data fig. 5b, the numbers on the UMAP do not correspond to the numbers assigned to the clusters on extended data fig. 5a subplot.

Thank you for noticing this mistake. We have corrected the numbers in the plot (now Extended Data Fig. 5c).

Changes to the manuscript

- Extended Data Figures:

Extended Data Fig. 5.

c, f, Uniform Manifold Approximation and Projection (UMAP) of 32,767 mouse, 73,492 human and 36,585 opossum RL/EGL cells coloured by their state (**c**) or subtype (**f**). Colours and numbers as in **a**.

**References**

- Aldinger, K. A. *et al.* Spatial and cell type transcriptional landscape of human cerebellar development. *Nat Neurosci* **24**, 1163–1175 (2021).
- Liao, Y., Wang, J., Jaehnig, E. J., Shi, Z. & Zhang, B. WebGestalt 2019: gene set analysis toolkit with revamped UIs and APIs. *Nucleic Acids Res* **47**, W199–W205 (2019).
- Arendt, D. *et al.* The origin and evolution of cell types. *Nat Rev Genet* **17**, 744–757 (2016).
- Hu, H. *et al.* AnimalTFDB 3.0: a comprehensive resource for annotation and prediction of animal transcription factors. *Nucleic Acids Res* **47**, gky822 (2018).
- Wittkopp, P. J., Haerum, B. K. & Clark, A. G. Evolutionary changes in cis and trans gene regulation. *Nature* **430**, 85–88 (2004).
- Wilson, M. D. *et al.* Species-Specific Transcription in Mice Carrying Human Chromosome 21. *Science* **322**, 434–438 (2008).
- Haldipur, P., Millen, K. J. & Aldinger, K. A. Human Cerebellar Development and Transcriptomics: Implications for Neurodevelopmental Disorders. *Annu Rev Neurosci* **45**, (2022).
- Kebschull, J. M. *et al.* Cerebellar nuclei evolved by repeatedly duplicating a conserved cell-type set. *Science* **370**, (2020).
- Kebschull, J. M. *et al.* Cerebellum Lecture: the Cerebellar Nuclei—Core of the Cerebellum. *Cerebellum* (2023) doi:10.1007/s12311-022-01506-0.
- Magielse, N., Heuer, K., Toro, R., Schutter, D. J. L. G. & Valk, S. L. A Comparative Perspective on the Cerebello-Cerebral System and Its Link to Cognition. *Cerebellum* 1–15 (2022) doi:10.1007/s12311-022-01495-0.
- Allen Institute for Brain Science. Allen Developing Mouse Brain Atlas. <http://developingmouse.brain-map.org/> (2008).
- Kozareva, V. *et al.* A transcriptomic atlas of mouse cerebellar cortex comprehensively defines cell types. *Nature* **598**, 214–219 (2021).
- Welch, J. D. *et al.* Single-Cell Multi-omic Integration Compares and Contrasts Features of Brain Cell Identity. *Cell* **177**, 1873–1887.e17 (2019).
- Kharchenko, P., Petukhov, V. & Biederstedt, E. leidenAlg: Implements the Leiden Algorithm via an R Interface. *R package version 1.0.5*. <https://github.com/kharchenkolab/leidenAlg> (2021).
- Sarropoulos, I. *et al.* Developmental and evolutionary dynamics of cis-regulatory elements in mouse cerebellar cells. *Science* **373**, eabg4696 (2021).
- Hashimoto, R. *et al.* Origins of oligodendrocytes in the cerebellum, whose development is controlled by the transcription factor, Sox9. *Mech Develop* **140**, 25–40 (2016).
- Perlman, K. *et al.* Developmental trajectory of oligodendrocyte progenitor cells in the human brain revealed by single cell RNA sequencing. *Glia* (2020) doi:10.1002/glia.23777.
- Vladoiu, M. C. *et al.* Childhood cerebellar tumours mirror conserved fetal transcriptional programs. *Nature* **572**, 67–73 (2019).
- Haldipur, P. *et al.* Spatiotemporal expansion of primary progenitor zones in the developing human cerebellum. *Science* **366**, 454–460 (2019).
- Yeung, J. *et al.* Wls Provides a New Compartmental View of the Rhombic Lip in Mouse Cerebellar Development. *J Neurosci* **34**, 12527–12537 (2014).
- Max-Planck-Institute of Biophysical Chemistry. GenePaint. <https://gp3.mpg.de/> (2004).

22. Petukhov, V. et al. Cell segmentation in imaging-based spatial transcriptomics. *Nat Biotechnol* **40**, 345–354 (2022).
23. Biancalani, T. et al. Deep learning and alignment of spatially resolved single-cell transcriptomes with Tangram. *Nat Methods* **18**, 1352–1362 (2021).
24. Phipson, B. et al. propeller: testing for differences in cell type proportions in single cell data. *Bioinformatics* **38**, 4720–4726 (2022).
25. Simmons, S. Cell Type Composition Analysis: Comparison of statistical methods. *bioRxiv* 2022.02.04.479123 (2022) doi:10.1101/2022.02.04.479123.
26. Büttner, M., Ostner, J., Müller, C. L., Theis, F. J. & Schubert, B. scCODA is a Bayesian model for compositional single-cell data analysis. *Nat Commun* **12**, 6876 (2021).
27. Huang, X. et al. Single cell, whole embryo phenotyping of pleiotropic disorders of mammalian development. *bioRxiv* 2022.08.03.500325 (2022) doi:10.1101/2022.08.03.500325.
28. Dann, E., Henderson, N. C., Teichmann, S. A., Morgan, M. D. & Marioni, J. C. Differential abundance testing on single-cell data using k-nearest neighbor graphs. *Nat Biotechnol* **40**, 245–253 (2022).
29. Zhao, J. et al. Detection of differentially abundant cell subpopulations discriminates biological states in scRNA-seq data. *bioRxiv* 711929 (2020) doi:10.1101/711929.
30. Team, S. D. RStan: the R interface to Stan. *R package version 2.21.2* <http://mc-stan.org/> (2020).
31. Meredith, M. & Kruschke, J. HDInterval: Highest (Posterior) Density Intervals. *R package version 0.2.2*. <https://CRAN.R-project.org/package=HDInterval> (2020).
32. Uhlén, M. et al. Tissue-based map of the human proteome. *Science* **347**, 1260419–1260419 (2015).
33. Liu, J. et al. Jointly defining cell types from multiple single-cell datasets using LIGER. *Nat Protoc* **15**, 3632–3662 (2020).
34. Bakken, T. E. et al. Comparative cellular analysis of motor cortex in human, marmoset and mouse. *Nature* **598**, 111–119 (2021).
35. Melville, J. uwot: The Uniform Manifold Approximation and Projection (UMAP) Method for Dimensionality Reduction. *R package version 0.1.10*. <https://CRAN.R-project.org/package=uwot> (2020).
36. González-Blas, C. B. et al. SCENIC+: single-cell multiomic inference of enhancers and gene regulatory networks. (2022) doi:10.1101/2022.08.19.504505.
37. Aibar, S. et al. SCENIC: single-cell regulatory network inference and clustering. *Nat Methods* **14**, 1083–1086 (2017).
38. Sande, B. V. de et al. A scalable SCENIC workflow for single-cell gene regulatory network analysis. *Nat Protoc* **15**, 2247–2276 (2020).
39. Buffo, A. & Rossi, F. Origin, lineage and function of cerebellar glia. *Progress in Neurobiology* **109**, 42–63 (2013).
40. Leung, A. W. & Li, J. Y. H. The Molecular Pathway Regulating Bergmann Glia and Folia Generation in the Cerebellum. *Cerebellum* **17**, 42–48 (2018).
41. Cerrato, V. et al. Multiple origins and modularity in the spatiotemporal emergence of cerebellar astrocyte heterogeneity. *Plos Biol* **16**, e2005513 (2018).
42. Allen, D. E. et al. Fate mapping of neural stem cell niches reveals distinct origins of human cortical astrocytes. *Science* **376**, 1441–1446 (2022).
43. He, Z. et al. Lineage recording in human cerebral organoids. *Nat Methods* **19**, 90–99 (2022).
44. Ábrahám, H., Tornóczky, T., Kosztolányi, G. & Seress, L. Cell formation in the cortical layers of the developing human cerebellum. *Int J Dev Neurosci* **19**, 53–62 (2001).
45. Cardoso-Moreira, M. et al. Gene expression across mammalian organ development. *Nature* **571**, 505–509 (2019).
46. Human Developmental Biology Resource (HDBR). HDBR Atlas. <http://hdbratlas.org>.
47. Kerwin, J. et al. The HUDSEN Atlas: a three-dimensional (3D) spatial framework for studying gene expression in the developing human brain. *J Anat* **217**, 289–299 (2010).
48. Gerrelli, D., Lisgo, S., Copp, A. J. & Lindsay, S. Enabling research with human embryonic and fetal tissue resources. *Development* **142**, 3073–3076 (2015).
49. Braun, E. et al. Comprehensive cell atlas of the first-trimester developing human brain. *bioRxiv* 2022.10.24.513487 (2022) doi:10.1101/2022.10.24.513487.
50. Manno, G. L. et al. Molecular architecture of the developing mouse brain. *Nature* **596**, 92–96 (2021).
51. Luecken, M. D. et al. Benchmarking atlas-level data integration in single-cell genomics. *Nat Methods* **19**, 41–50 (2022).
52. Song, Y., Miao, Z., Brazma, A. & Papatheodorou, I. Benchmarking strategies for cross-species integration of single-cell RNA sequencing data. *bioRxiv* 2022.09.27.509674 (2022) doi:10.1101/2022.09.27.509674.
53. Kuo, A., Hansen, K. D. & Hicks, S. C. Quantification and statistical modeling of Chromium-based single-nucleus RNA-sequencing data. *bioRxiv* 2022.05.20.492835 (2022) doi:10.1101/2022.05.20.492835.

Reviewer Reports on the First Revision:

Referees' comments:

Referee #1 (Remarks to the Author):

The authors have addressed my comments, and I support publication.

Referee #3 (Remarks to the Author):

Manuscript by Sepp et al has been submitted to Nature after revision. The authors have been responsive to prior critiques, and their efforts to validate profiles of gene expression using spatial transcriptomics and candidate gene in situ hybridization in an attempt to reaffirm their conclusions about cellular abundance and gene expression changes in the developing cerebellum.

While the manuscript has been substantially revised, and while additional datasets and analyses have been incorporated, I have lingering concerns about the novelty presented in this manuscript. I also have concerns about the validity of the claims made by the authors throughout the manuscript, and concerns about several of the conceptual approaches taken. The manuscript still reads as a piecemeal of different vignettes and loosely tied analysis, and currently does not provide major new insights into human cerebellum development, neurodevelopmental disorders, or evolution.

Major comments:

First, the authors state that they identify 25 cell types divided into 43 cell states, and that 12 states can be subdivided into 48 subtypes. This means that their analysis identifies 79 cell states, but neither Fig 1b,c nor Extended Data Fig. 2c helps me understand what those states are, where they are on the t-SNE, which samples, replicates or species are they found in, and the smFISH experiment only tests for more than 70 marker genes, it is hard to know how many exactly. This means that there are some cell states that are not captured. In Figure 1d, they only highlight 8. The incompleteness of the data and lack of clarity in the presentation is very difficult to follow or understand what the main findings are. The lack of transparency is a major point of concern as it pertains to the significance of the dataset.

Many steps of the analysis are very opaque in the manuscript. For example, description of the approach for comparing gene expression profiles across species is virtually non-existent in the main text, and the way the analysis is conducted is conceptually flawed. The integrative analysis and consensus classification is generated based on expression of orthologous genes only, and that draws the investigators to conclude that cell types are highly conserved. But if they never consider what the contribution of non-orthologous genes, non-conserved isoforms, segmental duplications, or even inconsistencies in transcript annotation might be, they simply don't test the hypothesis. Therefore, I have serious reservations about the very fundamental aspects of data analysis in this manuscript, which undermine my confidence in the study and the claims made by the authors.

The authors say that they haven't been able to reliably distinguish ependymal cells in human specimens, indicating that what is being presented as a comprehensive atlas is in fact highly incomplete, with no explanation provided.

The authors propose differences in the dynamics of Purkinje neuron production represent major evolutionary difference, but there are several problems with this analysis. One, they acknowledge that dissection artifacts could contribute to unexpected differences in cell abundance estimates. Two, estimates of cell proportions from single cell RNA seq are classically known to be problematic.

The only way to resolve this problem would be by conducting a more rigorous quantification of these cell types in primary tissue. Moreover, on a conceptual level it is well known that the timing of genesis of various cell types differs across species, and therefore the proportional abundance of Purkinje neurons at a particular stage of development may not be particularly surprising. The overall profiles of cell type emergence based on Figure 1e seem very similar, and I do not believe Bayesian modeling is suitable for this kind of problem given the limited number of biological replicates at these various time points.

As far as I can tell, the in situ hybridization was conducted in a single biological specimen, and in a single species. While this provides some additional information not present in the original manuscript, I am not convinced that it provides robust or substantially novel insights.

Minor comments:

"Across the three species, we identified 25 cell types divided into 43 cell states, and for 12 cell states, we further split the cells into 48 subtypes" – it is unclear what these interpretations are based on. What defines cell type versus state and subtype. The authors should explain this further or use a different terminology.

Data quality of at least some of the RNAseq data seems low. If the number of UMIs is on average 2354 UMI per cell, that is lower than most high quality studies.

TNC expressing progenitors have been shown to be neurogenic in other parts of the CNS. Why TNC expression is linked to gliogenic progenitor identity by the authors is unclear.

The authors claim to have identified progenitor cells in the prospective white matter based on RNA localization of SOX2, NOTCH1, and PAX3. To my knowledge these transcripts are also found in astrocytes, therefore, to support their claims, the authors should provide more direct evidence that these cells are indeed progenitors.

Analysis of transcription factor enrichment is based on a single model from SCENIC, and predicted differences in transcription factor activity across species are neither discussed nor validated. The interpretative value of the figure seems highly limited to a limited set of experts.

Analysis of genes with low LOEUF score in Figure 3 seems completely disconnected from the rest of the paper and of unclear significance. Are the authors implying that the selective pressure is higher on genes enriched in developing cerebellum. It is formally possible that loss of function of those genes leads to deleterious effects in other developmental processes outside the cerebellum and would still show the enrichment. Significance of this analysis, which furthermore seems to be limited to select few cell types of interest, is completely unclear.

The authors claim that MAML2 "evolved the strongest changes in expression trajectories". How they distinguished evolved changes from drift with no selective pressure is unclear, and represents yet another clear example of the highly flawed reasoning behind this study.

Claims pertaining to the more recent evolution of CPLX4 and ZP2 are potentially interesting but underdeveloped. The authors primarily rely on analysis of bulk expression data, but provide no evidence that these gene expression differences are adaptive, contrary to what they claim in the manuscript.

Referee #4 (Remarks to the Author):

General part

- The web resource has been improved. While the exact change I asked for was not made, I think the current visualization clarifies the matter.
- The authors clarify in a more nuanced way the exact discovery in the context of Purkinje cell expansion, while also admitting no experimental evidence is provided, which is acceptable considering the difficulty of such endeavor.

Major comment 1

- Excellent response, the histological visualization improves the confidence in the matching and overall clarity of the correspondence model to a wider audience. I appreciate the different images are shown with the right scaling, but maybe some of the smaller ones could benefit from having more room on the page to be more easily interpretable.

Major comment 2

- Extended figure 2k is very informative and transparent and allows us to track the different conclusions that can be drawn from different data (as one expects inevitably in research). Also, the corresponding clarification in the text improves the tone and sets a good precedent for the community on how to make this sort of claim.

Major comment 3

- Partially satisfactory but ok. The statement is a bit hidden in the method, but considering the compactness of the final manuscript, this might be ok.

Major comment 4

- Very good and complete response, which however, is particularly difficult to follow because one needs to be very accustomed to these population names, which is a bit of a learning curve. I followed most of the threads and lines of reasoning, but it is quite complex and overwhelming overall. The part on Purkinje that was the most important seems addressed.

Note: I am not a fan of Reviewer Figures (See Reviewer Fig. 4.1-3), I think each figure contributing to the final acceptance should be part of a display item of the original paper if none of the derived described in the main text (but I understand the journal policy might be difference)

Major comment 5

- Again here is a Reviewer Figure, for which I have the opinion stated above.
Overall this is a satisfactory answer

Major comment 6

- The authors claim in the response that proportion analysis is inherently difficult, and it is true. However, there are very smart approaches to control for that, many of such cool ideas are used in the Cocoa analysis tool from the Kharchenko lab, I think it could increase confidence with the differential proportion analysis.

The other aspects of the response are satisfactory.

Major comment 7

- Convincing response.

Major comment 8

- This comment is not super deeply tackled, but maybe this is acceptable considering the amount of care for the other points. I am not particularly worried here.

Major comment 9

- Succinct but to the point answer. Satisfactory.

Major comment 10

- The response is thoughtful and while I do not agree on all points, overall, I accept it. I think some parts Reviewer figure 4.7 should be promoted to an Extended data figure, I was not able to find all panels.

Author Rebuttals to First Revision:

Response to the referees, round 2

We thank the referees for thoroughly reviewing our manuscript. We have made every attempt to address their remaining concerns. We have also edited the manuscript text and figure panels in response to editorial requests.

To facilitate readability, we use various font emphases and colours, as outlined below:

The comments from the referees are in bold;

our responses are in normal text;

changes to the manuscript are highlighted in yellow.

Referee #1

The authors have addressed my comments, and I support publication.

We thank the reviewer for the constructive criticism in the first review round, which we believe resulted in an improved manuscript.

Referee #3

SUMMARY

Manuscript by Sepp et al has been submitted to Nature after revision. The authors have been responsive to prior critiques, and their efforts to validate profiles of gene expression using spatial transcriptomics and candidate gene in situ hybridization in an attempt to reaffirm their conclusions about cellular abundance and gene expression changes in the developing cerebellum.

While the manuscript has been substantially revised, and while additional datasets and analyses have been incorporated, I have lingering concerns about the novelty presented in this manuscript. I also have concerns about the validity of the claims made by the authors throughout the manuscript, and concerns about several of the conceptual approaches taken. The manuscript still reads as a piecemeal of different vignettes and loosely tied analysis, and currently does not provide major new insights into human cerebellum development, neurodevelopmental disorders, or evolution.

The lingering concerns expressed by the reviewer were unexpected to us, considering their prior feedback: *"Both data generation and bioinformatic analyses are well executed using well-established methods. The manuscript is well written and easy to follow, and the dataset is likely to be useful for the community. My main reservation relates to the issue of novelty."*

Notably, we meticulously explained the novelty of our study in our response to the reviewer's previous feedback (Round 1, Major comment 1) and highlighted the following points:

- Unprecedented atlas
- High-resolution annotations
- Novel finding about Purkinje cell dynamics and diversity
- First comprehensive (cross-development/mammals) "evo-devo" single-cell study

The novelty-related issues raised by the reviewer in this round pertain to the comprehensiveness of the atlas (Major comment 3) and the relative abundances of Purkinje cells (Major comment 4).

Additionally, the reviewer introduces new comments on analyses that were integral to our original manuscript (Major comments 1 and 2, Minor comments 1, 2 and 6) and were previously positively acknowledged by the reviewer as well-executed and easy to follow.

Furthermore, there are several comments that seem unjustified to us, particularly regarding the analyses based on orthologous genes (Major comment 2), use of Bayesian modelling (Major comment 3), and the comment on “evolved changes” (Minor comment 7).

Only a few comments address the newly added analyses and results (Major comment 5, Minor comments 3-5).

The table below gives an overview of these aspects and our revisions to the manuscript. Our point-by-point response to the comments is in what follows thereafter.

COMMENT	KEYWORDS	NOTE	REVISION PLAN
Major comment 1	Atlas presentation	previously positively acknowledged by the reviewer	new Supplementary table, minor revision of the text
Major comment 2	Conceptual flaw	previously positively acknowledged; unjustified	optimization of the text
Major comment 3	Atlas comprehensiveness	unjustified	minor revision of the text
Major comment 4	Purkinje abundances	partially unjustified	new analysis with external data to further bolster findings
Major comment 5	Spatial data	unique new data used for validation	-
Minor comment 1	Cell type/state terminology	issue not raised in the first round	-
Minor comment 2	Data quality	previously positively acknowledged; unjustified	-
Minor comment 3	Specific marker gene used	marker gene used in spatial transcriptomic analysis	-
Minor comment 4	Specific marker genes used	marker genes used in spatial transcriptomic analysis	minor revision of the text
Minor comment 5	SCENIC analysis	proposed by reviewer in the first round; misunderstanding	minor revision of the text
Minor comment 6	Functional constraints	misunderstanding	minor revision of the text
Minor comment 7	“Evolved changes”	unjustified; misunderstanding	minor revision of the text

MAJOR COMMENTS

Major comment 1: First, the authors state that they identify 25 cell types divided into 43 cell states, and that 12 states can be subdivided into 48 subtypes. This means that their analysis identifies 79 cell states, but neither Fig 1b,c nor Extended Data Fig. 2c helps me understand what those states are, where they are on the /UMAP, which samples, replicates or species are they found in, and the smFISH experiment only tests for more than 70 marker genes, it is hard to know how many exactly. This means that there are some cell states that are not captured. In Figure 1d, they only highlight 8. The incompleteness of the data and lack of clarity in the presentation is very difficult to follow or understand what the main findings are. The lack of transparency is a major point of concern as it pertains to the significance of the dataset.

It seems that the reviewer has overlooked the figure panels and tables that describe the cell states and types in the snRNA-seq atlases, and the genes included in the smFISH panel.

The atlas is presented in several figures of the manuscript (Fig. 1-2, Extended Data Fig. 3-6):

- Cell type level annotations are presented in Fig. 1.
- Cell state level annotations are presented in Extended Data Fig. 3c, 4c, 5c that show the VZ, RL/NTZ, and RL/EGL lineage UMAPs, respectively.

- Subtype level annotations are presented in Fig. 2a, Extended Data Fig. 3o, 4g, 5f, and 6d that show the UMAPs of Purkinje cells, interneurons, RL/NTZ, RL/EGL, and glial cells, respectively. In Extended Data Fig 6d, three cell state categories that were not divided into subtypes are additionally indicated (ependymal progenitor, ependymal, oligodendrocyte).
- Extended Data Fig. 2c, schematically summarises all the cell types and states detected in the datasets.
- Extended Data Fig. 3a, 4a, 5a, 6a list, in an hierarchical manner, the different annotation categories for the VZ, RL/NTZ, RL/EGL, and glial cells, respectively.
- Supplementary Tables 2-4 list for each subcluster and each species the annotation at the level of cell type, state, and subtype.

We strongly believe that presenting the cell state and subtype level annotations separately for subsets of the datasets improves the interpretability of the visualisations. Considering the complexity of the datasets, displaying the combined cell state and subtype level annotation categories on the global UMAPs becomes challenging to interpret (**Reviewer Fig. 1**). We would also like to highlight that visualisations at different annotation levels are conveniently accessible through our online resource (<https://apps.kaessmannlab.org/sc-cerebellum-transcriptome/>).

As for the description of the smFISH gene set, all genes were listed in Supplementary Table 5 (now Supplementary Table 6). This table states the purpose of each included gene (marker or expression difference), specifies the nature of the expression difference and the affected cell types, and summarises the detection in our human spatial dataset as well as public mouse datasets (GenePaint, Allen Developing Mouse Brain Atlas). The gene set includes 74 marker genes and 26 genes that display expression differences between human and mouse.

In the selection of the marker genes, we indeed aimed to cover as many cell categories as possible. Note that not all categories that are present in our snRNA-seq atlases - which cover embryonic, fetal, and postnatal development - are expected to be present at the single stage of 12 wpc. Additionally, we did not aim to target the cell types from other brain regions, unresolved categories, and rare immune types (GABA_MB, NTZ_mixed, nonparenchymal macrophages, T cells). The number of remaining categories present with more than 1 cell in the 11 wpc snRNA-seq data (the closest stage for which we have snRNA-seq data) is 46, which is lower than the number of marker genes included in the panel (74).

In Fig. 1d, we display the spatial distribution of 8 categories at the level of cell type. There are several reasons why:

- Fig. 1 describes the atlas at the level of cell types, so we prefer to keep the presentation of the spatial mapping consistent with the other panels in the figure.
- These 8 cell types represent the main cerebellar neural cell types that are also included in our analyses of presence/absence expression differences.
- The imputation of cell labels is more confident at the level of cell types than at the level of states/subtypes. Given that the distinction of many of the states/subtypes relies only on a few markers included in the panel, we prefer visualisation of individual markers for their mapping (Extended Data Fig. 3q, 4l,m, 5e,j,l, 6h,j)). The only exception is the mapping of Purkinje subtypes, for which we present both individual marker expression patterns and the distribution of imputed labels (Fig. 2f,g). Importantly, this was also stated in the Main text (lines 91-92) :*“This allowed us to map the spatial expression patterns of >70 marker genes and locate the cell types within the cerebellum (Fig. 1d, Extended Data Fig. 3-6, Supplementary Table 5).”*

Reviewer Fig. 1. Uniform Manifold Approximation and Projection (UMAP) of 115,282 mouse, 180,956 human and 99,498 opossum cells coloured by their precisest label (subtype, if assigned -> cell state, if assigned -> cell type). The broad neuronal lineages are shown with arrows. In the legend, coloured dots indicate the precisest labels, and all categories are ordered into cell types (lines) coloured as in Fig. 1b.

Changes to the manuscript

- To further improve the presentation of the annotations, we provide a new supplementary table (Supplementary Table 5) that summarises the cell numbers in each annotation category by replicates, as requested by the Reviewer.
- We specify the exact number of marker genes in the Main text:

Lines 85-89: *As a validation of the annotations, we mapped spatial expression patterns of 74 marker genes using multiplexed single molecule fluorescence in situ hybridisation (smFISH) in the 12-week post-conception (wpc) human cerebellum, and located the cell types by aligning the spatial data with our snRNA-seq data (Fig. 1d, Extended Data Fig. 3-6, Supplementary Table 6).*

- We further improved the description of the approach used for the presentation of imputed cell type labels of the spatial dataset in the Methods section, to make it more transparent:

Lines 983-993: *For the presentation of imputed cell type labels in Fig. 1d and Extended Data Fig. 11f, we required at least 15 transcripts and an area of >500 pixels (9.522 μm^2) per segment, and filtered out outliers that had the transcript numbers or area larger than the sum of the 3rd quantile and 1.5 times the interquartile range ($n_{\text{transcripts}} > \sim 140$; $\text{area} > \sim 7040$ pixels (134 μm^2)). As a result we kept 87,140 (73%) of the initial 119,059 Baysor-estimated segments. The median metadata prediction score (Tangram) of the selected segments was 0.072 in case of cell type imputation and 0.050 in case of precisest label imputation. The prediction scores varied per group, for instance among the more abundant cell types (>1000 segments) the median score was the highest for the predicted Purkinje cell segments (0.084) and the lowest for GC/UBC segments (0.042). Similarly, among the precisest labels, the prediction score was the highest for the most abundant categories Purkinje_defined_EB (0.062) and Purkinje_defined_LB (0.046). Therefore, we standardised the prediction scores per group, and further filtered for the segments with a prediction score above the 1st quantile. As a result, we confidently assigned cell type labels to 65,355 (75%) of the 87,140 segments. For the predicted precisest labels presented in Fig. 2g (Purkinje subtypes), we required the segments to have a concordant cell type assignment (Purkinje cell). 36,531 (89%) of the 41,036 Purkinje cell segments had concordant and high confidence precisest label assignments, with 29,560 segments assigned as Purkinje_defined_EB and 6,971 as Purkinje_defined_LB.*

- Additionally, to provide easy access to our spatial dataset we have now added the possibility to visualise the gene expression patterns and imputed cell type labels via our online resource: <https://apps.kaessmannlab.org/sc-cerebellum-transcriptome/>

Major comment 2: Many steps of the analysis are very opaque in the manuscript. For example, description of the approach for comparing gene expression profiles across species is virtually non-existent in the main text, and the way the analysis is conducted is conceptually flawed. The integrative analysis and consensus classification is generated based on expression of orthologous genes only, and that draws the investigators to conclude that cell types are highly conserved. But if they never consider what the contribution of non-orthologous genes, non-conserved isoforms, segmental duplications, or even inconsistencies in transcript annotation might be, they simply don't test the hypothesis. Therefore, I have serious reservations about the very fundamental aspects of data analysis in this manuscript, which undermine my confidence in the study and the claims made by the authors.

Unfortunately, there is no space in the main text to describe the methodological details, which were, however, comprehensively described in the Methods section (lines 1221-1306). But we agree that a brief description of the approach in the Results section is helpful.

Changes to the manuscript

- we added a sentence that highlights the main aspects of our methodological approach:

Line 352-354: *To mitigate technical biases in cross-species expression level comparisons from snRNA-seq data, we took a conservative approach: we analysed exonic read pseudobulks of cell types and replicates, considered only the orthologous genes with comparable genomic annotation in the three species, assessed relative expression levels within each species, and required at least 5-fold differences in absolute expression levels to call a difference between species (Methods, Extended Data Fig. 10e-h).*

We have full confidence in the conceptual integrity of our approach. Identification of differences between the species is one of the principal aims of our study. However, to achieve this, it is extremely important to ensure that we are comparing "apples to apples". This motivated us to meticulously align datasets across species in terms of developmental stages and cell type categories. Achieving this alignment necessitates the utilisation of orthologous genes, as it is not possible to compare the expression of lineage/species-specific genes between species. We would like to stress that in the space of 1:1 orthologous genes, corresponding cell types show the highest similarity; e.g., Purkinje cells in human are more similar to Purkinje cells than interneurons in mouse, demonstrating that 1:1 orthologous gene expression programs in the developing cerebellar cell types are largely shared between the species. Nonetheless, this does not preclude the existence of differences between the species. These differences include, as mentioned by the reviewer, the contributions from lineage-specific genes and isoforms, whose analysis is beyond the scope of the current study, which is already extensive. Furthermore and importantly, the results of our study demonstrate that there

are also many 1:1 orthologous genes that differ in expression specificities (species-specific markers, Fig. 3b,c), expression trajectories (Fig. 4a-c), or even expression presence (Fig. 4e) in the cerebellar cell types. We thus conclude that even though the cell types are overall conserved (i.e., can be matched across species based on their transcriptome signatures), there are differences in the expression patterns of many genes, and at least some (but not all, see our response to Minor comment 7) of these differences might be associated with species-specific adaptations.

Changes to the manuscript

- We made every effort to further optimize the presentation of these aspects throughout the manuscript.
- Additionally, we expanded the Discussion section to outline the limitations of our study, particularly in relation to differences in lineage-specific genes and isoform usage between the species:

Lines 440-446: *This pattern is consistent with the progressively increasing molecular divergence of the cerebellum (and other organs) between species during development due to overall decreasing purifying selection, which enables drift and facilitates adaptations driven by positive selection^{19,52}. A limitation of our study is that we did not evaluate lineage-specific genes and isoforms, which additionally contribute to the transcriptome differences between the species. Moreover, further work is required to distinguish between adaptive changes, driven by positive selection, and changes resulting from genetic drift; and to assess the potential functional relevance of individual expression shifts in the context of interspecies phenotypic differences.*

Major comment 3: The authors say that they haven't been able to reliably distinguish ependymal cells in human specimens, indicating that what is being presented as a comprehensive atlas is in fact highly incomplete, with no explanation provided.

Our dataset represents the hitherto largest and most complete atlas of human cerebellum development. Ependymal cells are rare and, importantly, not focal to the study. Sampling ependymal cells in numbers that would be sufficient for tracking their developmental and evolutionary gene expression programs (in all species), would require production of datasets that are either orders of magnitude larger or specifically enriched for the ependymal cell type lineage. This requires dedicated studies.

Changes to the manuscript

- We more clearly acknowledge the limitations of our study, concerning the sampling of rare cell types and subtypes in all species, in the Discussion.

Lines 411-417: *Based on our snRNA-seq atlases of ~400,000 cells from the mouse, human, and opossum cerebellum, we established a consensus classification of the cellular diversity in the mammalian cerebellum and identified gene sets that underlie core ancestral transcriptional programs of cell fate specification in the cerebellum. Although a few rare cell (sub)type categories were not recovered in all studied species due to technical limitations, our analyses revealed that the overall cellular architecture of the developing cerebellum is similar across therian mammals, consistent with the previously posited conservation of its developmental program throughout amniotes^{4,49}.*

Major comment 4: The authors propose differences in the dynamics of Purkinje neuron production represent major evolutionary difference, but there are several problems with this analysis.

- (i) One, they acknowledge that dissection artifacts could contribute to unexpected differences in cell abundance estimates.
- (ii) Two, estimates of cell proportions from single cell RNA seq are classically known to be problematic. The only way to resolve this problem would be by conducting a more rigorous quantification of these cell types in primary tissue.
- (iii) Moreover, on a conceptual level it is well known that the timing of genesis of various cell types differs across species, and therefore the proportional abundance of Purkinje neurons are a particular stage of development may not be particularly surprising.

(iv) The overall profiles of cell type emergence based on Figure 1e seem very similar, and I do not believe Bayesian modeling is suitable for this kind of problem given the limited number of biological replicates at these various time points.

- (i) The possibility of dissection differences is the reason why we only compared the relative abundances at the level of cell types, and focussed on earlier developmental stages that are less affected by sampling differences.
- (ii) We note that compared to single-cell RNA-seq data, estimation of cell type abundances from single-nucleus RNA-seq data - as done in our study - has been shown to be more reliable (Bakken et al. 2018). Importantly, any bias introduced during nuclei preparation would be expected to influence the relative abundances in similar ways in all three species. We consider consistent methodology, used for the generation of the datasets for the three species, as one of the major advantages of our study.

Nevertheless, we agree with the reviewer that quantification of Purkinje cell abundances in more individuals would increase the confidence of our findings. Therefore, we have performed a meta-analysis of available mouse and human cerebellum datasets. Importantly, the results from this analysis agree with the results based on our data only (Reviewer Fig. 2).

Changes to the manuscript

- We provide a meta-analysis of Purkinje cell abundances in the available mouse and human datasets (Reviewer Fig. 2) and have incorporate this into Extended Data figure 7f, and revised the text accordingly.

Extended Data Fig. 7f. Purkinje cell relative abundances in available human and mouse cerebellum datasets^{10,11,13,21,22}. The estimation of abundances is based on the annotations reported in the original studies, except for Braun et al., where cell type annotations were not provided in the original study and were instead transferred from our human dataset (Methods).

Main text, lines 159-165: *A meta-analysis of 19 mouse (E13.5-E15.5) and 20 human (8-11 wpc) cerebellum samples from this and other studies^{10,11,13,21,22} confirmed the significantly higher Purkinje cell abundances in human (Welch tests; Extended Data Fig. 7f). This change in Purkinje cell dynamics in the human lineage could be related to differences in developmental durations between species and/or the unique presence of basal progenitors in the human cerebellum⁹ that may serve as an additional pool of Purkinje cell progenitors. Altogether, our snRNA-seq atlases provide a comprehensive view of cerebellar cell types in mammals revealing the largely conserved developmental sequence of their generation but also notable differences in human Purkinje cell dynamics.*

Methods, lines 247-258: *Cell type annotations of mouse developing cerebellum scRNA-seq dataset by Smith et al.²¹ were downloaded from <https://www.ncbi.nlm.nih.gov/geo/query/acc.cgi?acc=GSE209915>. Cell type annotations of mouse developing cerebellum scRNA-seq datasets by Carter et al.¹⁰ and by Vladoiu et al.¹¹ were provided by the authors through personal communication. Processed data of human embryonic and fetal (6.6, 6.7, 6.9, 8, 8.1, 8.5 11.5, 12, and 14 wpc) cerebellum scRNA-seq data by Braun et al.²² were downloaded from <https://github.com/linmarsson-lab/developing-human-brain>. We used Seurat⁷⁰ v4.0.6 to transfer the cell type annotations from our human dataset to the Braun et al.²² cerebellum data. For each dataset, gene expression counts were scaled to 10,000 and log-normalised. After identifying the 5000 most highly variable genes for each dataset, we used their intersection (2694 genes) to identify anchors between the datasets with the function FindTransferAnchors().*

Annotations were then transferred from our dataset (reference) to Braun et al.²² (query) with the function `TransferData()` with `k.weight = 30` and `weight.reduction = "cca"`.

Methods, lines 283-289: For Extended Data Fig. 7f, we estimated the proportions of Purkinje cells using the cell type annotations provided by the original studies, if available. For Carter et al. mouse dataset, we additionally included cells originally annotated as GABA Progenitor among the Purkinje cell counts, as they express markers of differentiating Purkinje cells (*Lhx5*, *Foxp2*)¹⁰. For Vladiou et al. mouse dataset, we additionally included cells originally annotated as Brainstem progenitors among the Purkinje cell counts, as our previous analyses indicated that these represent differentiating Purkinje cells²⁵. For Braun et al. human dataset, cells with a transferred label Purkinje were included among the Purkinje cell counts.

- (iii) The analysis of our data shows that the developmental sequence of cell type genesis in the cerebellum is conserved across therian mammals. The relative abundances of Purkinje cells are about 2-fold higher in human 8-9 pcw cerebellum compared to the mouse and opossum cerebellum at corresponding stages. Such a big difference - now also validated based on the aforementioned meta-analysis - is interesting and notable, and implies that the early progenitor cells in the cerebellum give rise to relatively more Purkinje cells in human than in the other studied species. We are not aware of any previous study that has reported this. Furthermore, our observations about the shifts in Purkinje cell subtype ratios in humans underscore the functional relevance of this transient change in Purkinje cell relative abundances.
- (iv) Bayesian modelling is an established approach to compare cell type proportions in single cell datasets (Phipson et al. 2022, Büttner et al. 2021). Importantly, Bayesian modelling approaches were specifically developed to perform reliable comparisons based on a few biological replicates. Therefore, we are confident in the suitability of our approach. Nevertheless, as described in (ii), we now additionally provide a meta-analysis of Purkinje cell proportions that includes 20 human and 19 mouse samples. Frequentist statistics (Welch test) of these combined data demonstrates high statistical significance of the detected difference between mouse and human Purkinje cell relative abundances (Reviewer Fig. 2).

Major comment 5: As far as I can tell, the *in situ* hybridization was conducted in a single biological specimen, and in a single species. While this provides some additional information not present in the original manuscript, I am not convinced that it provides robust or substantially novel insights.

Our spatial dataset of a single 12 wpc human specimen provides supportive evidence for our main analysis of snRNA-seq data that is based on 78 libraries, including 31 human libraries - that is, these new data were used for additional validation. Additionally, we used public mouse ISH data (GenePaint, Allen Developing Mouse Brain Atlas) to examine spatial profiles in the mouse cerebellum (Supplementary Table 5).

In general, the use of large-scale ISH-based datasets is still limited, especially for humans, where it is already highly challenging to obtain suitable samples. Even a recent mouse brain atlasing study only relied on a single biological replicate: Langlieb et al. bioRxiv (Slide-seq; adult). We note that our 12 wpc spatial dataset will potentially represent one of the first multiplexed smFISH datasets of any region of the developing human brain. We are aware of only one more dataset (EEL FISH, 5 wpc human brain, 3 sections from a single embryo) included in a recent preprint on the first-trimester developing human brain by Braun et al. bioRxiv.

MINOR COMMENTS

Minor comment 1: “Across the three species, we identified 25 cell types divided into 43 cell states, and for 12 cell states, we further split the cells into 48 subtypes” – it is unclear what these interpretations are based on. What defines cell type versus state and subtype. The authors should explain this further or use a different terminology.

The use of the terms “cell type” and “cell state” follows the established conventions in the field (see, e.g., Osumi-Sutherland et al. 2021, *Nature Cell Biology*; Domcke and Shendure 2023, *Cell*). We also explain these concepts in:

- Extended Data Fig. 2a,
- the Main text (lines 81-85): *Consistent with the ongoing efforts in establishing cell ontologies¹⁷, we grouped the cells into broad lineages based on their developmental origin, into cell types (25 across the three species), into cell differentiation states (43; hereafter, cell states), and for 12 cell states that displayed remaining variability, we further split the cells into subtypes (36-38 in each species; Fig. 1b,c, Extended Data Fig. 2c).*
- and the Methods section (lines 172-174): *...we mostly use the term “type” to group cells committed to a distinct mature cell fate and “state” to refer to differentiation status that often form a continuum within each cell type category... .*

Minor comment 2: Data quality of at least some of the RNAseq data seems low. If the number of UMIs is on average 2354 UMI per cell, that is lower than most high quality studies.

Considering the used snRNA-seq methodology and the studied tissue, the number of UMIs in our study is comparable to other studies (**Reviewer Table 1**). In general, the number of UMIs detected is lower for single-nucleus (sn) RNA-seq data compared to single-cell (sc) RNA-seq data. The number of UMIs can also differ between cell types/states. Additionally, we used two different Chromium versions, which have different sensitivities: less UMIs are detected with Chromium v2 chemistry compared to Chromium v3 chemistry (Extended Data Fig. 1e). We note that usage of two different Chromium chemistries has also been inevitable in other large-scale studies (which take time; i.e., chemistries change), for instance Braun et al. bioRxiv. Most importantly, the quality of our datasets is high enough to facilitate the analyses presented in our study, and the datasets and annotations provide an unprecedented resource for further analyses on cerebellar development, evolution and associated diseases, as indeed already demonstrated by recent studies, which use our data and insights (Tan et al 2023, Science; Apsley and Becker 2022, Cells; Okonechnikov et al 2023, Neuro-Oncology).

Reviewer Table 1. Comparison of numbers of UMIs and genes detected between snRNA-seq datasets

Tissue/cells	Median number of UMIs/genes detected	
Adult human cerebellar hemispheres	this study: 4279/2447 (Chromium v3)	Siletti et al. bioRxiv : ~4500/2200 (Chromium v3)
Developing human Purkinje cells	this study: 1984/1422 (Chromium v2 and v3)	Aldinger et al. 2021* : 2067/1222 (SPLiT-seq)

* filtered as presented in Fig. 2e in our study

Changes to the manuscript

- Instead of reporting the overall median UMI counts in the Main text, we now report the median numbers of UMIs separately for each species and Chromium version in the Extended Data Fig. 1e, given that the overall median could be misleading.

Minor comment 3: TNC expressing progenitors have been shown to be neurogenic in other parts of the CNS. Why TNC expression is linked to gliogenic progenitor identity by the authors is unclear.

The enrichment of *TNC* in gliogenic progenitors was revealed by our data (Extended Data Fig. 6a). Annotation of gliogenic progenitors was not based on a single marker, and these cells express additional markers (e.g., *SLC1A3*) characteristic to astrocyte lineage cells including Bergmann glia. Note that *TNC* is also expressed at lower levels in bipotent progenitors, which give rise to both astrocytes and interneurons.

TNC was included in our spatial gene panel because of the availability of the probeset at the service provider (Resolve Biosciences).

Minor comment 4: The authors claim to have identified progenitor cells in the prospective white matter based on RNA localization of SOX2, NOTCH1, and PAX3. To my knowledge these transcripts are also found in astrocytes, therefore, to support their claims, the authors should provide more direct evidence that these cells are indeed progenitors.

We agree with the reviewer that these genes also mark astrocytes. Therefore, to pinpoint progenitors in the prospective white matter, we had additionally required the presence of *TOP2A* transcripts that mark mitotic cells (Extended Data Fig. 6i).

Changes to the manuscript

- We include *TOP2A* among the markers listed in the text:

Lines 236-238: *Spatial mapping of progenitors (SOX2, NOTCH1, PAX3, TOP2A) in the human 12 wpc cerebellum revealed their presence not only in the VZ and RL, but also scattered in the prospective white matter (PWM) and cortical transitory zone ...*

Minor comment 5: Analysis of transcription factor enrichment is based on a single model from SCENIC, and predicted differences in transcription factor activity across species are neither discussed nor validated. The interpretative value of the figure seems highly limited to a limited set experts.

We included the SCENIC analysis in response to this reviewer's previous comment (Round 1, Minor comment 6), but we agree that it has several limitations as also explained in our previous response. In our analyses we focussed on transcription factor (TF) expression and not activity since the estimation of TF activities based on gene expression data only (SCENIC) has several limitations: a limited proportion of a gene's regulatory space (up to 20 kb around the transcription start site) is considered in the detection of TF motifs, the quality of the motif database is not equivalent between species, and the regulons are built separately for each species. Therefore, it is important not to interpret "absence of evidence as evidence of absence": if a regulon is not identified by SCENIC in one of the species, it should not be interpreted as absence of TF activity in that species. This means that comparisons of TF activities can be made across cell states but not across the species.

Given these limitations, we only used the SCENIC analysis to provide supporting evidence to the relevance of the TFs that we highlighted based on their conserved expression specificity, as stated in the Main text (lines 279-282): "*Among all mouse and human TF markers, conservation of expression specificity is associated with higher expression levels of their predicted target genes in the respective cell states as revealed by SCENIC³⁶ modelling (Extended Data Fig. 8i,j, Supplementary Table 10).*"

Changes to the manuscript

- We state the limitations of the SCENIC analysis in the Extended data figure legend to avoid misinterpretations:

Line 1126-1129: *Note that activities of individual TFs modelled by SCENIC can be compared across cell states but not across species, given that the regulons were built separately for human and mouse.*

Minor comment 6: Analysis of genes with low LOEUF score in Figure 3 seems completely disconnected from the rest of the paper and of unclear significance. Are the authors implying that the selective pressure is higher on genes enriched in developing cerebellum. It is formally possible that loss of function of those genes leads to deleterious effects in other developmental processes outside the cerebellum and would still show the enrichment. Significance of this analysis, which furthermore seems to be limited to select few cell types of interest, is completely unclear.

We do not imply that the selective pressure is higher on genes enriched in the developing cerebellum (compared to genes with dynamic expression in other brain regions). Instead, we demonstrate that the *in vivo* functional constraints (as measured by the LOEUF scores) are higher on genes that show dynamic expression during granule and/or Purkinje

cell differentiation compared to genes that are expressed in these cell types in a non-dynamic manner. This result is consistent with our previous studies on gene expression (bulk RNA-seq) across mammalian organ development (Cardoso-Moreira et al. 2019, Nature; Sarropoulos et al. 2019, Nature), where we showed that developmentally dynamic genes are under stronger functional constraints than non-dynamic genes, and the constraints increase with the number of organs in which genes show temporal dynamic expression, as also stated in the Main text (lines 294-297): *“The dynamic genes show low tolerance to heterozygous inactivation in human population³⁹, with those dynamic in both neuron types under the strongest functional constraint (Fig. 3e). This is in line with studies linking phenotypic severity to expression pleiotropy^{19,40}.”*

This analysis is important for coherence of the manuscript, given that the identification of genes with dynamic expression is an essential step for the downstream analyses on gene expression trajectories. The high constraints as well as enrichments for TFs and disease genes among the dynamic genes support the relevance of these genes in neuronal differentiation.

Minor comment 7: The authors claim that MAML2 “evolved the strongest changes in expression trajectories”. How they distinguished evolved changes from drift with no selective pressure is unclear, and represents yet another clear example of the highly flawed reasoning behind this study.

Claims pertaining to the more recent evolution of CPLX4 and ZP2 are potentially interesting but underdeveloped. The authors primarily rely on analysis of bulk expression data, but provide no evidence that these gene expression differences are adaptive, contrary to what they claim in the manuscript.

We are of course aware that evolutionary changes at the molecular level (“evolved changes”, as put by the reviewer), such as the observed changes in expression trajectories, may be (effectively) neutral (i.e., driven by genetic drift) or may be beneficial/adaptive (i.e., driven by positive selection). Notably, our sentence pertaining to MAML2 “...evolved the strongest changes in expression trajectories” does not distinguish between the two scenarios; that is we never excluded that this change occurred as a consequence of genetic drift.

Instead, in the Main text, we generally suggest that some of the observed expression differences (but not all) could underlie adaptive phenotypic differences between the species; that is, the genes with expression changes represent “candidates” for adaptive/selectively driven phenotypic change. Specifically, we stated: *“... our comparative molecular analyses of mammalian cerebellar development unveiled numerous candidate genes with potentially adaptive expression changes ...”*

Changes to the manuscript

- To make these aspects clear for the general audience, we improved our wording in the Discussion.

lines 440-446: *This pattern is consistent with the progressively increasing molecular divergence of the cerebellum (and other organs) between species during development due to overall decreasing purifying selection, which enables drift and facilitates adaptations driven by positive selection^{19,52}. A limitation of our study is that we did not evaluate lineage-specific genes and isoforms, which additionally contribute to the transcriptome differences between the species. Moreover, further work is required to distinguish between adaptive changes, driven by positive selection, and changes resulting from genetic drift; and to assess the potential functional relevance of individual expression shifts in the context of interspecies phenotypic differences.*

Referee #4

General part

- The web resource has been improved. While the exact change I asked for was not made, I think the current visualization clarifies the matter.

- The authors clarify in a more nuanced way the exact discovery in the context of Purkinje cell expansion, while also admitting no experimental evidence is provided, which is acceptable considering the difficulty of such endeavor.

We are glad that the reviewer found that these aspects have been improved.

MAJOR COMMENTS

Major comment 1. Excellent response, the histological visualization improves the confidence in the matching and overall clarity of the correspondence model to a wider audience. I appreciate the different images are shown with the right scaling, but maybe some of the smaller ones could benefit from having more room on the page to be more easily interpretable.

Thank you for the positive feedback. We have increased the size of the mouse and opossum images by 10%.

Major comment 2. Extended figure 2k is very informative and transparent and allows us to track the different conclusions that can be drawn from different data (as one expects inevitably in research). Also, the corresponding clarification in the text improves the tone and sets a good precedent for the community on how to make this sort of claim.

Thank you for the positive feedback.

Major comment 3. Partially satisfactory but ok. The statement is a bit hidden in the method, but considering the compactness of the final manuscript, this might be ok.

Major comment 4. Very good and complete response, which however, is particularly difficult to follow because one needs to be very accustomed to these population names, which is a bit of a learning curve. I followed most of the threads and lines of reasoning, but it is quite complex and overwhelming overall. The part on Purkinje that was the most important seems addressed.

Note: I am not a fan of Reviewer Figures (See Reviewer Fig. 4.1-3), I think each figure contributing to the final acceptance should be part of a display item of the original paper if none of the derived described in the main text (but I understand the journal policy might be difference)

Given the space restrictions we cannot add these figure panels to the manuscript.

Major comment 5. Again here is a Reviewer Figure, for which I have the opinion stated above.

Overall this is a satisfactory answer

We note that some of the panels in this reviewer figure were already included in Extended Data Fig. 4f.

Major comment 6. The authors claim in the response that proportion analysis is inherently difficult, and it is true. However, there are very smart approaches to control for that, many of such cool ideas are used in the Cocoa analysis tool from the Kharchenko lab, I think it could increase confidence with the differential proportion analysis.

The other aspects of the response are satisfactory.

Given the space and time restrictions, we have not implemented the Cocoa analysis tool in the current study. However, we performed a meta-analysis of available single cell datasets of the developing human and mouse cerebellum to provide further support to our results on altered Purkinje cell dynamics in human (see our response to Referee #3, Major comment 4, page 7)

Major comment 7. Convincing response.

Major comment 8. This comment is not super deeply tackled, but maybe this is acceptable considering the amount of care for the other points. I am not particularly worried here.

Major comment 9. Succinct but to the point answer. Satisfactory.

Major comment 10. The response is thoughtful and while I do not agree on all points, overall, I accept it. I think some parts Reviewer figure 4.7 should be promoted to an Extended data figure, I was not able to find all panels.

We have added panels (c) and (f) from the Reviewer Fig. 4.7 as Extended Data Fig. 11b and d.